# Extending resolution within a single imaging frame

Esley Torres-García [1,2], Raúl Pinto-Cámara [1,2], Alejandro Linares[2,3], Damián Martínez [2], Víctor Abonza[2], Eduardo Brito-Alarcón [2], Carlos Calcines-Cruz [4], Gustavo Valdés-Galindo[5], David Torres[2], Martina Jabloñski[6], Héctor H. Torres-Martínez [7], José L. Martínez[8], Haydee O. Hernández [9], José P. Ocelotl-Oviedo[2], Yasel Garcés[2,8], Marco Barchi [10], Rocco D'Antuono [11], Ana Bošković[12], Joseph G. Dubrovsky [7], Alberto Darszon[8], Mariano G. Buffone[6], Roberto Rodríguez Morales[13], Juan Manuel Rendon-Mancha[1], Christopher D. Wood [2], Armando Hernández-García[5], Diego Krapf [14], Álvaro H. Crevenna [12] & Adán Guerrero[2] ✉

The resolution of fluorescence microscopy images is limited by the physical properties of light. In the last decade, numerous super-resolution microscopy (SRM) approaches have been proposed to deal with such hindrance. Here we present Mean-Shift Super Resolution (MSSR), a new SRM algorithm based on the Mean Shift theory, which extends spatial resolution of single fluorescence images beyond the diffraction limit of light. MSSR works on low and high fluorophore densities, is not limited by the architecture of the optical setup and is applicable to single images as well as temporal series. The theoretical limit of spatial resolution, based on optimized real-world imaging conditions and analysis of temporal image stacks, has been measured to be 40 nm. Furthermore, MSSR has denoising capabilities that outperform other SRM approaches. Along with its wide accessibility, MSSR is a powerful, flexible, and generic tool for multidimensional and live cell imaging applications.

Super-resolution Microscopy (SRM), which encompasses a collection of methods that circumvent Abbe's optical resolution limit, has dramatically increased our capability to visualize the architecture of cells and tissues at the molecular level. There are several approaches to SRM which vary in terms of the final attainable spatial and temporal resolution, photon efficiency, as well as in their capacity to image live or fixed samples at depth[1,2]. Instrumentation-based techniques, such as SIM and STED, exceed the diffraction limit by engineering the illumination or the point spread function (PSF)[3–5]. These techniques can be used for live imaging although they require specialized hardware and dedicated personnel for maintenance and operation. Single-molecule localization microscopy (SMLM) methods (e.g., STORM, PAINT, PALM)[6–9] localize individual emitters with nanometer precision but require temporal analysis of several hundred-to-thousands of images

and are prone to error due to fast molecular dynamics within live specimens.

Some SRM computational methods have few or no demands on hardware or sample preparation and provide resolution improvements beyond the diffraction limit, i.e., fluorescence fluctuation-based super-resolution microscopy (FF-SRM) approaches[10–13]. Both, the quantity and performance of these methods have increased over the past decade given the advantages they present, such as their low barriers to entry and generic applicability to data acquired with a variety of microscopy modalities (widefield, confocal, or light-sheet). However, these methods also present some limitations, such as the possible introduction of artifacts[14], the requirement for high signal-to-noise ratio (SNR) data and the acquisition of tens to hundreds of frames[10–13], which limit their applicability to reconstruct fast dynamical processes.

The problem of spatial resolution in optical microscopy can be addressed from the statistical point of view. In the case of fluorescence microscopy, the process of photon emission from point sources (fluorescence emitters) can be considered as a discrete distribution of information, where the unitary element of the distribution is the photon[15]. In this scenario, the problem of spatial resolution gets reduced to the problem of finding modes of information, regardless of the shape of the distribution, hence, disconnecting the problem of optical resolution from the diffraction boundary[16].

Here, we introduce the Mean Shift Super-Resolution principle for digital images 'MSSR' (pronounced as *messer*), derived from the Mean Shift (MS) theory[17,18]. MSSR extends the resolution of any single fluorescence image up to 1.6 times, including its use as a resolution and contrast enhancement complement after the application of other super-resolution methods.

By computing the local magnitude of the Mean Shift vector, MSSR generates a probability distribution of fluorescence estimates whose local magnitude peaks at the source of information. As a result of that, the spatial distribution becomes 'refined' (i.e., for a Gaussian distribution of fluorescence its width shrinks). Additionally, we demonstrate the extended-, enhanced- and super-resolving capabilities of MSSR as a standalone method for a variety of fluorescence microscopy applications, through a single-frame and temporal stack analysis, allowing resolution improvements toward a limit of 40 nm.

Open-source implementations of MSSR are provided for ImageJ (as a plugin), Python, R, and MATLAB, some of which take advantage of the parallel computing capabilities of regular desktop computers (Supplementary Note 7). The method operates almost free of parameters; users only need to provide an estimate of the PSF (in pixels) of the optical system, choose the MSSR order, and decide whether a temporal analysis will take place (Supplementary Material and Supplementary Methods). The provided open-source implementations of MSSR represent a user-friendly alternative for the bioimaging community for unveiling life at its nanoscopic level.

## Results

### The MSSR principle

MSSR is tailored around the assumption that fluorescence images are formed by discrete signals collected (photons) from point sources (fluorophores) convolved with the PSF of the microscope (Supplementary Notes 1, 2 and 3). Processing a single image with MSSR starts with the calculation of the MS, which guarantees that large intensity values on the diffraction-limited (DL) image coincide with large positive values in the MSSR image (Supplementary Note 4). Further algebraic transformations then restore the raw intensity distribution and remove possible artifacts caused by the previous step (edge effects and noise dependent artifacts), giving rise to an image that contains centers of density with a narrower full width at half maximum (FWHM) (Fig. 1a). This procedure is denoted by MSSR of zero order ($MSSR^0$), and it is the first stage which shrinks emitter distribution.

The MS is locally computed by a kernel window that slides throughout the entire image, subtracts the sample mean (weighted local mean) as well as the central value of the kernel using a spatial-range neighborhood (Supplementary Notes 2 and 3, Supplementary Fig. S5, Supplementary Table S1)[17,18]. The MS is a vector that always points towards the direction of the intensity gradient and its length provides a local measure of the fluorescence density and brightness[19–21]; its magnitude depends on the value difference between the central pixel of the neighborhood and the surrounding pixels. A mathematical proof, provided in Supplementary Note 4, demonstrates that the minimum MS value, computed from a Gaussian distribution, matches with the point of maximum intensity of the initial distribution (Supplementary Note 4, Supplementary Fig. S6).

The increase in resolution offered by $MSSR^0$ was evaluated by the Rayleigh and Sparrow limits[22–24], which are two criteria that establish resolution bounds for two near-point sources (Fig. 1b). Processing with $MSSR^0$ of two point sources located at their resolution limit (2.5 σ and 2 σ for Rayleigh and Sparrow limit respectively, Fig. 1c vertical discontinuous lines) decreases the dip (height at the middle point)[25] within their intensity distributions (Fig. 1b, c). Processing a single image with $MSSR^0$ shifts the resolution limit by 26 and 20% according to the Rayleigh and Sparrow limits, respectively, and reduces the FWHM of individual emitters (Fig. 1c vertical continuous lines). A comparison of the shrinkability of $MSSR^0$ applied to Gaussian and Bessel PSFs are shown in Supplementary Fig. S9. The reduction of FWHM of Bessel PSF at different wavelengths of the visible spectrum are shown in Supplementary Fig. S10.

Since the result of $MSSR^0$ is an image, the resulting image is used to seed an iterative process (Fig. 2a). We refer to this as higher-order MSSR ($MSSR^n$, with $n > 0$), which delivers a further gain of resolution per $n$-iteration step (Fig. 2a and Supplementary Fig. S11). As the order of $MSSR^n$ increases, both the FWHM of emitters (Supplementary Fig. S12) and the dip of their intensity distribution decrease (Fig. 2b). Numerical approximations indicate that two point sources separated at 1.6 σ are resolvable with $MSSR^3$, but not when their separation is 1.5 σ (Fig. 2b). The separation of 1.6 σ sets the theoretical resolution limit of $MSSR^n$.

In summary, $MSSR^n$ processing extends the spatial resolution of single DL images. The procedure of applying $MSSR^n$ to a single DL image will be defined as sf-$MSSR^n$.

### MSSR is a deconvolution approach which operates at the nano scales

In optical microscopy, objects significantly closer to the diffraction boundary can be resolved with clever illumination and detection schemes (i.e., SIM, Airy Scan, 4 Pi, I5M, STED, SMLM, etc.)[3,26–28], or by careful image analysis, reviewed in[29]. Rayleigh criterion is conservative, in the sense that achieving a decrease of the Dip formed by the joint distribution shaped by two adjacent emitters might be interpreted as surpassing the diffraction boundary (Supplementary Fig. S13a,b). Repeating the same procedure using a joint distribution shaped by two adjacent emitters located at the Sparrow limit yields no further gain of resolution, as the dip remains constant, taking the value of 1 (Supplementary Fig. S13e,f).

MSSR aims to revert the effect of diffraction on optical microscopy, so it can be considered as a deconvolution process. As the diffraction can be modeled with a Gaussian spread, the pixel value is a superimposition of spreads from individual emitters (Supplementary Notes 1, 2 and 3). The goal is to reduce the spread. The latter can be accomplished by "sharpening by blur". In MSSR, the computation of the MS is the blurring process used to sharpen the image. What makes sf-$MSSR^n$ unique is the fact that it extends spatial information down below the Sparrow limit. Processing the joint distributions of Supplementary Fig. S13 with sf-MSSR of any order leads to a decrease of the dip value (Supplementary Fig. S13c, d, g, h). Furthermore, sf-$MSSR^3$ processing collapses the dip to zero for both Rayleigh and Sparrow conditions (Supplementary Fig. S13d, h).

To illustrate how MSSR works by sharpening features down the diffraction barrier, we provide comparative data against: Wiener deconvolution[30], Richardson-Lucy deconvolution[31,32] and the Radiality Maps (RMs)[11]. Supplementary Fig. S14 shows that Wiener deconvolution partially restores the effect of diffraction, but without a dramatic increase in spatial resolution. Interestingly, Richardson-Lucy deconvolution provides a noticeable increase in resolution at the boundaries of the Rayleigh limit but fails to extend spatial resolution down below the Sparrow limit (Supplementary Fig. S14).

Gustafsson et al. showed that the RMs of SRRF provide a resolution increase down to 0.7 times the Gaussian FWHM[11], when the peak

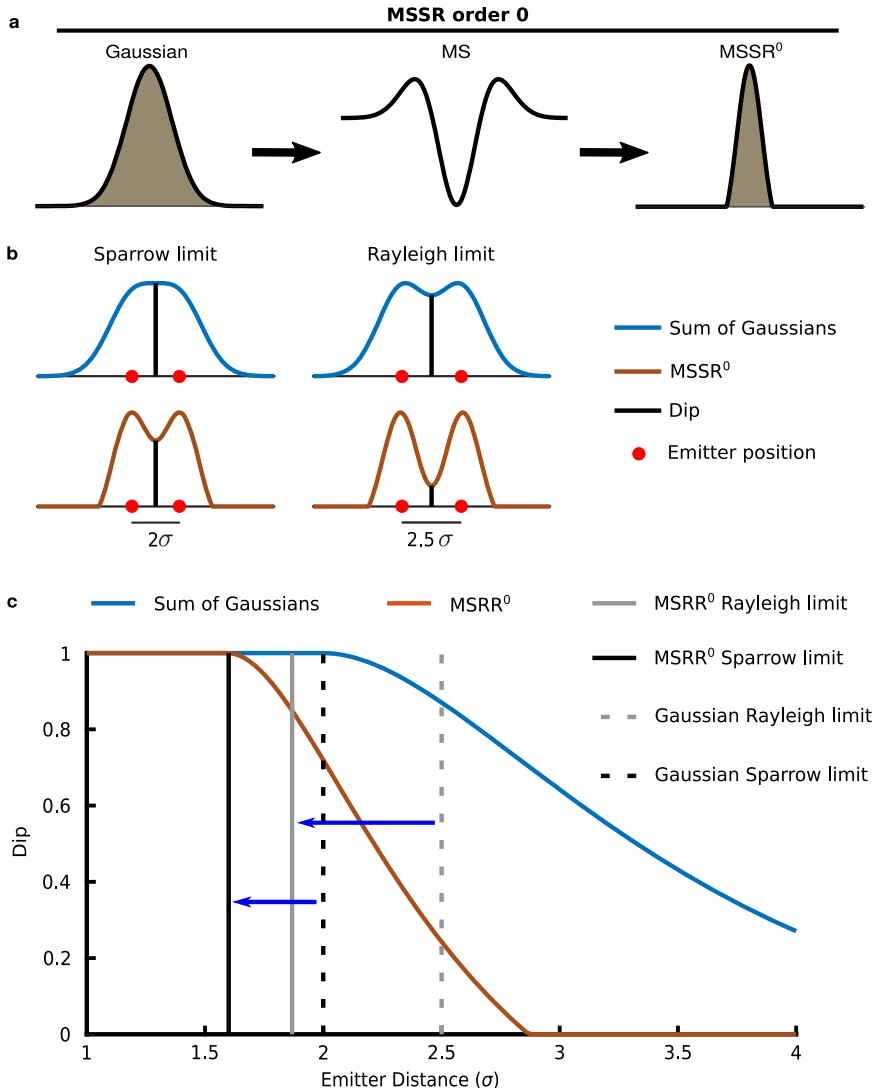

**Fig. 1 | MSSR of zero order increases resolution by reducing the width of the spatial distribution of photons from simulated fluorescent emitters. a** The MS is applied to the initial Gaussian distribution of photons emitted by a point source (left) resulting in a MS graph (center). Application of further algebraic transformations (see Supplementary Note 5 and Supplementary Fig S11 (ii–iv)) provides the $MSSR^0$ distribution (right). **b** Sparrow and Rayleigh limits (blue, DL) and the corresponding $MSSR^0$ transformation (brown) for two point sources. Red dots represent each emitter's location. The dip is indicated by a vertical black line. The inter-emitter distance is expressed as σ-times their individual standard deviation before MSSR processing. **c** Dip computed for two point source emitters of Gaussian distribution located away at distinct σ (blue line) where the corresponding $MSSR^0$ result is also depicted (red line). For Gaussian: Rayleigh limit−gray discontinuous line, Sparrow limit−black discontinuous line. For $MSSR^0$: Rayleigh limit−gray solid line, Sparrow limit−gray solid line. The solid vertical lines represent the distance between emitters such that when processed with $MSSR^0$, the Rayleigh and Sparrow criteria are met (for detail see Online Methods section Simulation of fluorescent emitters).

separation between isolated distributions is greater than 0.7 times the FWHM of the PSF, they can be directly resolved without further enhancement provided by higher-order statistical analysis. $MSSR^0$ and the RMs are similar in the sense that both perform sharpening and smoothing. Supplementary Fig. S14a shows that both MSSR and the RMs overcome the Rayleigh diffraction limit[22]. However, the RMs produce undesired artifacts which are absent when using $sf\text{-}MSSR^0$ (Supplementary Fig. S14b), which is in agreement with the reported spatial artifacts introduced[33]. Figures 1, 2 and Supplementary Fig. S14 show that $sf\text{-}MSSR^0$ reliably provides artifact-free spatial resolution gainsclose to the Rayleigh limit, hence, allowing the study of nanoscopic regimes at the boundaries of the Sparrow limit.

Such observations support the conclusion that $sf\text{-}MSSR^n$ is a deconvolution process that extends spatial information of DL images at the nano scales.

## MSSR extends spatial resolution in fluorescence microscopy images

To empirically test the ability of MSSR to extend spatial resolution within a single DL image, a commercial nanoruler sample (GATTA-SIM140B, GATTAquant) was imaged by Structured Illumination Microscopy (SIM) and Total Internal Reflection Fluorescence (TIRF) microscopy, which was then processed by $sf\text{-}MSSR^n$. The iterative processing of the TIRF image with $sf\text{-}MSSR^3$ reveals the two fluorescence emitters located at a separation of 140 nm, which is consistent with the result obtained by SIM (Fig. 2c).

A theoretical approximation of the Rayleigh limit for a GATTA-SIM 140B nanoruler (Fig. 2c of the main manuscript) with $\lambda_{em} = 525$ nm and NA = 1.4, is d = (0.61*525)/1.4 = 229 nm (Table 1). Figures 1c and 2b show that MSSR processing theoretically extends spatial resolution. Computation of resolution on $sf\text{-}MSSR^0$ using the Rayleigh criterion (at 0.74

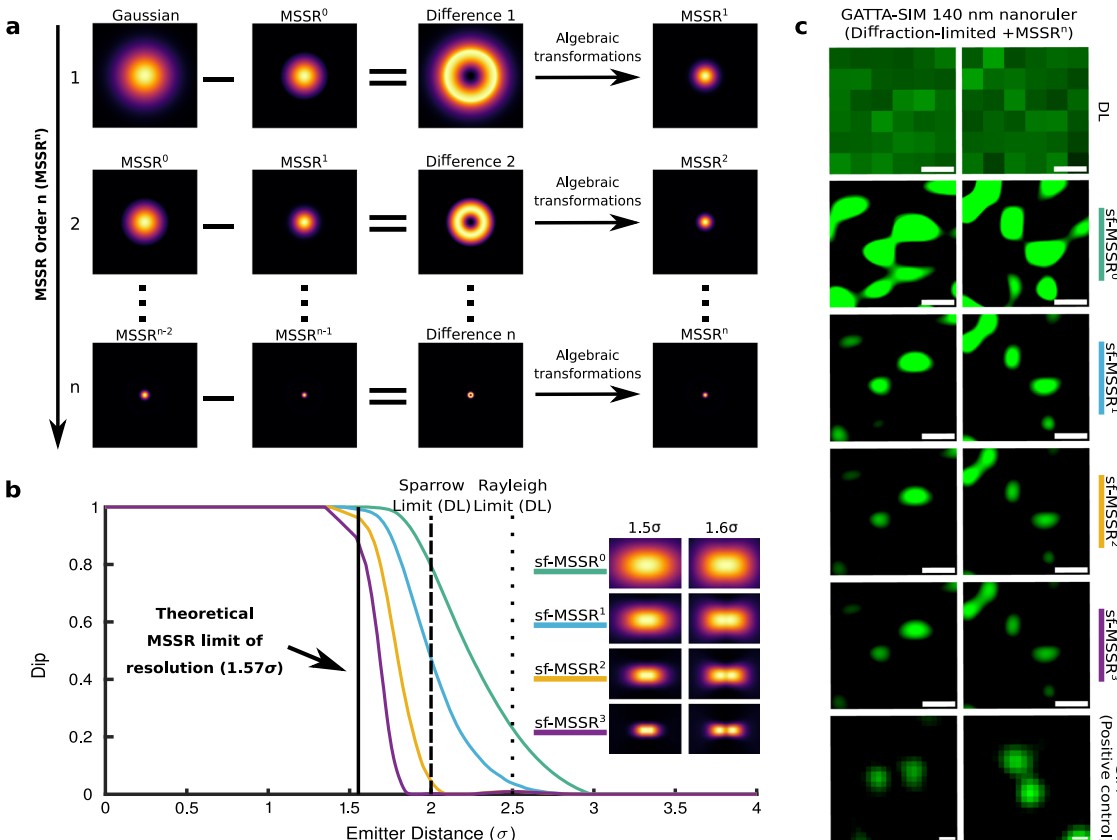

**Fig. 2 | Single-frame MSSR analysis of higher order attains a resolution limit of 1.6 σ for nearby emitters. a** Higher-order MSSR algorithm (MSSR$^n$). The first iteration of MSSR (MSSR$^1$) is given by subtracting the MSSR$^0$ from the original image, resulting in a doughnut-like region centered at the emitter's location. MSSR$^1$ is computed after applying further algebraic transformations (see Supplementary Note 5 and Supplementary Fig. S11 (ii–iv) for a full description). The second iteration encompasses the subtraction of MSSR$^1$ from MSSR$^0$ and the same algebraic transformations as used for generation of MSSR$^1$. The process is repeated by updating consecutive MSSR images which generates higher MSSR orders. **b** Theoretical limit of resolution achievable by MSSR$^n$. Dip computed for two Gaussian emitters in accordance with the variation of the inter-emitter distance (expressed as σ-times their standard deviation before MSSR processing). Colored lines represent the dip of MSSR order, from 0 to 3, computed at a given σ distance between emitters. Images on the right are the bidimensional representation of the MSSR$^n$ processing for two single emitters separated at distances of 1.5 σ and 1.6 σ. Note that, for 1.5 σ, emitters are unresolved up to the third order of MSSR (for detail see Online Methods section Simulation of fluorescent emitters). Dot and dashed lines indicate the Rayleigh and Sparrow limits for the DL case, and the continuous line marks the MSSR limit of resolution. **c** Experimental demonstration of the resolution increases attainable with higher-order MSSR using the GATTA-SIM 140B nanoruler system. The intensity distribution of the emitter shrinks, both in σ and intensity, as the order of the MSSR increases (Supplementary Fig. S12). Nearby emitters (Alexa Fluor® 488) located 140 nm apart are resolved using MSSR$^1$, MSSR$^2$ and MSSR$^3$ (right side). SIM images collected from the same sample (distinct fields) are shown as a positive control. sf-MSSR parameters: AMP = 10, FWHM of PSF = 3.48, order = 0–3. Scale bar: 100 nm.

of the Dip) gives a spatial resolution of 160 nm, which corresponds to a resolution change to 0.69 times (0.69×) the resolution limit. Using higher orders of sf-MSSR$^n$ with $n$ = 1, 2, 3, gives a resolution change of 0.66–0.64× the resolution limit (Table 1). Table 1 also shows the spatial resolution measured on the GATTA-SIM 140B nanorulers (experimental data), computed through decorrelation[34], using the Image Decorrelation plug-in for Fiji/ImageJ. Decorrelation computes the maximal observable frequency in an image ($K_0$) as a proxy of spatial resolution (Resolution = 1/$K_0$). Note that sf-MSSR$^n$ noticeably reduces resolution as a function of the $n$-order.

To further test the attainable resolution by sf-MSSR$^n$ we used the ArgoLight test slide, acquiring images of the pattern formed by gradually-increasing spaced lines of fluorescent molecules (Argo-SIM, pattern E). The distance between lines increases from 0 to 390 nm with a step change of 30 nm: 0 nm, 30 nm, 60 nm, etc. Figure 3 shows the application of sf-MSSR$^n$ to a confocal microscopy image of the Argo-SIM micropattern. As expected, the confocal acquisition allows to resolve parallel rows of fluorophores located at 240 nm. Remarkably, sf-MSSR$^0$ processing extended spatial resolution down to 0.5 times the confocal resolution, allowing to discriminate parallel rows of fluorophores

located at 120 nm. It is worth mentioning that this improvement comes at zero hardware cost, compared to other methods requiring specific optics/detectors such as the Airyscan[35] and Re-scan confocal microscopy[36]. Higher orders of sf-MSSR$^n$ create a saddle point between parallel rows of emitters located in the range of 60–90 nm at the boundaries of the Rayleigh limit (Fig. 3b, c).

## MSSR enhances the resolution of images with extended resolution

Based on the MSSR capabilities to generate a micrography with extended spatial resolution after processing a single fluorescence image, we explored if a pre-existing image with extended resolution can be further enhanced by sf-MSSR$^n$.

The Argo-SIM micropattern was imaged using an Airyscan detector with other experimental settings as in Fig. 3. Images were further deconvolved with the corresponding Airyscan algorithm[37]. Figure 4 shows the application of sf-MSSR$^n$ to the Airyscan microscopy images of the Argo-SIM micropattern. Within the Airyscan processed image it is possible to resolve parallel rows of fluorophores located at 180 nm, but not 120 nm or less. sf-MSSR$^0$ processing enhanced the

spatial resolution of the same Airyscan data, allowing to discriminate parallel rows of fluorophores located at 120 nm, or less (Fig. 4a, b). Higher orders of sf-MSSR$^n$ create a saddle point between parallel rows of emitters located in the range of 60–120 nm (Fig. 4b, c). Remarkably, compared with the confocal original data in Fig. 3, where the last resolvable line pair is the one corresponding to 240 nm, the value of 120 nm obtained in Fig. 4 by applying sf-MSSR$^0$ to Airyscan processed data corresponds to a 2-fold improvement in resolution. The first reported applications of Airyscan technology allowed an improvement in resolution of 1.7×[38], while only following protocols claim that a 2-fold improvement might be achieved[37], compared to standard confocal detection.

### Table 1 | sf-MSSR extends spatial resolution on simulated and real experimental conditions

| GATTA-SIM 140B | Resolution | diffraction limited | sf-MSSR$^0$ | sf-MSSR$^1$ | sf-MSSR$^2$ | sf-MSSR$^3$ |
|---|---|---|---|---|---|---|
| Simulation | Rayleigh | 229 nm (1×) | 160 nm (0.69×) | 152 nm (0.66×) | 148 nm (0.65×) | 146 nm (0.64×) |
| | limit | 2.90 σ | 2.02 σ | 1.92 σ | 1.87 σ | 1.841 σ |
| | PSF FWHM | 192 nm | 114 nm | 84 nm | 56 nm | 38 nm |
| | | 2.35 σ | 1.10 σ | 0.74 σ | 0.49 σ | 0.34 σ |
| | PSF sigma | 79 nm | 44 nm | 34 nm | 24 nm | 16 nm |
| | | 1 σ | 0.63 σ | 0.43 σ | 0.28 σ | 0.20 σ |
| Experiment | resolution | 260 nm (1×) | 58 nm (0.22×) | 33 nm (0.13×) | 21 nm (0.08×) | 13 nm (0.05×) |

For simulated conditions the Rayleigh limit and Sparrow limit were computed by the simulation of two fluorescent emitters. The values of FWHM of PSF (values in nm units) have been computed by measuring directly on the PSF distribution. The values of PSF sigma (values in nm units) were computed by fitting a Gaussian distribution and reporting the corresponding sigma parameter. sf-MSSR parameters: AMP = 1, FWHM of PSF = 198 nm, order = 0–3. ImageDecorrelation parameters for DL case: Rmin = 0, Rmax = 0.7, Nr = 50, Ng = 10. ImageDecorrelation parameters for sf-MSSR$^n$ (n = 0–3): Rmin = 0, Rmax = 0.3, Nr = 50, Ng = 10.

We then applied sf-MSSR$^0$ on a SIM image of sister meiotic chromatids of mouse chromosomes[39]. Similarly to the Airyscan images, Fig. 4d shows that sf-MSSR$^0$ processing of SIM images enhances both the contrast and resolution.

### MSSR enhances the resolution of super-resolved images

We explored the possibility for any previously super-resolved image to be further enhanced by sf-MSSR. First, we used a temporal stack of DL images of tubulin-labeled microtubules collected at high fluorophore density (previously used to test and compare a variety of SMLM and SRM algorithms)[40–43], which were subject to FF-SRM analysis[10,44]. ESI, SRRF or MUSICAL were used to compute a single SRM image (Fig. 5a)[11–13]. Supplementary Note 9 contains an in-depth comparison of sf-MSSR$^0$ reconstructions combined with either ESI, SRRF and MUSICAL, which achieve super-resolution through a temporal analysis[11–14]. Figure 5a shows that post-processing of ESI, SRRF or MUSICAL images with sf-MSSR$^0$ enhances contrast and spatial resolution (Fig. 5a).

Second, a sequence of images of randomly blinking emitters placed along a synthetic tubular structure[41] was processed with sf-MSSR$^0$ after analysis with MUSICAL. In both reconstructions, three regions (gray squares in Fig. 5b) were chosen to assess the gain in resolution, visualized in terms of the distance between the normalized intensity distributions peaks. MUSICAL further resolves the edges of the synthetic structures on the MUSICAL-processed image without changing the position of the distribution peaks (Fig. 4c) as predicted by our theory (Supplementary Notes 1 to 4).

Lastly, we set up an experimental assay to examine the achievable resolution by MSSR alone, or in combination with either confocal or stimulated emission depletion (STED) microscopy. Figure 6a, b shows that STED, but not confocal imaging allows to discern chromosomal territories within the chromatin of 2-cell stage embryo, as observed by the lack of colocalization of acetylated (ac, i.e., 'active') or methylated (me, i.e., 'inactive') chromatin states, H3K27me and H3K27ac, respectively[45]. Remarkably, Fig. 6c shows that sf-MSSR$^1$ processing of the confocal image allows to reach a similar experimental

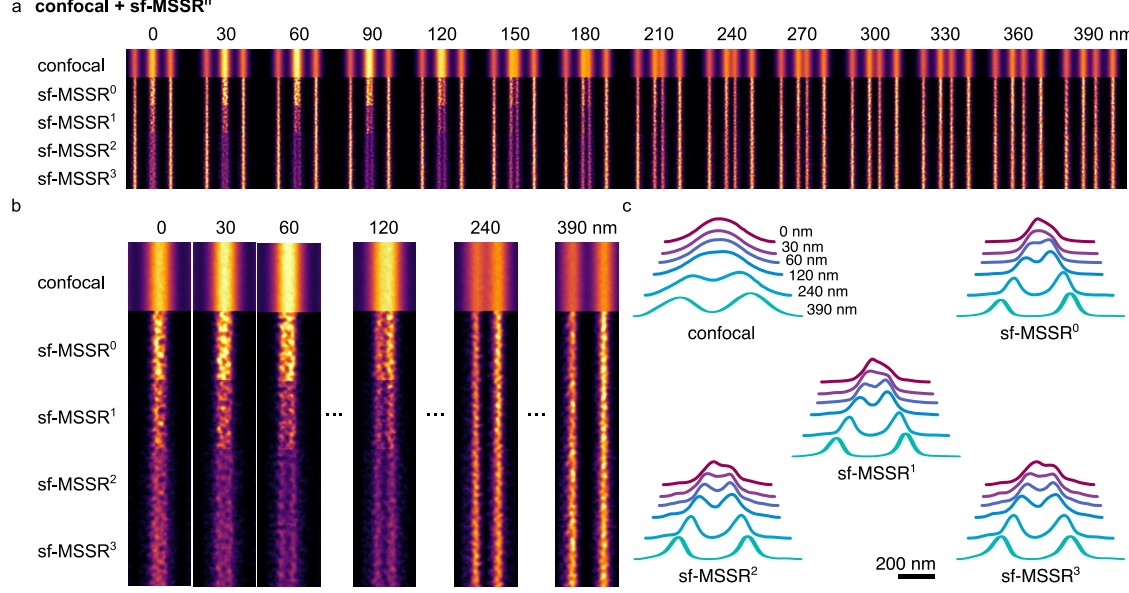

**Fig. 3 | sf-MSSR$^n$ extends spatial resolution in confocal microscopy.**
**a** Comparison of confocal and sf-MSSR$^n$ reconstruction (n = 0–3), applied to a spaced fluorescent line pattern. Central lines are gradually being separated by steps of 30 nm (0 nm, 30 nm, 60 nm, …, 390 nm). **b** Results of confocal and sf-MSSR$^n$ reconstruction (n = 0–3) of line patterns separated at 0 nm, 30 nm, 60 nm, 120 nm, 240 nm and 390 nm. **c** Average profiles of images obtained in **b**. Images were acquired using a Plan-Apochromat 63x/1.4 Oil immersion objective, exciting the Argo-SIM micropattern E with a 405 nm laser and detecting the fluorescence in the range 420–480 nm (Zeiss LSM880). The pixel size was 44 nm. Images in (**a**) a (**b**) correspond to the ensemble average of 31 consecutive sections (width = 1 μm) along the Argo-SIM micropattern E.

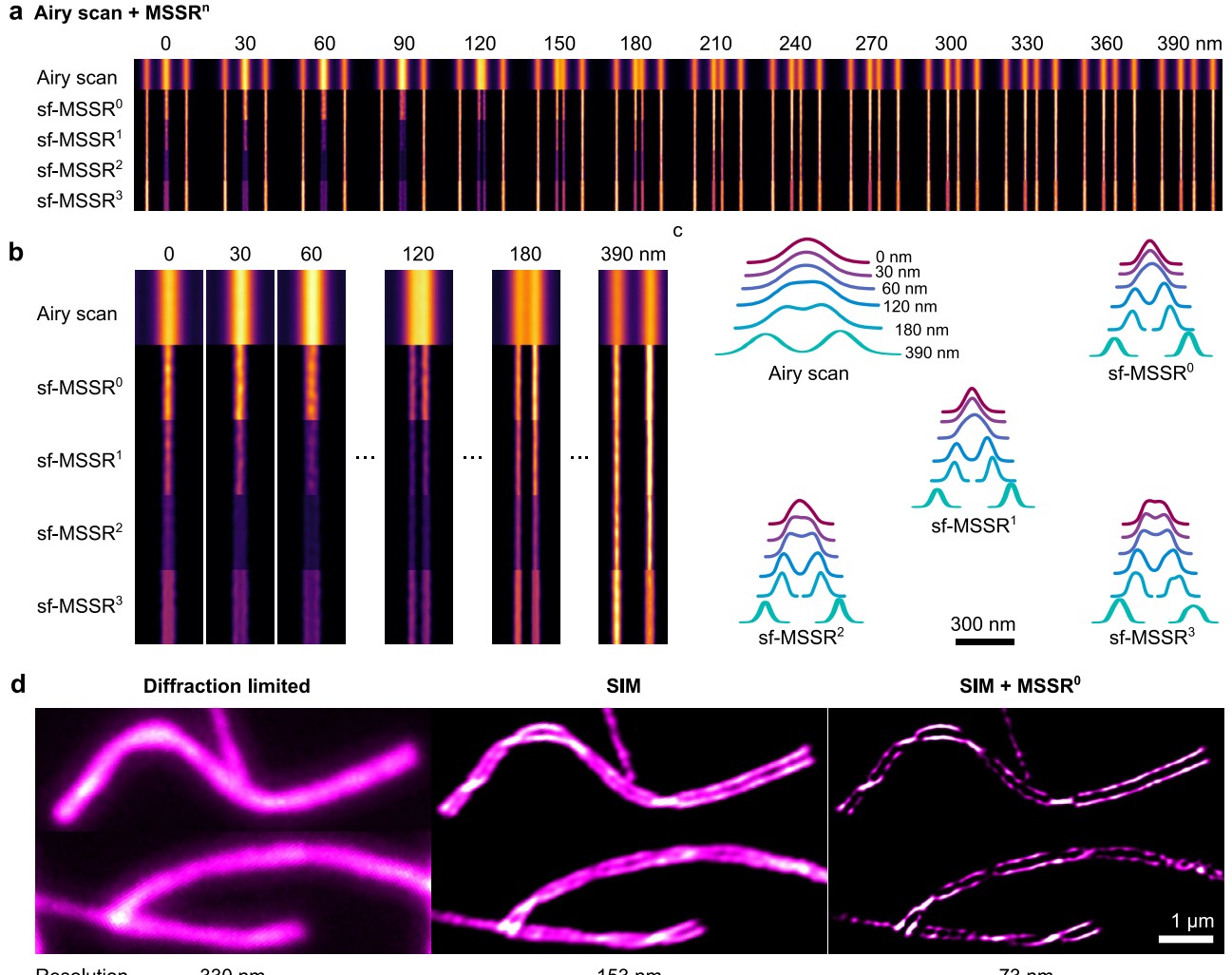

**Fig. 4 | sf-MSSR$^n$ enhances the resolution and contrast of Airyscan and SIM reconstructions.** **a** Comparison of confocal and sf-MSSR$^n$ reconstruction ($n = 0$–3), applied to a spaced fluorescent line pattern. Central lines are gradually being separated by steps of 30 nm (0 nm, 30 nm, 60 nm, ... 390 nm). **b** Results of confocal and sf-MSSR$^n$ reconstruction ($n = 0$–3) of line patterns separated at 0 nm, 30 nm, 60 nm, 120 nm, 240 nm and 390 nm. **c** Average profiles of images obtained in (**b**). **d** sf-MSSR$^0$ applied to SIM reconstruction of chromosome axis. **d** Synapsed homologs meiotic chromosomes of mouse, visualized by TIRFM (left), SIM (middle) and SIM + MSSR$^0$ (right). Images in (**a**) a (**b**) correspond to the ensemble average of 31 consecutive sections (width = 1 μm) along the Argo-SIM micropattern E. Resolution in **d** was measured with the image decorrelation plugin of FIJI/image[34].

conclusion, i.e., no colocalization exists between H3K27me and H3K27ac signals.

Figure 6 also shows the result of post-processing of the STED image with sf-MSSR$^1$ (Fig. 6g). We note an increase in contrast and resolution from confocal to STED and then to STED + sf-MSSR$^1$ (compare panels e–g in Fig. 6). To assess the resolution change provided by either STED or STED + sf-MSSR$^1$ we used the image decorrelation approach[34]. The bottom panels of Fig. 6e–g, show the resolution computed at a confocal image (315–330 nm), STED (80–100 nm) and STED + sf-MSSR$^1$ (19–22 nm). Based on these results, it is concluded that sf-MSSR$^1$ processing succesfully increases the spatial resolution of STED images.

## MSSR achieves super-resolution by analyzing fluorescence intermittency over time

Analyzing the temporal dynamics of fluorescence is central to achieving the highest attainable gain of resolution by FF-SRM approaches (reviewed in[44,46]). FF-SRM methods analyze higher-order temporal statistics looking to discern correlated temporal information (due to fluorescent fluctuations) from uncorrelated noise (i.e., noise detector). In theory, MSSR can be applied to a sequence of images (Supplementary Note 5). Based on the increase in resolution offered by FF-SRM approaches (SRRF, MUSICAL), we investigated whether a further resolution gain could be achieved by applying a temporal analysis to a sequence of single-frame MSSR images (denoted by t-MSSR$^n$) (Fig. 7a). Pixel-wise temporal functions (PTF), such as average (Mean), variance (Var), the temporal product mean (TPM), coefficient of variation (CV) or auto-cumulant function of orders 2 to 4 (SOFI$_2$, SOFI$_3$, SOFI$_4$)[10], can be used to create an image with enhanced spatial resolution (Supplementary Note 5.2, Supplementary Table S2).

To experimentally validate the increase in resolution by both sf-MSSR$^n$ and t-MSSR$^n$ we used two different nanoruler systems, an in-lab CRISPR/dCas12a nanoruler, used to score nanoscopic distances between individual fluorescent sites down to 100 nm, and a commercial nanoruler with fluorophores positioned at 40 nm of separation (GATTA-PAINT, 40 G, and 40RY. Gattaquant).

The CRISPR/dCas12a nanoruler system consists of a dsDNA with four binding sites for dCas12a uniformly distributed every 297 bp (equivalent to ~100 nm of separation) (Supplementary Fig. S42a). To validate this system, we imaged the association of the CRISPR/dCas12a complex to the binding sites on the dsDNA by atomic force microscopy (AFM) and measured the distance between each dCas12a complex (Supplementary Fig. S42b).

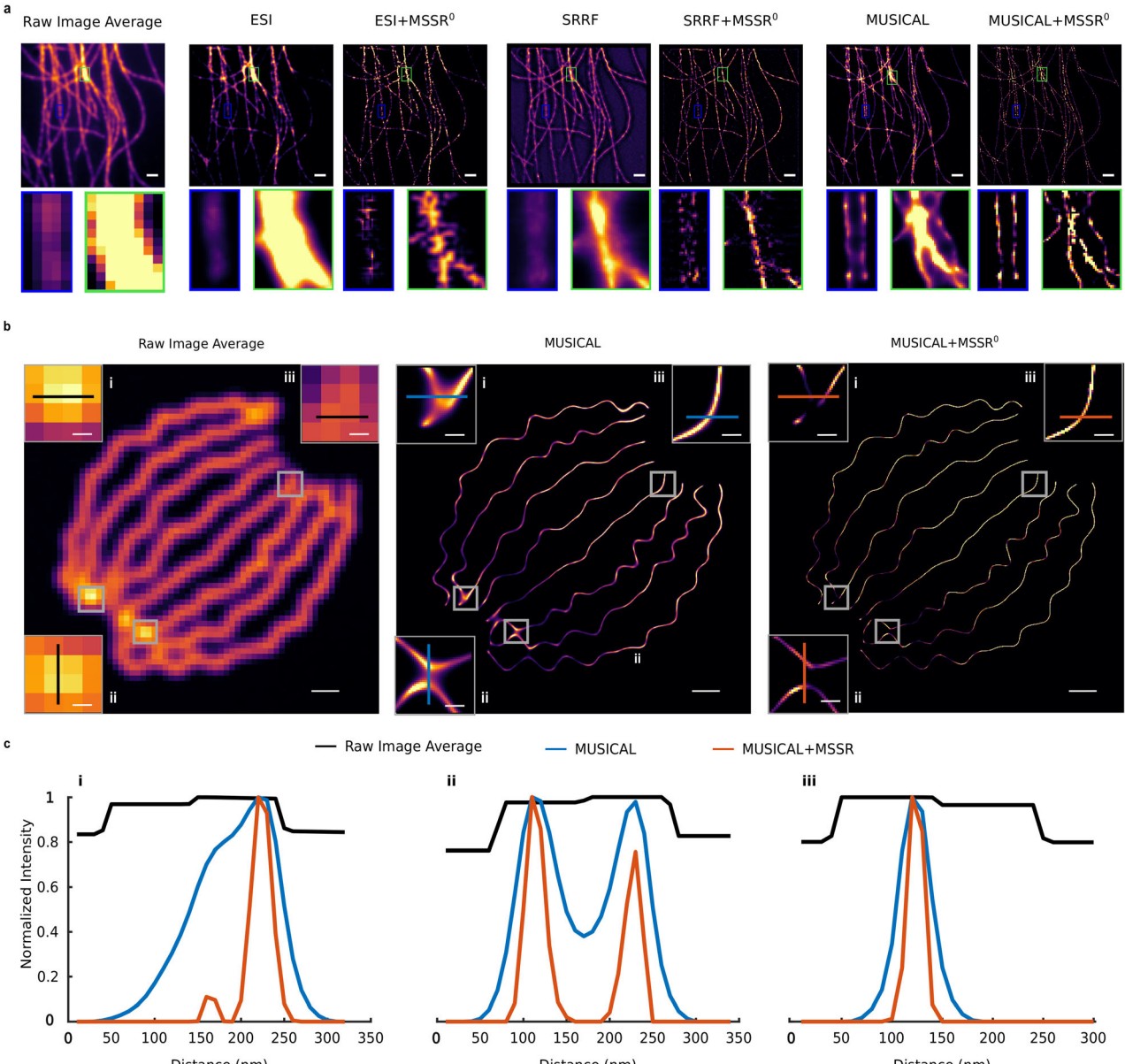

**Fig. 5 | MSSR further enhances the resolution and contrast of SIM or previously super-resolved images. a** Comparison of SRM results of ESI, SRRF and MUSICAL alone and after post-processing with MSSR⁰ (ESI + sf-MSSR⁰, SRRF + sf-MSSR⁰, MUSICAL + sf-MSSR⁰), over a temporal stack of 500 DL images of tubulin-labeled microtubules. The average projection of the DL stack is shown on the leftmost side. **b** Comparison of the increase in spatial resolution of MUSICAL with and without post-processing with MSSR⁰ (MUSICAL + sf-MSSR⁰), over a temporal stack of 361 DL images of modeled fluorophores bounded to a synthetic array of nanotubes (average projection shown on left). The graphs show the intensity profiles along

the lines depicted in each of the insets in the images of the upper row; black, blue and red lines correspond to the average DL, MUSICAL and MUSICAL + sf-MSSR⁰ images, respectively. Scale bars: **a** 200 nm; **b** 1 μm, insets = 200 nm; **c** 500 nm, insets = 100 nm. MSSR parameters: AMP = 1, FWHM of PSF = 2, order = 0; ESI parameters: two iterations, first iteration: image output = 250, bin = 50, order = 2, second iteration: image output = 50, bins = 5, order = 2; SRRF parameters: default parameters; MUSICAL parameters: $\lambda_{em}$= 650 nm, NA = 1.4, pixel size = 100 nm, threshold = −0.6, α = 4, subpixel per pixel = 4. The datasets used to build this figure are available at https://srm.epfl.ch/Datasets.

The CRISPR-dCas12a nanorulers were then imaged by TIRF microscopy for further MSSR analysis. We used a DNA-PAINT approach for fluorescence indirect tagging[47], in which a fluorescent ssDNA probe hybridizes with an extension of the gRNA. The "blinking" of the fluorescence signal is attained by events of association and dissociation between the fluorescent probe and the gRNA on the CRISPR/dCas12a nanoruler at the binding site.

In the DL image, amorphous spot-like fluorescent patterns were observed (Fig. 7b). sf-MSSR³ processing of either an isolated frame or an average projection of the corresponding stack of 100 images (DL-AVG) could not resolve individual CRISPR/dCas12a binding sites (Fig. 7b), and only after processing by t-MSSR³ did individual binding

sites become resolved (Fig. 7c). The result of t-MSSR³ varied in relation to the temporal function used (Fig. 7c). The best result for this nanoruler was obtained by the pixel-wise temporal variance (Var) of the sf-MSSR³ stack (Fig. 7c). t-MSSR³-Var resolved nearby emitters engineered to recognize binding sites located at 100 nm (Supplementary Movie S1), provided by scoring association-dissociation events between the imaging probe and the gRNA.

Once the CRISPR/dCas12a nanorulers were super-resolved by t-MSSR³-Var (Fig. 7c), we scanned all the individual emitters in close proximity, to determine the average distances between different dCas12a associated to the dsDNA binding sites (theoretically inter-spaced by 100 nm, Supplementary Fig. S42). Since DNA in solution is a

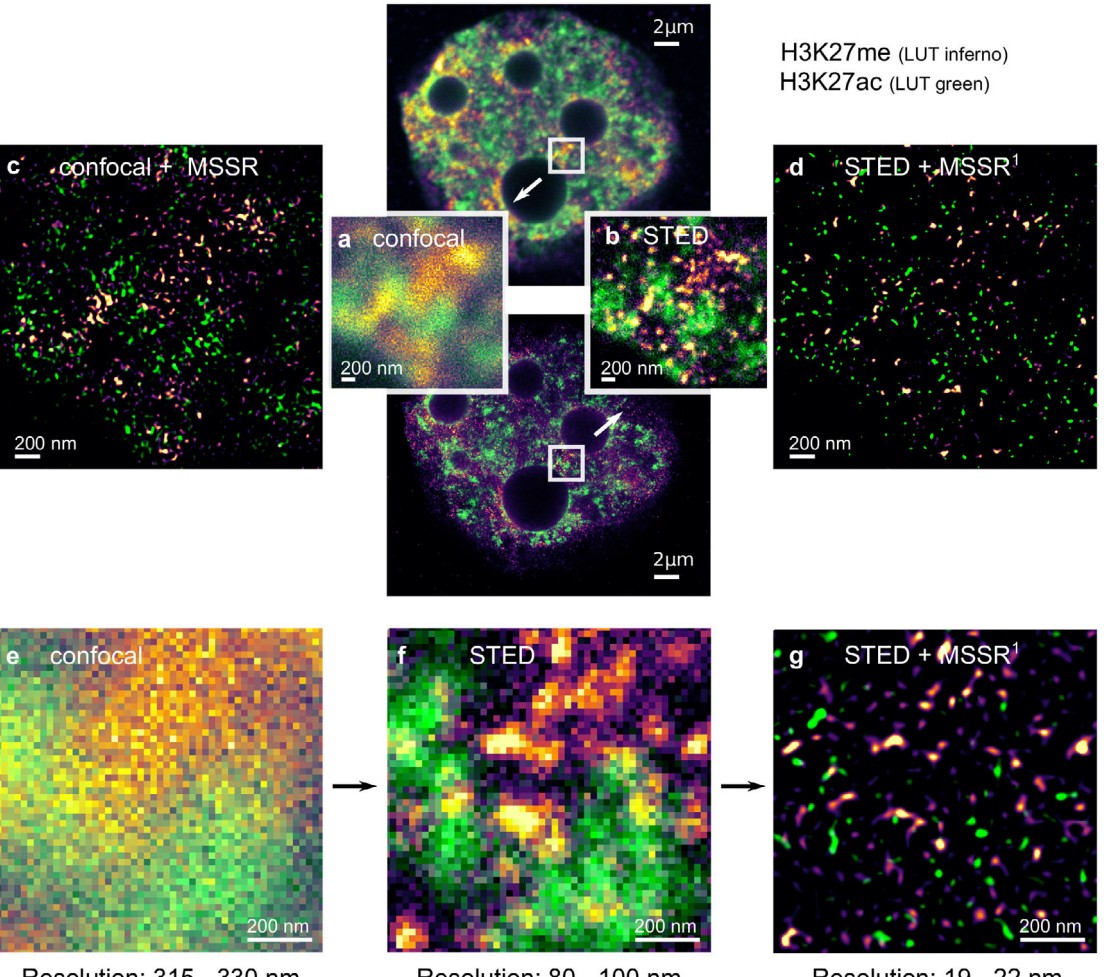

**Fig. 6 | sf-MSSR[n] enhances spatial resolution in STED microscopy images.**
**a** Confocal and **b** STED micrographs of fluorescent histone protein H3K27 in its acetylated (H3K27ac, i.e., 'active', LUT = inferno) or methylated (H3K27me, i.e., 'inactive', LUT = green) states, imaged in a 2-cell stage mice embryo (top-center). **c, d** show the corresponding resolution-enhanced scenes provided by post-processing of the confocal and STED images with sf-MSSR[1], respectively. The bottom row shows the sequential increase in resolution when comparing the same sample imaged with a (**e**) confocal microscope, an (**f**) STED microscope and (**g**) after processing the STED image with sf-MSSR[1]. For each image, the resolution range (in nanometers) was assessed using the ImDecorr algorithm in ImageJ[34]. Immunofluorescence imaging was carried out using a STEDYCON mounted on an upright Zeiss microscope in confocal or STED modes. Samples were imaged with a 20 nm pixel size. Primary antibodies: anti-H3K27me (Abcam, ab6002) and anti-H3K27ac (Active Motif, 39034). Secondary antibodies: anti-mouse labeled with STAR Red and anti-rabbit labeled with STAR Orange. The parameters used for sf-MSSR processing were AMP = 5, order = 1, FWHM of PSF = 4, interpolation = bicubic, mesh-minimization = true.

semi-flexible polymer[48], the measured distances resulted in three different distributions: $91 \pm 31$ nm, $220 \pm 52$ nm, $323 \pm 19$ nm (Fig. 7d), which accounts for dCas12a separated by one, two and three binding sites along the dsDNA, respectively. These results confirm that t-MSSR[3] can successfully resolve nanoscopic distances.

To explore the limit of the resolution attainable by t-MSSR[n], we looked at a nanoruler system with smaller separation between fluorophore sites (from Gattaquant) (Supplementary Fig. S43a). Figure 7e shows that using the TRA or TRM pixel-wise temporal functions in combination with MSSR[n] (t-MSSR[n]) does not provide extra resolution enhancement. TRA provides t-MSSR[3] reconstructions less influenced by noise (recommended in imaging scenarios of marginal signal-to-noise ratios) and TRM delivers reconstructions whose intensity distribution is less dominated by constantly emitting sources, i.e., fiducial markers. In stark contrast, higher spatial resolution regimes, -0.15 times the FWHM of the PSF (PSF σ: [0.4−0.5]), are achieved by t-MSSR[n] when encompassing pixel-wise temporal functions of higher-order statistics such as Variance (Var), Coefficient of Variation (CV),

or SOFI (TRAC2−4). Analysis with t-MSSR[3] of 300 DL images using either Var or CV revealed individual fluorescent spots at 40 nm apart (Fig. 7e and Supplementary Fig. S43b). The data presented in Fig. 7e demonstrate that t-MSSR[3] resolves nanoscopic distances at 40 nm, validating a lower experimental spatial resolution bound of 0.5 σ (≈40 nm), which depends on the emission wavelength of the fluorophore (Fig. 7e, Supplementary Fig. S10c). In comparison, SRRF and MUSICAL were not able to resolve fluorescent emitters located 40 nm apart, consistent with their limit within the range of 50−70 nm (Fig. 7f)[11–13].

Analyzing the temporal dynamics of fluorescence intermittency is central for any FF-SRM approach[10–13]. The highest attainable spatial resolution is influenced by blinking statistics, i.e., photokinetics for fixed fluorophores[11–13,49], and binding energies for diffusible fluorescent probes[50]. These factors, including the density of fluorophores emitting per frame, impinge on the final resolution of the reconstruction. Supplementary Fig. S44 shows the pixel-wise temporal dynamics of two GATTA paint nanorulers (ATTO 655), where the temporal

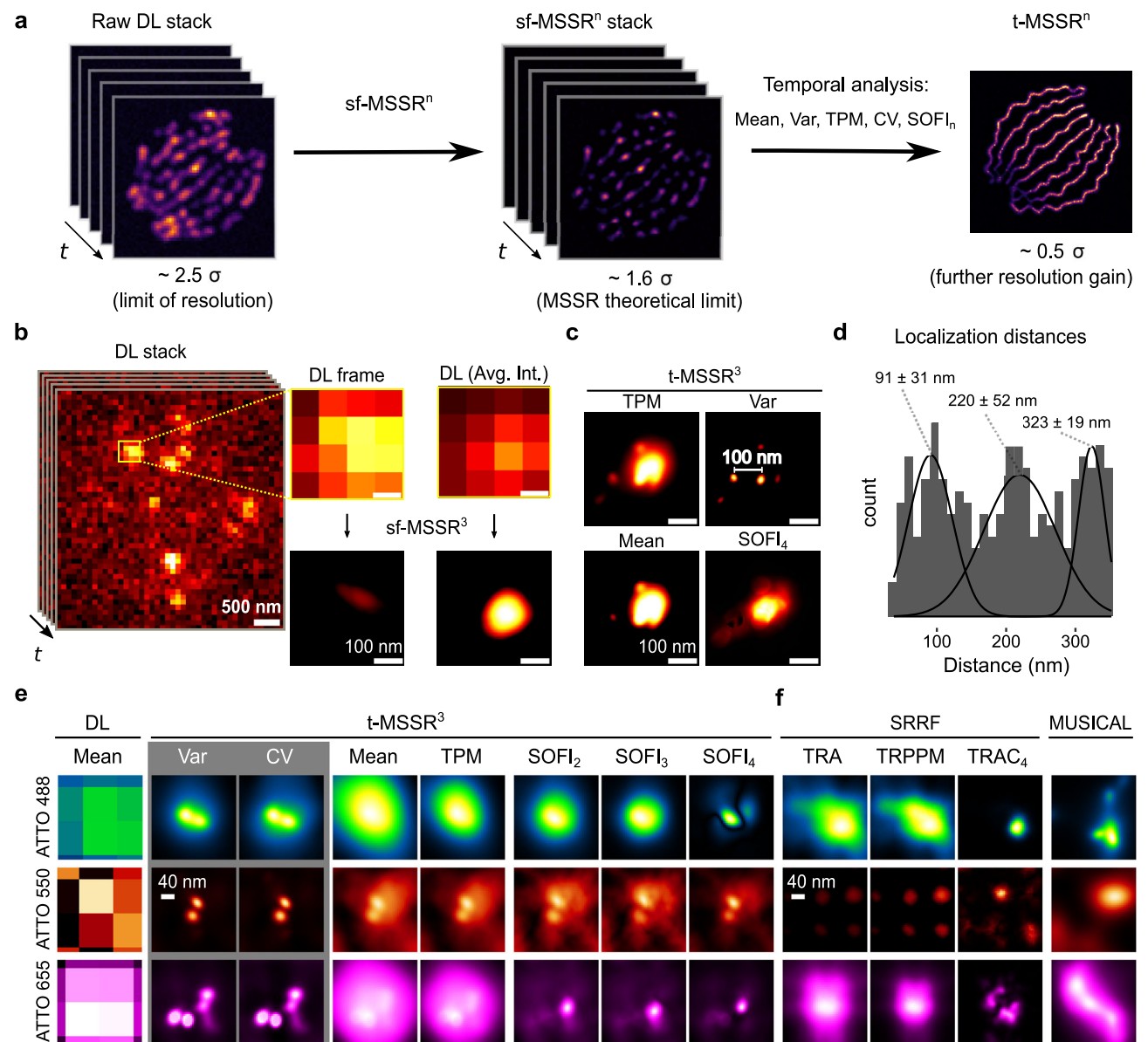

**Fig. 7 | The temporal analysis of MSSR provides a further increase in resolution to 40 nm. a** Single-frame analysis of MSSR of a given order $n$ is applied to each frame of a sequence, becoming the sf-MSSR$^n$ stack. Next, a pixel-wise temporal function (PTF) converts the MSSR stack into a single super-resolved t-MSSR$^n$ image. **b** Left: a stack of DL images of a CRISPR/dCas12a nanoruler system. Right: zoomed region of the first frame in the stack, along with the average projection (DL-AVG) of a stack of 100 images, before and after MSSR processing. **c** PTF applied to a stack of MSSR$^3$ images (t-MSSR$^3$). Fluorescent emitters are separated by 100 nm, as established by the CRISPR/dCas12a nanoruler system. Four types of PTF were computed: TPM, Var, Mean and SOFI$_4$. MSSR parameters: AMP = 20, FWHM of PSF = 3.74, order = 3, number of images for PTF = 300. pixel size of the DL dataset = 100 nm. **d** Euclidean distances between nearby emitters automatically computed from t-MSSR$^3$-Var images, following a worm-like chain model (16 regions of interest used,

1.5 $\mu m^2$ each). **e** t-MSSR$^3$ analysis (see Supplementary Table S3) for a commercially available GATTA-PAINT nanoruler system. The Var column shows inter-emitter distances resolved at 40 nm. ATTO 488 (green), ATTO 550 (orange) and Atto 655 (magenta) fluorescent probes were used. **f** Same nanorulers shown in (**e**) but analyzed using SRRF and MUSICAL. MSSR parameters: AMP = 20, FWHM of PSF for Gattpaint ATTO 488 = 2.79, Gattapaint ATTO 550=3.31, Gattapaint ATTO 655 = 3.74, order = 3, number of images for PTF = 300. Nano J - SRRF parameters: bicubic interpolation magnification = 2, ring radius = 1, radiality magnification = 10, axes in ring = 8, PTF = TRA, TRPPM, TRCA$_4$, other parameters were set as default. MUSICAL parameters: emission wavelength = same as ATTO emission wavelength (520, 575, 680) used for each row in (**e**) and (**f**), NA = 1.49, pixel size = 100 nm, threshold = −0.3 (ATTO 488), −0.02 (ATTO 550), −0.2 (ATTO655), α = 4, subpixel per pixel = 20. frames = 300, musicJ v0.94 of imageJ.

dynamics of a given nanoruler containing three binding sites can be studied by analyzing the temporal fluorescence fluctuations at nearby pixels. Note that diffraction imposes constraints to distinguish the temporal dynamics at individual PAINT binding sites. Noteworthy, sf-MSSR$^3$ processing allows to untangle, in space and in time, the fluorescence dynamics at individual PAINT binding sites, where the fluorescence signal peaks due to the transient binding of a fluorescent labeled DNA strand to its corresponding binding site within the nanoruler.

## Single frame nanoscopy, free of noise-dependent artifacts
The theory of image processing by MSSR (Supplementary Note 5) suggests that it should be robust over a wide range of SNR, granted by four factors. First, when processing a single frame, MS works as a local spatial frequency filter (a smoothing filter); regions corresponding to the image background are homogenized by the kernel window, reducing variation in background noise. Second, one of the steps of the MSSR procedure is to remove the MS negative constraints. The goal of this step is to remove an artifact caused by the MS calculation itself

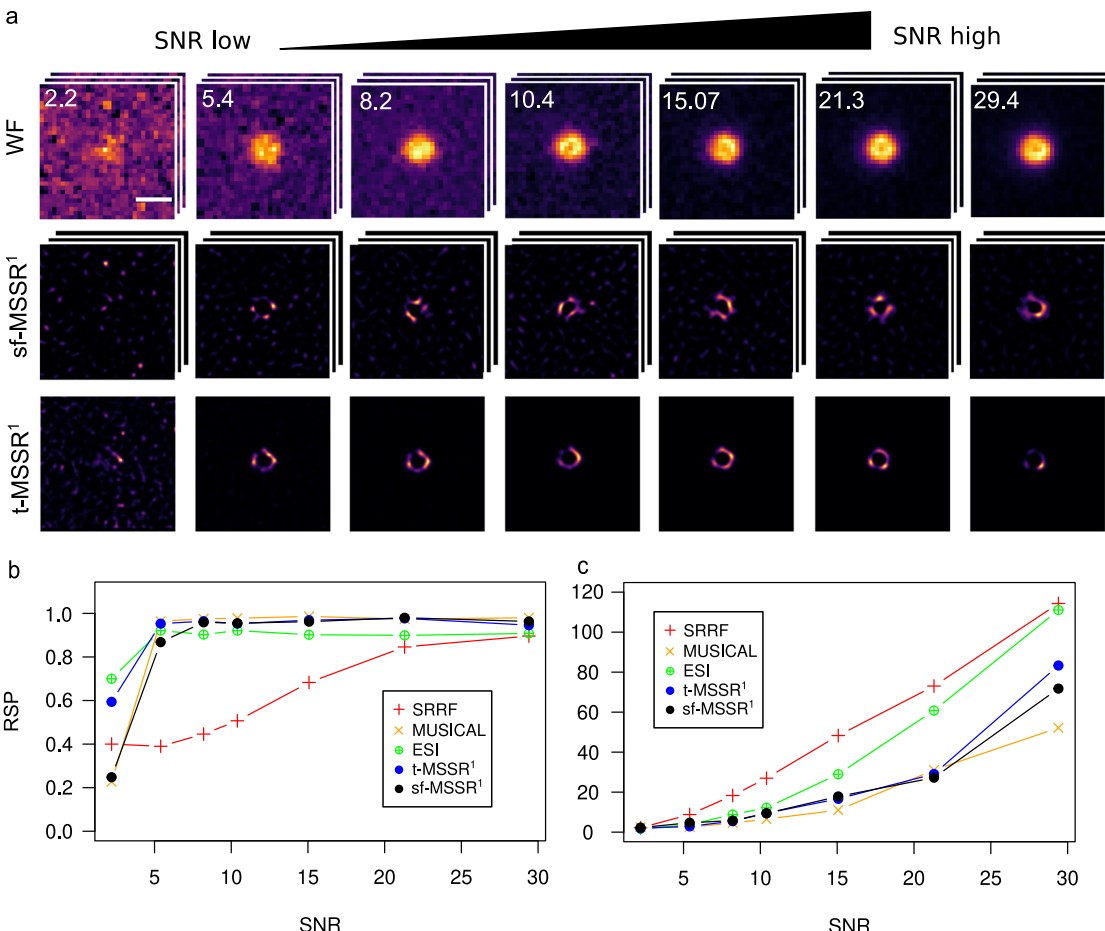

**Fig. 8 | MSSR is robust to image noise and shows high global performance when compared to other SRM analytical procedures. a** sf-MSSR[1] and t-MSSR[1] of 100 images provide consistent reconstructions across a wide range of SNR. The expected feature is a uniform fluorescent ring located at the center of the image with a dark background lacking fluorescence. Each image is displayed to show its full intensity range. The row for DL images (Widefield, WF) exemplifies a stack of 100 frames collected at the corresponding SNR. The central row represents an extended-resolved stack using sf-MSSR[1], AMP = 4, FWHM of PSF = 2.77, order = 1. The third row shows the super-resolved micrography after t-MSSR[1] analysis of 100

DL images using TPM for temporal analysis (see table S2). **b** Resolution Scaled Pearson (RSP) coefficient and (**c**) Resolution Scaled Error (RSE), computed for the super-resolution reconstructions provided by SRRF, MUSICAL, ESI, sf-MSSR[1] and t-MSSR[1] (100 frames). RSP measures a global correlation between reconstruction and reference (input DL image), values closer to 1 indicate a reliable reconstruction. RSE measures the absolute difference of the reconstructed image and its reference. Lower values of RSE at a particular SNR mean reduced global error in the reconstruction Scale bar: 1 μm.

when applied to Gaussian and Bessel distributions. When calculating the complement of the resulting distribution, a valley (or depression) is generated between the peak intensity and the tails, which lies in the negative values and is referred to as negative constraints. After this, the artifact is removed. Third, when using a PTF, nanoscopic information is enriched due to temporal oversampling of the hidden fluorescent structure. Fourth, the spatial kernel of the MSSR algorithm operates within the subpixel realm; the number of neighboring pixels is digitally increased through interpolation (i.e., bicubic interpolation[51] for single frame analysis and Fourier interpolation[52] for temporal analysis, Supplementary Note 6) providing digital oversampling of the emitters' locations (Supplementary Note 6).

We then experimentally assessed the capacity of MSSR to denoise fluorescence images and determine whether it introduces noise-related artifacts. We used a PSFcheck slide[53], which contains an array of regular fluorescent nanoscopic patterns shaped by laser lithography (Fig. 8). Analysis with sf-MSSR[n] or t-MSSR[n] showed, in comparison to alternative approaches, striking denoising capabilities without introducing noticeable artifacts (Fig. 8a, Supplementary Note 9). These artifacts, resembling amorphous nanoscopic structures around the fluorescent ring or within it, were commonly found at reconstructions generated by other analytical FF-SRM techniques (Supplementary Fig. S29).

Starting at a SNR > 2, sf-MSSR[1] provides reliable SRM reconstructions of comparable quality to other SRM approaches, which demand the temporal analysis of the fluorescence dynamics (Fig. 8a and Supplementary Note 9). We quantified the quality of the reconstructions by calculating the Resolution Scaled Pearson (RSP) coefficient and the Resolution Scaled Error (RSE), which provide a global measurement of the quality of the reconstruction by comparing the super-resolution image and the reference image (in this case, the DL image)[14]. Higher RSP and lower SRE values are associated with reliable reconstructions (Supplementary Note 8). When the SNR is above 5, all tested algorithms perform similarly well in quality (Fig. 8b), but their global errors differ from each other (Fig. 8c). As expected, the RSE increased as a function of the SNR of the input images for any tested algorithm (Fig. 8c).

The performance of MSSR in achieving a satisfactory reconstruction was assessed by varying the number of input images using a temporal analysis scheme (Supplementary Note 8). With SNR > 2 input data, RSP reaches near maxima values and RSE near minima values when processing a single frame (Supplementary Fig. S28, Supplementary Movie S2). However, when computing MSSR using low SNR input data (SNR ~ 2) a temporal analysis is required as RSP and RSE values reach a plateau only when a temporal stack of as few as 25

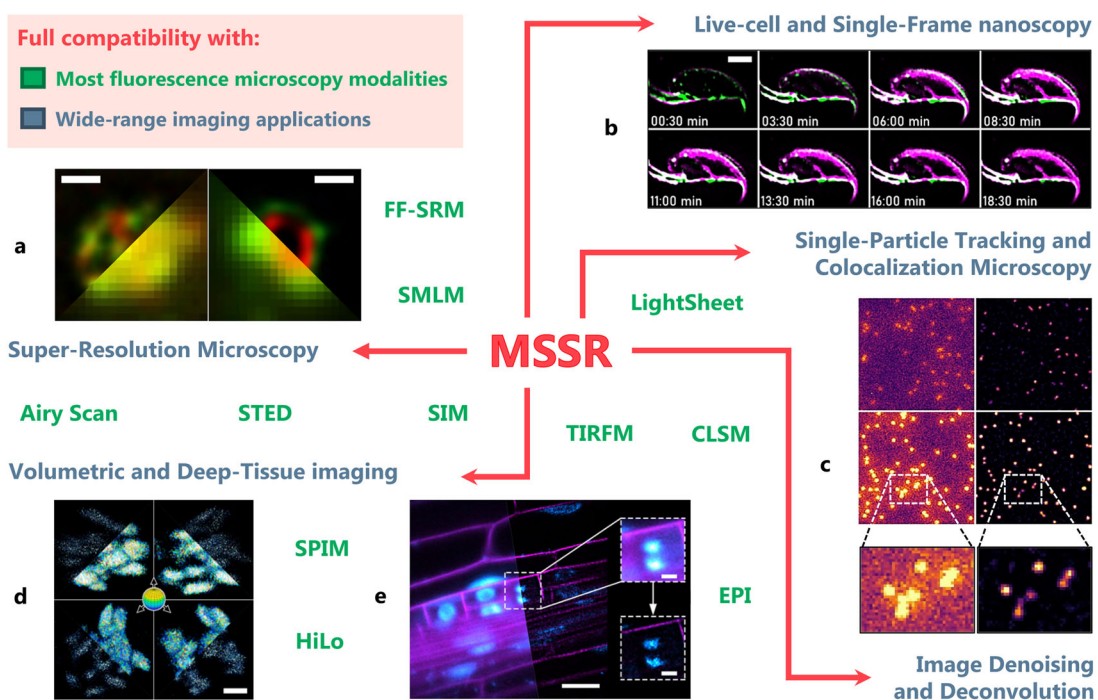

**Fig. 9 | MSSR applications in fluorescence microscopy.** MSSR operates over images acquired with most fluorescence microscopy modalities available (e.g., widefield, confocal, light-sheet, etc.), denoted by the text in green. It can be applied to achieve enhanced image resolution, SNR and contrast in live-cell imaging, single-particle tracking, deep tissue and volumetric imaging, among others (denoted by the text in pale blue). Some examples of these applications are detailed in the main text and Supplementary Note 10. *FF-SRM* Fluorescence Fluctuation-based Super-Resolution Microscopy, *SMLM* Single-Molecule Localization Microscopy, *STED* Stimulated Emission Depletion microscopy, *SIM* Structured Illumination Microscopy, *SPIM* Selective Plane Illumination Microscopy, *EPI* Epifluorescence, *HiLo* Highly inclined and Laminated optical microscopy, *TIRFM* Total Internal Reflection Fluorescence Microscopy, *CLSM* Confocal Laser-Scanning Microscopy. Scale bars: **a** 500 nm; **b** 2 μm; **c** no scale provided; **d** 2 μm; **e** 20 μm, insets = 5 μm.

images is used (Supplementary Movie S3). These findings illustrate that the minimal number of frames needed by MSSR to provide a reliable reconstruction depends on the information itself, i.e., on the SNR and on the fluorescence blinking statistics of the specimen (Supplementary Movies S1–S3); and can be determined by computing RSP and RSE as function of the number of analyzed frames with t-MSSR$^n$ (Supplementary Fig. S28).

**Nanoscopic resolution with conventional fluorescence imaging**
MSSR addresses the problem of spatial resolution in optical fluorescence microscopy from the statistical point of view, where the process of photon emission from punctual sources is considered as a discrete distribution of information, provided by detected photons. Enhanced or extended resolution is achieved by finding local modes of information without taking any prior about the local shape of the distribution. The latter makes the MSSR principle compatible with any imaging approach where the distribution of information is of discrete nature (i.e., photons), and emanates from discrete sources (i.e., fluorophores).

To showcase the versatility of MSSR to extend-, enhance- and super-resolve data acquired from different fluorescence applications, we evaluated its performance over a collection of experimental scenarios (Fig. 9) (Supplementary Note 10).

Analysis with MSSR provided nanoscopic resolution of rotavirus replication machineries (Fig. 9a, Supplementary Fig. S32), which were described by Garcés et al. as a layered array of viral protein distributions[54]. Originally, it took the authors several days to weeks to generate a single extended-resolution image by means of analyzing several stacks of hundreds of DL images using 3B-ODE FF-SRM. With MSSR, we were able to achieve comparable results, through analyzing single DL frames within seconds with a regular desktop computer with either sf-MSSR[1] or t-MSSR[1] (Supplementary Note 7).

Mouse sperm cells are used to study the acrosomal exocytosis (AE), a unique secretory process which results from fusion events between the plasma membrane and a specialized vesicle called acrosome located in the sperm head[55,56]. Nanoscopic remodeling of both plasma membrane and actin cytoskeleton was imaged during the AE by means of sf-MSSR[1], showing single frame temporal resolution (of milliseconds) (Fig. 9b, Supplementary Figs. S33–34). At the onset of the AE, the FM4-64 fluorescence (a probe that fluoresces when bound to membranes) was confined to the plasma membrane and was visible above a F-actin cytoskeleton fringe. During the AE, several fenestration events were observed to occur at both the plasma and acrosome membranes, as consequence of that, a notorious increase of FM4-64 was observed close-bellow the F-actin fringe (Supplementary Movie S5a–f). The AE is a dynamic remodeling process that takes minutes to occur, sf-MSSR[1] allows the observation of events occurring at the millisecond scales, which are hindered when using other SRM multi-frame analytical approaches, such as SRRF or 3B[11,57], due to their mandatory need of a temporal analysis of the fluorescence dynamics to unveil nanoscopic detail (compare Supplementary Figs. S33 and S34).

Background noise is known to be an important issue in single-particle tracking (SPT) applications as it decreases the ability to faithfully localize particles and follow them through time[58,59]. Moreover, the spatial overlap of PSFs derived from individual particles makes it challenging for SPT algorithms to recognize them as separate entities. The denoising capabilities of sf-MSSR[1] enhanced both the contrast and spatial resolution of freely diffusing in-silico particles (Fig. 9c, Supplementary Fig. S35), previously used as benchmarks to test a variety of SPT algorithms[60]. Pre-processing of the images with sf-MSSR[1] improved the tracking performance of three commonly employed SPT tracking algorithms: (i) the LAP framework for Brownian motion as in[61,62], (ii) a linear motion tracker based on Kalman filter[63,64], and (iii) a

tracker based on Nearest neighbors[65–68] within a wide range of particle densities and SNR (Supplementary Fig. S36). Additional testing with sf-MSSR[n] showed an increase in contrast for moving comet-like particles related to microtubule growth dynamics in live LLC-PK1 cells (Supplementary Fig. S37 and Supplementary Movies S10 and S11), as well as higher nanoscopic colocalization accuracy in double imaging experiments in single-molecule DNA curtain assays (Supplementary Fig. S38)[69].

Plasmalemma- and nuclear-labeled transgenic *Arabidopsis thaliana* plants are routinely used to study cell fate and proliferation during root development in time-lapse confocal microscopy experiments in two and three dimensions[70,71]. When applied to lateral root primordium cells, located deep inside the parent root, sf-MSSR[1] demonstrated the capacity to achieve multidimensional nanoscopic resolution as it revealed isolated nanodomains resembling nucleosome clutches, previously reported in mammalian cells[72,73], within the nuclei of a lateral root primordium cells (Fig. 9d–e, Supplementary Fig. S39 and Supplementary Movie S12). Similar observations were performed upon epidermal root tissues visualized via selective plane illumination microscopy (SPIM) after examination of volumetric data with sf-MSSR[1] (Supplementary Fig. S40).

Supplementary Fig. S41 shows a comparison of a 3D reconstruction from a stack of epifluorescence images of bovine pulmonary artery endothelial (BPAE) cells. When a 2D image is processed by sf-MSSR[1], an improvement in spatial resolution and contrast is observed (Supplementary Fig. S41a, b). On the contrary, when a stack is used for a 3D representation (images taken at z-planes), the resolution obtained by sf-MSSR along the z-axis is not as refined as in the xy-plane (Supplementary Fig. S41c, d). Nonetheless, a noticeable increase in contrast is attained in the overall dataset (Supplementary Fig. S41, see Supplementary Movies S13 and S14). To further extend the resolution of DL images in 3D, it is necessary to extend the current implementations of the MSSR algorithm to account for 3D information for MSSR processing.

In combination, these studies provide evidence for the capabilities of MSSR to resolve biological detail at nanoscopic scales using either relatively simple or advanced fluorescence microscopy technologies.

## Discussion

From the historical point of view, since the seminal development of the MS theory[17,18] and until the present day, few statistical and imaging applications based on the theory of MS compute the MS vector itself[74]. This can be explained, in part, because previous applications of MS are based on finding modes in the features space but did not calculate the MS vector, which is the key component of the working principle of MSSR. In contrast, MSSR represents an application of MS theory which also operates in the second derivative space. By computing the MS vector and estimating (photon) densities among pixels, MSSR computes a probability function for the fluorophore estimates whose individual fluorescence distributions are narrowed in comparison with the PSF of the optical system. The exploration of the information stored on the second derivative space of the image can be also achieved by substituting the MS by similar functions that operate in such space, e.g., Laplacian, Hessian, Difference of Gaussians[75] which, in comparison with the MS, offer computational advantages as they can be expressed in the Fourier space and implemented using the FFT algorithm[75]. The information harbored in the second derivative space of the DL image is used by MSSR to compute an image with higher spatial frequencies than the corresponding DL image, hence, overcoming both the Rayleigh and Sparrow limits, and setting up an undescribed limit of spatial resolution which deserves further exploration and characterization.

The MS theory is not restricted by the number of dimensions of the information required to compute the kernel windows over which MSSR operates (Supplementary Notes 2 and 3). Given that, MSSR parameters are suitable to extend its application to assess data with higher dimensions. For example, in 2D images, the spatial parameter of MSSR, which encompasses the lateral resolution width of the PSF, is defined to be the same in the x and y dimensions of the image. In such a case, the shape of the kernel is circle- or square-like, depending on the application used. For three-dimensional (3D) microscopy imaging, the lateral (x-y plane) and axial (x-z and y-z planes) dimensions are affected in different ways by diffraction. The MSSR principle can be further extended for explicit volumetric imaging by means of using an asymmetric kernel which can be defined following the 3D lateral-axial aspect ratio of the PSF (Supplementary Note 11). In addition, the definition of the spatial kernel can be refined to also consider possible deformations of axial symmetry of the PSF due to optical aberrations introduced by the imaging system or by the sample itself. A similar reasoning aimed to extend the portfolio of applications of MSSR can be envisaged considering spatial-range parameters, the latter narrowing down the working intensity space where local calculations of MSSR take place.

We present a analytical approach capable of achieving multidimensional nanoscopy through single-frame analysis under low SNR conditions and with minimal noise-dependent artifacts. Limited only by the imaging speed of the optical system setup, MSSR increases resolution by analyzing either a single frame, or by applying MSSR to each individual image in a stack followed by the application of a pixel-wise temporal function. MSSR is a powerful stand-alone method for either single or multi-frame fluorescence nanoscopy approaches, or as a post-processing method which can be applied to other analytical multi-frame (restricted to camera-based systems) or hardware dependent SRM methods for further enhancement of resolution and contrast. We demonstrated MSSR compatibility with other SRM methods and showed that its usage enhanced resolution and overall image quality in all the cases tested.

FF-SRM analytical multi-frame approaches such as SRRF, ESI, MUSICAL and 3B demand a temporal analysis which limits their utility for multi-dimensional imaging of live samples[65]. The need to collect hundreds to thousands of images of the same pseudo-static scene, challenges the applicability of these methods in multidimensional imaging. The temporal multi-frame requirement imposes a tradeoff between the achievable temporal and spatial resolutions. MSSR removes these constraints while maintaining computational efficiency (Supplementary Note 7).

We present applications of the MSSR principle that revealed fast molecular dynamics through single-frame analysis of live-cell imaging data, with reduced processing times in comparison with similar SRM approaches (Supplementary Notes 7 and 10). Moreover, MSSR improves the tracking efficacy of SPT methods by means of reducing background noise and increasing both the contrast and SNR of noisy SPT movies, enhancing the ability to resolve and track the position of single emitters. MSSR pushes the limits of live-cell nanoscopy by its excellent single-frame performance. This flexibility extends its utility to most fluorescence microscopy and alternative SRM methods.

Achieving both high (or sufficient) temporal and spatial resolution within a broad range of fluorescence microscopy applications is a common goal in the bioimaging field. With recent advances in microscopy equipment and imaging protocols, the gap between the highest attainable resolution in the temporal and spatial dimensions within the same experiment, has narrowed. This has been a challenge especially because both parameters often involve mutually exclusive optical instrumentation and experimental strategies. In this regard, MSSR drastically reduces the amount of data needed to reconstruct a single super-resolved micrography.

No longer having to sacrifice either temporal or spatial resolution over the other, has led some scientists to propose alternative ways to analyze imaging data. Some approximations have been tailored to

study millisecond molecular dynamics and structural feature changes within the same experiment[76], e.g., by taking advantage of the simultaneous use of image correlation spectroscopy (ICS) and FF-SRM methods such as SRRF[11]. In these contexts, MSSR could improve the analysis in three ways: (a) it delivers reliable SRM images in low SNR scenarios, which are common in the experimental regimes of ICS due to the relatively fast frame rates of its applications, (b) MSSR introduces no noise-dependent artifacts which further refines the quality of the spatial analysis and (c) since no temporal binning is necessary for MSSR, there is no restriction in the level of temporal detail retrievable from the ICS analysis.

Sub-millisecond time-lapse microscopy imaging can now be achieved by sCMOS technologies, with applications for particle velocimetry[77], rheometry[78], and optical patch clamp[79]. We envisage future applications of MSSR in these areas by means of unveiling nanoscopic details hidden in single DL images. Moreover, MSSR can facilitate correlative nanoscopic imaging through crosstalk with other imaging techniques such as electron microscopy, i.e., CLEM: correlative light electron microscopy[80]; or atomic force microscopy, i.e., CLAFEM: Correlative light atomic force electron microscopy[81]. In addition, MSSR can be applied to nanoscopic volumetric imaging by using it together with expansion microscopy[82], oblique angle microscopy[83], SPIM and lattice light sheet microscopy[84], extending their reach to previous unattainable resolution regimes.

A recent study by Chen et al., suggests that deep learning based artificial intelligence (AI) can reconstruct a super-resolution image from a single DL image[85]. Such AI-based SRM approaches are promising, however, they are limited to the existence of a maximum likelihood image obtained with another SRM, such as STORM, that is required for neural network training and error minimization. Otherwise, the method is prompted to bias the final reconstruction toward the topological information used for training[85]. Our approach works completely independent of other SRM methods and provides evidence of the existence of a lower resolution limit bound which lies on the second derivative space of the DL image, information inaccessible when using deep learning approaches.

MSSR applications might impact far beyond the field of microscopy, as its principles can be applied to any lens-based system such as astronomy[86] and high-resolution satellite imagery[87].

## Methods

### Source data availability

All source data used or generated in this study has been made publicly available in the Zenodo OpenAIRE database, and are accessible through a unique DOI, here provided.

| Dataset title | Location | DOI |
|---|---|---|
| Gatta-SIM nanorulers. | Fig. 2c. | https://doi.org/10.5281/zenodo.6941792 |
| Airyscan and Confocal line pattern. | Figs. 3 and 4. | https://doi.org/10.5281/zenodo.6848342 |
| Synapsed homologs of meiotic mouse chromosomes visualized by TIRFM. | Fig. 4d. | https://doi.org/10.5281/zenodo.6865142 |
| STED immunofluorescence imaging of histone protein H3K27 in a 2-cell stage mice embryo. | Fig. 6. | https://doi.org/10.5281/zenodo.6865168 |
| CRISPR-PAINT nanorulers. | Fig. 7b–f, Supplementary Figs. S42–44, Supplementary Movie S1. | https://doi.org/10.5281/zenodo.6850637 |
| PSFcheck ring pattern at various SNR. | Fig. 8, Supplementary Figs. S21–23, S25, S27–31, Supplementary Movies S2–3. | https://doi.org/10.5281/zenodo.6955019 |
| Rotavirus viroplasms. | Fig. 9a, Supplementary Fig. S32. | https://doi.org/10.5281/zenodo.6850357 |
| Mouse sperm acrosome exocytosis. | Fig. 9b, Supplementary Figs. S33–34, Supplementary Movies S4–9. | https://doi.org/10.5281/zenodo.6850232 |
| Volumetric imaging of *Arabidopsis thaliana* root cells. | Fig. 9d–e, Supplementary Figs. S39–40, Supplementary Movie S12. | https://doi.org/10.5281/zenodo.6850745 |
| EM-CCD noise image sequence. | Supplementary Figs. S16–18. | https://doi.org/10.5281/zenodo.6955070 |
| Live-cell imaging of LLC-PK1 cells microtubule dynamics. | Supplementary Fig. S37, Supplementary Movies S10–11. | https://doi.org/10.5281/zenodo.6850280 |
| DNA curtain assay for dCas12a/CS10B colocalization. | Supplementary Fig. S38. | https://doi.org/10.5281/zenodo.6865120 |
| Volumetric imaging of fluorescently labeled BPAE cells. | Supplementary Fig. S41. | https://doi.org/10.5281/zenodo.6865066 |

### Reagents

All chemicals were purchased from Sigma-Aldrich Chemical Co. (St Louis, MO) except otherwise indicated. SiR-actin was obtained from Cytoskeleton (Denver, CO) and FM4-64 was purchased from Thermo Fisher Scientific (Waltham, MA).

**AFM reagents.** The *Acidaminococcus sp* dCas12a protein was expressed in *Escherichia coli* BL21 and purified by chromatography on Ni-NTA (Cytiva), HiTrap SP HP (Cytiva), and HiLoad Superdex 200 16/60 (Cytiva) columns and determined purity through polyacrylamide gel electrophoresis. The 55 nt guide RNA (gRNA) was transcribed in vitro and then purified by TRIzol (Invitrogen) and verified integrity through denaturing urea polyacrylamide gel electrophoresis.

### Antibodies

The following list of antibodies was used in this study:
- Mouse monoclonal antibody VP4 (2G4) (Harry B. Greenberg, Stanford University. PMID: 2431540). Dilution 1:1000.
- Mouse monoclonal antibody VP7 (M60) (Harry B. Greenberg, Stanford University. PMID: 2431540). Dilution 1:2000.
- Mouse monoclonal antibody VP7 (159) (Harry B. Greenberg, Stanford University. PMID: 2431540). Dilution 1:2000.
- Mouse polyclonal antibody NSP2 (Made by our laboratory, PMID: 9645203; RRID: AB_2802096). Dilution 1:100.
- Rabbit polyclonal antibody NSP2 (Made by our laboratory, PMID: 9645203; RRID: AB_2802097). Dilution 1:2000.
- Rabbit polyclonal antibody NSP4 (Made by our laboratory, PMID: 18385250; RRID: AB_2802094). Dilution 1:1000.
- Rabbit polyclonal antibody NSP5 (Made by our laboratory, PMID: 9645203; RRID: AB_2802098). Dilution 1:2000.
- Goat anti-rabbit Alexa 568 (Invitrogen, A-11011). Dilution 1:10000.
- Goat anti-mouse Alexa 488 (Invitrogen, A-10680). Dilution 1:10000.
- Primary mouse anti-H3K27me (Abcam, ab6002). Dilution 1:200.
- Primary rabbit anti-H3K27ac (Active Motif, 39034). Dilution 1:200.

- Secondary goat anti-mouse STAR Red (Sigma-Aldrich, 52283). Dilution 1:500.
- Secondary goat anti-rabbit STAR Orange (Sigma-Aldrich, 41367). Dilution 1:500.
- Secondary anti-mouse Alexa 568 (Thermo, A11004). Dilution 1:400.
- Primary SCP-3 (D-1) antibody (Santa Cruz Biotechnology, SC-74569). Dilution 1:300.
- Monoclonal ANTI-FLAG ® BioM2-Biotin (Sigma-Aldrich, F9291) conjugated to quantum dots (Thermo, Q21361MP). Dilution 1:66665.

## Animals

CD1 mature (10- to 12-week old) male mice were used. Animals were maintained at 23 °C and 55 ± 15% humidity, with a 12-h light−12-h dark cycle. Water was always accessible. Animal and plant experimental procedures treated at the Instituto de Biotecnología (IBt) were approved by the Bioethics Committee of the Instituto de Biotecnología of the Universidad Nacional Autónoma de México (UNAM). Animal experimental procedures treated at the Department of Biomedicine and Prevention at Faculty of Medicine were approved by the "Ministero della Salute" of Italy, authorization n.701/2018-PR. Animal experimental procedures treated on the Neurobiology and Epigenetics Unit of the European Molecular Biology Laboratory were approved by the EMBL Rome Animal Facility in accordance with European and Italian legislations.

## CRISPR/Cas protein expression and purification

Nuclease-dead dCas12a from *Acidaminococcus* sp. (Addgene, #171668) fused to an N-terminal 6His-SUMO tag was expressed in *Escherichia coli* BL21. Cells were grown in Luria-Bertani broth at 37 °C and transferred to 12 °C when OD600 reached 0.8. After 1 h, IPTG was added to a final concentration of 1 mM. After 24 h growth, cell pellets were collected by centrifugation and stored at −70 °C until protein purification.

The pellet was thawed in a lysis buffer (20 mM Tris-HCl pH 8.0, 250 mM NaCl, 10 mM imidazole) and sonicated for 6 min in 5 s ON-25 s OFF intervals. This was followed by centrifugation at 35,000 g at 4 °C for 35 min and filtration using membranes with 0.22 μm pore-size. The cell-free extract was injected into a Ni-NTA column (Cytiva) and eluted with an elution buffer: 20 mM Tris-HCl pH 8.0, 1 M NaCl, 250 mM imidazole. The protein was mixed with the SUMO protease and dialyzed overnight at 4 °C in dialysis buffer (50 mM phosphate buffer pH 6.0, 100 mM KCl, 5 mM MgCl$_2$, 10% glycerol, 2 mM DTT) to remove the 6His-SUMO tag. The cleaved protein was injected into a cation exchange HiTrap SP HP column (Cytiva) and eluted with a linear gradient from 0 to 50% IEX buffer (20 mM HEPES-KOH pH 7.2, 100 mM KCl, 5 mM MgCl$_2$, 10% glycerol, 2 mM DTT). The protein was further purified by size exclusion chromatography on a HiLoadSuperdex 200 pg 16/60 column (Cytiva) in storage buffer (20 mM HEPES-KOH pH 7.5, 500 mM KCl, 10% glycerol) and aliquots were stored at −70 °C until use. Purity and identity of dCas12a were confirmed via polyacrylamide gel electrophoresis and western-blot against the 6xHis region on dCas12a. Methods used for Supplementary Fig. S38.

## Production of crRNAs

The crRNA used in atomic force microscopy (AFM) experiments (Supplementary Fig. S42b) was produced by in vitro transcription of DNA templates previously amplified by PCR. The templates were produced using two self-complementary oligos (purchased from IDT).

Fw 5′-GAAATTAATACGACTCACTATAGGTAATTTCTACTCTTGTAGAT-3′ and

Rv 5′-CCCTGGTCAACCAGGTGAACAAGGATCTACAAGAGTAGAAATT-3′.

In vitro transcription and crRNA purification were performed using HiScribe T7 (NEB, E2040S) and RNA Cleanup (NEB, T2040S) kits, respectively. The crRNA pool for DNA curtains was produced according to the following procedure. Partially double stranded DNA templates for in-vitro transcription were obtained by hybridizing a 24 nt long forward oligo encoding the promoter for T7 RNA polymerase:

5′ GAAATTAATACGACTCACTATAGG,

with a pool of five 68 nt long reverse oligos (purchased from IDT):

5′-AUGAUGUUCUGCUGGAUAUGCACU-3′,
5′-CCUGACACCGGACGGAAAGCUGAC-3′,
5′-AAUGUCGGCUAAUCGAUUUGGCCA-3′,
5′-GCUAGCAAUUAAUGUGCAUCGAUU-3′ and
5′-AUGAACGCAAUAUUCACAAGCAAU-3′,

which encoded the crRNA sequence and the region complementary to the T7 promoter. Forward and reverse oligos were annealed at 1.5:1 ratio (Fw:Rv) in 10 mM Tris-HCl, pH 7.5, 50 mM NaCl, 1 mM EDTA buffer, incubated at 75 °C for 5 min and cooling to 25 °C during 25 min. HiScribe T7 kit was used for in vitro transcription and the crRNA pool was purified with TRizol (Ambion). Methods used for Supplementary Fig. S38.

## CRISPR/dCas12a nanoruler preparation

First, the CRISPR/dCas12a ribonucleoprotein (RNP) complex was formed. On a 0.6 mL microcentrifuge tube, the following reagents were added (final concentration): 1× CRISPR action buffer (TRIS-HCl 200 mM, NaCl 500 mM, DTT 5 mM) and dCas12a 20 nM. The gRNA was pre-heated at 90 °C for 1 min and then let cool to room temperature. Upon gRNA cooling, it was added to the dCas12a to a final concentration of 30 nM. The CRISPR/dCas12a components were incubated for 20 min at room temperature. When the complex was formed, the target dsDNA was added and the microtube was incubated at 37 °C for 1 h so the CRISPR/dCas12a complex binds to the target sequences in the dsDNA and thus forms the CRISPR/dCas12a nanoruler. Methods used for Supplementary Fig. S38.

## CRISPR/dCas12a nanoruler slide preparation for fluorescence microscopy

The sample imaging volume was delimited by a perforated double-sided tape attached to a coverslip and a slide on each side. The following reagents were prepared on an independent microtube: CRISPR/dCas12a nanorulers (described in the previous section), 20 μg/mL Hoechst33342, and 5 nM ssDNA fluorescent probe PS3[88] (5′-TCCTCCC-3′-ATTO 647 N, Integrated DNA Technologies') and graded with ddH$_2$O. The mix was transferred to the perforated double-sided tape in the coverslip and covered with the slide. The following sequence represents the dsDNA used for CRISPR/dCas12a binding, in which **bold** sequences corresponds to the association sites and *italics* are the protospacer adjacent motif (PAM) sequence:

AATTCTTAGGCACCCTTCTTTTTCTTCTTCTTCTTTTTCTTCTTT
TTCTTAGCACCTTGGCCGGCTCCAGCACCGGCTCCTTGACCAGCACC
AGCACCAGCACCTTGGCCGGCTCCAGCACCGGCTCCTTGACCAGCA
CCAGCACCAGCACCTTGGCCGGCTCCAGCACCGGCTCCTTGACCAGC
ACCAGCACCAGCACCTTGGCCGGCTCCAGCACCGGCTCCTTGACCAG
CACCAGCACCAGCACCTTGGCCGGCTCCAGCACCGGCTCCTTGACCA
GCACCAGCACCAGCACCAGCACCGGCTGGACCCTGGTTTCCTGG*TTT
A***CCTTGTTCACCTGGTTGACCAGGG**TTACCTGGCTGACCAGGGGAAC
CTTGGTTACCTGGAGAGCCTTGTGAACCTGGGGATCCAGGTTGACCA
TTCTTTCCAGGGTTACCCTGAGAACCTTGTGGACCGTTGGAACCTGG
CTCACCAGGTTGTCCGTTCTGACCAGGTTGACCAGGTTGACCTTCG
TTTCCTGGTTGACCTGGATTACCTGGAGAACCCTTGTTACCGGGCTG
TCCTTGGTTACCAGGAGATCCTGGGTTACCTGGCTCACCGGCTGGAC
CCTGGTTTCCTGG*TTTA***CCTTGTTCACCTGGTTGACCAGGG**TTACCT
GGCTGACCAGGGGAACCTTGGTTACCTGGAGAGCCTTGTGAACCTGG
GGATCCAGGTTGACCATTCTTTCCAGGGTTACCCTGAGAACCTTGTG
GACCGTTGGAACCTGGCTCACCAGGTTGTCCGTTCTGACCAGGTTGA
CCAGGTTGACCTTCGTTTCCTGGTTGACCTGGATTACCTGGAGAACC
CTTGTTACCGGGCTGTCCTTGGTTACCAGGAGATCCTGGGTTACCTG
GCTCACCGGCTGGACCCTGGTTTCCTGG*TTTA***CCTTGTTCACCTGGT

**TGACCAGGG**TTACCTGGCTGACCAGGGGAACCTTGGTTACCTGGAGA
GCCTTGTGAACCTGGGGATCCAGGTTGACCATTCTTTCCAGGGTTAC
CCTGAGAACCTTGTGGACCGTTGGAACCTGGCTCACCAGGTTGTCCG
TTCTGACCAGGTTGACCAGGTTGACCTTCGTTTCCTGGTTGACCTGG
ATTACCTGGAGAACCCTTGTTACCGGGCTGTCCTTGGTTACCAGGAG
ATCCTGGGTTACCTGGCTCACCGGCTGGACCCTGGTTTCCTGG*TTTA*
***CCTTGTTCACCTGGTTGACCAGGG***TTACCTGGCTGACCAGGGGAAC
CTTGGTTACCTGGAGAGCCTTGTGAACCTGGGGATCCAGGTTGACCA
TTCTTTCCAGGGTTACCCTGAGAACCTTGTGGACCGTTGGAACCTGG
CTCACCAGGTTGTCCGTTCTGACCAGGTTGACCAGGTTGACCTTCGT
TTCCTGGTTGACCTGGATTACCTGGAGAACCCTTGTTACCGGGCTGT
CCTTGGTTACCAGGAGATCCTGGGTTACCTGGCTCACCGGGTGCAC
CAGCACCGAGACCACAAGCTTCAGCTTCTCTCTTCTCGAGAGAT 3'.

Methods used for Supplementary Fig. S38.

## GATTA-PAINT and CRISPR/dCas12a nanoruler sample imaging

The GATTA-PAINT 40 RG nanoruler was provided as a single slide ready for imaging (GATTAquant DNA nanotechnologies). It has three fluorophores at a separation of 40 nm between them (ATTO 542/ATTO 655) and 80 nm between the furthest. Imaging was performed on an Olympus IX-81 inverted microscope using total internal reflection fluorescence (TIRF) illumination with a penetration depth of 200 nm (Olympus, cellTIRF Illuminator). Images were collected with an iXon 897 EMCCD camera (Model No. DU-879-CS0-#BV). A set of 300 frames were acquired at an exposure time of 50 ms per image, excitation laser of 488 and 561 nm with full laser power (23.1 mW measured at the back focal plane of the lens), and an effective pixel size of 160 nm in the object plane (Olympus UApo N 100×/1.49 numerical aperture, oil-immersion). For MSSR only the first 100 frames were analyzed.

The CRISPR/dCas12a nanoruler sample was visualized on the same imaging setup as the GATTA-PAINT 40RG nanoruler, except that a 20 ms as acquisition time was employed. Nearby emitters were automatically identified from t-MSSR$^3$-Var images using the Maximum Finder function of FIJI/ImageJ. A Maxima was accepted only if its intensity value (digital gray levels) was higher than a threshold value (prominence = 1900), in comparison with the intensity values from the ridge to a higher maximum. The coordinates of the identified local maxima (emitter's location) were computed from 16 regions of interest (1.5 μm$^2$ each) and exported to R to further quantify the intermitter's distances considering a worm-like chain model[48]. Briefly, the intermitter distances were computed for any pair of identified local maxima within the same t-MSSR$^3$-Var image. The CRISPR/dCas12a nanoruler system was design to with four binding sites for dCas12a distributed uniformly every 297 bp (equivalent to ~100 nm), hence, two emitters are considered to be part of the same dsDNA if their intermitter distance is shorter than the accumulated distance of four binding sites (300 nm). All measured intermitter distances were pooled on a single histogram and fitted in the context of Gaussian mixture models. Fitting was performed in R with the normalmixEM routine of mixtools with parameters μ: {$μ_1 = 100$, $μ_2 = 200$, $μ_3 = 300$} nm, and $σ = \sqrt{μ}$. Multiple fields of GATTA-PAINT and CRISPR/dCas12a nanorulers were imaged with similar results, but only one representative dataset for each sample is showcased in Fig. 7. Methods used for Fig. 7, Supplementary Fig. S43 and Supplementary Movie S1.

## AFM visualization

The RNP particle was assembled from dCas12a and crRNA (1:1.5 molar ratio) in AFM buffer (20 mM Tris-HCl pH 8.0, 100 mM NaCl, 15 mM MgCl$_2$, 1 mM DTT) at 37 °C for 20 min. The DNA template (1,500 bp) with four dCas12a target sites was added to the mix at 40:1 (RNP:DNA) molar ratio and incubated for 1 h at 25 °C. The sample was diluted 5-fold to a final concentration of 1 nM DNA and deposited on a freshly cleaved mica for 10 min, followed by rinsing with 0.5 mL filtered milli-Q water and air-drying. Images were acquired with an atomic force

microscope (NanoScope V, Bruker) on ScanAsyst-Air mode at room temperature and 1,024 samples/line. Images were processed with NanoScope Analysis Software v1.89. Methods used for Supplementary Fig. S42b. This experiment was performed once.

## DNA curtain assay

DNA from bacteriophage λ (λDNA) (NEB, N3011S) was mixed with biotinylated oligos complementary to the cohesive ends of λDNA in reaction buffer for T4 DNA ligase (NEB, M0202S), incubated at 70 °C for 15 min and cooled down to 15 °C, over 2 h. T4 DNA ligase was used for overnight ligation at room temperature. After ligase inactivation with 2 M NaCl, the biotinylated DNA was purified on a Sephacryl S-1000 size exclusion column (GE Healthcare).

The flowcell was passivated with a lipid solution (1.954% DOPC, 0.04% DOPE-mPEG2k and 0.006% DOPE-biotin) in buffer (10 mM Tris-HCL pH 8, 100 mM NaCl) for 30 min at room temperature. The flowcell was washed with BSA buffer (40 mM Tris-HCl pH 8, 2 mM MgCl$_2$, 0.2 mg/mL BSA) and incubated for 10 min. The biotinylated DNA in BSA buffer was injected into the flowcell and non-tethered DNA was washed out. BSA buffer supplemented with 100 mM NaCl, 5 mM MgCl$_2$, 2 mM DTT was used for imaging.

RNP particles were prepared by mixing dCas12a with the crRNA pool at 1:10 molar ratio in buffer (20 mM Tris-HCl pH 8.0, 100 mM NaCl, 5 mM MgCl$_2$, 2% glycerol, 2 mM DTT) at 37 °C for 30 min. The complex (10 nM) was injected into the flowcell for DNA binding during 30 min at room temperature. Labeling of dCas12a was achieved with anti-FLAG antibodies conjugated to quantum dots (Thermo, Q21361MP). C-S10-B was labeled with maleimide-Alexa488 at a single N-terminal cysteine.

Images were acquired at 60× with an inverted Nikon Ti-E microscope with 488 nm excitation laser. Emission was split with a 638 nm dichroic beam splitter (Chroma) and captured by two EM-CCD cameras (Andor iXon DU897). Images were processed with FIJI[89]. Methods used for Supplementary Fig. S38.

## Structured-illumination microscopy

The GATTA-SIM 140B nanoruler was provided as a single slide ready for imaging (GATTAquant DNA nanotechnologies). Spreads of germ cell chromosomes were performed according to Faieta[90]. In brief, testes were removed from euthanized animals, decapsulated, macerated in high-glucose MEM and mixed. The suspension was left to settle, and the supernatant was spun down at 7200 rpm for 1 min. The pellet was resuspended in 0.5 M sucrose and the suspension was added to slides coated with 1% paraformaldehyde in 0.015% Triton X-100 and incubated for 2 h in a humidified chamber at room temperature. At the end of the incubation, slides were rinsed twice in 1:250 Photo-Flo (Kodak, 1464510) in water and allowed to air dry. Surface chromosome spreads were either immediately processed for immunofluorescence or stored at −80 °C for up to 6 months. SYCP3 was stained using a primary antibody from Santa Cruz SC-74569 (SYCP3 D1) and a secondary antibody anti-mouse Alexa 568 (Thermo, A11004). Imaging of both the nanorulers and the mouse chromosomes was performed using an Elyra 7 microscope (Zeiss). Image acquisition was made with a 60 × 1.4 NA oil immersion objective and a 1.4× lens as extra magnification. Image reconstruction was done in ZEN Black with default parameters. Both the experiments involving the GATTA-SIM nanorulers and mouse germ cell chromosomes were performed once to obtain representative datasets. Methods used for Figs. 2c and 4d.

## Rotavirus cell infection and viral replication machinery immunofluorescence imaging

MA-104 Clone 1 cells (American Type Culture Collection; ATCC: CRL-2378.1; RRID:CVCL_3846) were cultured in DMEM-RS media supplemented with 5% fetal bovine serum at 37 °C and 5% CO$^2$. Prior to infection, Rhesus rotavirus (RRV) was activated with trypsin (10 μg/ml)

for 30 min at 37 °C. MA104 cells grown on glass coverslips were infected with RRV at a multiplicity of infection (MOI) of 1[54]. At six hours post infection, the cells were fixed and processed for immuno-fluorescence. The coverslips were mounted onto the center of glass slides with a STORM buffer mounting medium (1.5% glucose oxidase + 100 mM β-mercaptoethanol). All images were kindly provided by Garcés and collaborators[54]. Briefly, images of the rotavirus viroplasm were acquired on an Olympus IX-81 inverted microscope configured for total internal reflection fluorescence (TIRF) excitation (Olympus, cellTIRFM illuminator) using a critical angle such that the evanescence field had a penetration depth of 200 nm. The fluorophores Alexa Fluor 488 and Alexa Fluor 568 were excited with light of 488 nm and 568 nm respectively. The optical setup consists of an Olympus UApo N 100×1.4 NA, oil-immersion objective lens, with an extra 1.6× intermediate magnification lens. The images were acquired by an EMCCD camera (iXon 897, Model No: DU-897E-CS0-#BV; Andor) at a frequency of 20 fps and effective pixel size of 100 nm at the object plane. MSSR processing was performed considering the following parameters: AMP = 5, PSF = 3, order = 1. GPU parallel computing was enabled, and 100 images were used with t-MSSR-Mean. Methods used for Fig. 9a and Supplementary Fig. S32.

Note: ATCC:CRL-2378 cells were discovered to be contaminated with cells of African Green Monkey (AGM) origin. MA-104 was developed by initial explant culture from embryonic Rhesus Monkey kidney tissue. The cells were deposited at early passage to ATCC in the 90 s, and, after extended passage at ATCC, isoenzymology detected AGM cells. Observations suggest that the Rhesus Monkey cells were completely overtaken by the AGMs between passage 7 and 12. A pure population of Rhesus Monkey cells could not be obtained from the original deposit, and CRL-2378 was discontinued from the collection. However, the AGM subpopulation was cloned out, expanded and preserved as MA-104 Clone 1 (ATCC CRL-2378.1).

## Live imaging of sperm acrosomal exocytosis and F-actin dynamics

The non-capacitating medium (NC) used was a modified Toyoda–Yokoyama–Hoshi (modified TYH) which contains 119.3 mM NaCl, 4.7 mM KCl, 1.71 mM $CaCl_2.2H_2O$, 1.2 mM $KH_2PO_4$, 1.2 mM $MgSO_4.7H_2O$, 0.51 mM sodium pyruvate, 5.56 mM glucose, 20 mM HEPES and 10 μg/ml gentamicin. For capacitating conditions 15 mM $NaHCO_3$ and 5 mg/ml BSA were added (CAP).

Animals were euthanized and cauda epididymal mouse sperm were collected. Both cauda epididymis were cut at multiple sites and placed in 500 μl of NC. After 15 min incubation at 37 °C the epididymis were removed. Sperm were pre-incubated for 10 min in the presence of 100 nM SiR-actin in NC. Once loaded, sperm were incubated for another 60 min in CAP, the concentration of SiR-actin was 100 nM during the whole experiment.

Sperm were immobilized on concanavalin-A (1 mg/ml)-coated coverslips. The chamber was then filled with a recording medium (NC) containing 100 nM SiR-actin and 0.5 μM FM4-64. 100 images were obtained every 30 s for 20 min using the NanoImager S microscope (Oxford Nanoimaging Ltd), equipped with a 100×, 1.4 NA, oil-immersion objective (Olympus). For SiR-actin excitation, a 640 nm laser was used and for FM4-64 excitation a 561 nm laser was used. Effective pixel size at object plane = 117 nm. Sperm cells were imaged multiple times with similar results, but only a representative sequence is shown. Methods used for Fig. 9b, Supplementary Figs. S33–S34 and Supplementary Movies S4–9.

## Imaging of *Arabidopsis thaliana* root cells

*Arabidopsis thaliana* seeds were surface sterilized, germinated and grown in 0.2× Murashige and Skoog medium (prepared based on Linsmaier and Skoog medium L477; PhytoTechnology Laboratories, Lenexa, KS, USA), pH 5.7, supplemented with vitamins (0.1 mg l[-1]

pyridoxine, 0.1 mg l-1 nicotinic acid), 1% sucrose, and 0.8% agar. The plants were grown in a chamber at 21 °C, 16/8 h light/dark photoperiod and a light intensity of 105 μmol photons $m^{-2}s^{-1}$.

Confocal imaging of the double transgenic line, an F1 of a cross between plasmalemma pUBQ10::NPSN12-YFP[70] and nuclear p35S:H2B:RFP[91] marker lines, was performed with a Zeiss Axiovert 200 M microscope equipped with a C-APO ×63, 1.2NA objective (Oberkochen, Germany) and a coupled confocal system with a 488-nm laser source, a filter cube with 525/45 nm and 630/92 nm bandpass filters for yellow and red fluorescent protein emission, respectively, and a linear motor travel XY Stage and a Z-axis piezo stage with controllers (Thorlabs, Inc. Newton, NJ, USA). The XY pixel size of 404 nm and Z step size of 500 nm were implemented. Supplementary Fig. S39a–c show a Z-projection (sum) of ten slices, with the red and green channels represented by the "Cyan Hot" and "Magenta" look-up tables (LUTs), respectively.

A nuclear marker line, p35S:H2B:RFP, was imaged with an inverted Olympus FV1000-IX81 confocal microscope equipped with a LUMFLN×60, 1.3NA S objective. The 543 nm laser was used to excite RFP and emitted light was filtered with BA560–660. The oversampled XY pixel size of 41 nm and Z step size of 100 nm were implemented. The final image in Supplementary Fig. S39d was made by a Z-projection of 86 slices with MAX intensity mode and the LUT used was "Royal" of FIJI[89].

SPIM imaging of *A. thaliana* root cells was performed over a transgenic primary root expressing p35s:H2B-R with an in-house SPIM system inspired on the OpenSPIM project[92], with some setup modifications of the original design. Briefly, the illumination path consists of a C-flex laser combiner providing laser excitation sources at 405, 488, 561, 638 nm (Hubner Photonics, Cobolt Series 01–06, DPL for 561 and MDL for 405,488,638), which is coupled to the SPIM optics through a single multimode laser guide (Fiber optic with FC-APC output). Laser light is focused on a horizontal plane shaped via cylindrical lens (Thorlabs ACY254-050-A, f = 50.0 m ± 1%, Ø 25.4 mm, AR Coating: 350 – 700 nm). The focal plane of the cylindrical lens is imaged by a telescope in the back focal plane of the illumination objective (Olympus UMPLFLN10XW, 10X water immersion, NA = 0.3 mm WD = 3.5 mm). The excitation light-sheet is confined within the imaging area by a slit (Thorlabs VA100/M, Adjustable Mechanical Slits, Internal thread =2.4 mm), which is placed in the center of the telescope. The resulting light-sheet has a beam waist of about 3 μm in the focal plan of the illumination objective. The detection unit consists of a 20× water immersion objective (Olympus UMPLFLN20XW, NA = 0.5 mm WD = 3.5 mm), a tube lens (Ø60 mm × 104 mm), a multi-bandpass emission filter set (Semrock, FF01-446/523/600/677-25 BrightLine, Ø25 mm × 3.5 mm), and a sCMOS camera (Hamamatsu, ORCA-Flash4.0 V2 - Camlink 100fps). A 3D printed water filled imaging chamber (internal volume = 22 × 22 × 30mm without objectives) embodies the illumination objective and the detection objective aligned at 90°, a custom-made 3D printed sample holder and the sample.

Fluorescence excitation of the H2B-R-RFP expressing root cells was provided via the 561 nm laser light using stroboscopic illumination. SPIM volumetric imaging was achieved by mounting the sample on a four dimensional (XYZ, and Y rotation) motorized stacked stage (Picard Industries, USB 4D Stage, linear range = 9 mm, Includes Sample-Arm). Computer control of stroboscopic illumination, image acquisition and sample translation were provided by the OpenSPIM plugin 64-Bits of μmanager (v.1.4 for windows)[93]. Images were collected at a final pixel size of 0.325 μm, a z-step of 1.524 μm and a rotation step of 1.8°. MSSR processing was performed considering the following parameters: AMP = 10, PSF = 2, order = 0. Imaging of *A. thaliana* root cells was performed repeatedly but only one representative experiment was showcased. Methods used for Fig. 9d–e, Supplementary Figs. S39 and S40 and Supplementary Movie S12.

## Volumetric imaging of BPAE cells

FluoCells™ Prepared Slide #1 (BPAE cells with MitoTracker™ Red CMXRos, Alexa Fluor™ 488 Phalloidin, and DAPI, Thermo, #F36924) were imaged using the Nanoimager-S microscope (Oxford Nanoimaging Ltd), with a 100×, 1.4 NA, oil-immersion objective (Olympus) with and a sCMOS camera (Hamamatsu, ORCA-Flash4.0 V2 - Camlink 100fps). For DAPI, Alexa Fluor™ 488 phalloidin and MitoTracker™ Red CMXRos excitation, 405 nm, 473 nm and 561 nm lasers were used, respectively (Channel Splitter dichroic 561 LP. Emission Filter 1: 525/50, Emission Filter 2: Band 1 575–616.5). A set of 22 images (33 ms) were collected as a z-stack for each channel. Each frame was collected with a separation in z of 50 nm. Imaging of BPAE cells was performed several times but only one representative experiment was showcased. Methods used for Supplementary Fig. S41 and Supplementary Movies S13 and S14.

## Live-cell imaging of LLC-PK1 cells microtubule dynamics

LLC-PK1 (ATCC:CL-101) cells stably expressing mEmerald-EB3 were cultured and imaged using an ORCA-Fusion back-thinned sCMOS camera (Hamamatsu, C15440-20UP) and a 100×/1.47 NA oil-immersion objective (Plan-Apochromat, Zeiss) in a Zeiss Celldiscoverer 7 microscope with $CO_2$ and temperature control set at 5% and 37 °C, respectively. Fluorescence excitation was provided by a 488 nm laser at 1% laser power and emission light ($\lambda_{em}$ = 510 nm) was collected using a FITC filter. Image collection was done using the ZEN 3.2 (blue edition) acquisition software, with an exposure time of 100 ms, 2 s$^{-1}$ frame rate and a 43 nm pixel size. Stable cell lines were generated and provided by Michael W. Davidson[94,95]. Imaging of LLC-PK1 cells was performed repeatedly but only one representative experiment was showcased. Methods used for Supplementary Fig. S37 and Supplementary Movies S10 and S11.

## STED microscopy

Immunofluorescence imaging was carried out using a STEDYCON mounted on an upright Zeiss microscope in confocal or STED modes. Samples were imaged with a Zeiss 100×1.46 NA objective, 20 nm pixel size, 5 μs pixel dwell time, 15-line accumulations and a pinhole of 64 μm. Immunofluorescence was performed as in[45]. Primary antibodies used were anti-H3K27me (Abcam ab6002) and anti-H3K27ac (Active Motif, 39034), both at 1:200 dilution. Secondary antibodies used were anti-mouse labeled with STAR Red and anti-rabbit labeled with STAR Orange. STED laser powers were 3% of the 640 nm and 775 nm 96.5% for the STAR red channel, whereas for the STAR Orange channel was 7.8% of the 561 nm laser and 100% for 775 nm. Multiple cells were imaged with similar results but only one representative experiment was showcased. Methods used for Fig. 6.

## ArgoLight "Argo-SIM" test slide

Confocal images were acquired on a Zeiss LSM880 inverted microscope using a Plan-Apochromat 63×/1.4 Oil immersion objective, exciting the micropattern with the laser 405 nm and detecting the fluorescence in the range 420–480 nm. The pixel size was 0.044 micrometers. The same area has been acquired using the Airyscan detector with the same settings for laser power, detector gain, image format, bit depth, and line averaging, using a selective optical filter BP 420–480 + LP 605, and processing the images with the Airyscan algorithm (Zen Black, AIMApplication version 14.0.22.201) with strength parameter set to 6. Methods used for Fig. 4.

## PSFcheck imaging

We employed an immobile fluorescence pattern as a calibration sample for SRM[53]. The fluorescent patterns were fabricated using direct laser writing via infrared ultrashort-pulses that create regions of autofluorescence in a two-part epoxy mixture polymer sandwiched between a coverslip and a microscope slide (PSFcheck). One of the patterns present on a PSFcheck calibration slide consists of a 3D array of small diffraction limited shell features separated 10 μm from each other. The fluorescent thickness of a shell is small compared to the PSF so that the average FWHM across several features was calculated to be of 208 nm in a SIM microscope. We imaged this shell pattern using widefield fluorescence excitation on a NanoImager-S (Oxford Nanoimaging Ltd), equipped with a 100×, 1.4 NA, oil-immersion objective (Olympus). The PSFcheck sample was excited with a 561 nm laser and the emitted fluorescence acquired in the Emission Filter 2: Band 1 575–616.5 and recorded on a sCMOS Hamamatsu Orca Flash 4.0 V3. Acquisition time = 33 ms, effective pixel size at object plane = 117 nm. Imaging of the PSFcheck pattern was done once. Methods used for Fig. 8, Supplementary Figs. S21–23,S25,S27–31, and Supplementary Movies S2 and S3.

## ORCA Flash 4.0 V3 sCMOS detector characterization and Automatic Correction of sCMOS-related Noise (ACsN)

The fixed noise patterns of a sCMOS detector characterize the offset, variance and gain of each pixel in so-called calibration maps. The offset and variance are the average and variance, respectively, of the digital pixel-wise values that result from a video where no photons hit the detector. The gain is a multiplicative value of the signal when photons are detected. To characterize the maps of the sCMOS ORCA Flash 4.0 V3 detector, a code was written in the R programming language that implements the previously described calibration[96].

For the calculation of the offset and variance, a video of 60 thousand images was acquired without illumination on the NanoImager-S microscope (Oxford Nanoimaging Ltd) with no sample or laser turned on. In addition, a uniform fluorescent sample was used to recreate a uniform illumination of the detector and 5 videos of one thousand images each were taken. Each video had an average number of photons, chosen to be between 20 and 200 photons per pixel with even increments between each consecutive video as described by Huang et al.

ACsN is a noise correction method for sCMOS images that uses a principle of similarity between patches within the same image to characterize noise using 3D filtering[97]. The camera noise together with the signal from the incident photons can be represented by a distribution whose standard deviation is approximated based on frequency thresholds of the modulation transfer function (MTF). This threshold is calculated from the optical parameters of the system. The ACsN application was used in Matlab version 2020a. The input parameters were 1.4 of numerical aperture, wavelength of 610 nm, and pixel size of 117 nm, without video filter and with parallel computation. Methods used for Supplementary Figs. S30 and S31.

## SNR calculation for raw data

From the stacks of 100 images limited by diffraction, the average number of electrons per pixel was estimated for each image based on the following equation:

$$electrons_i = \frac{I_i - O_i}{G_i} \qquad (1)$$

Where O and G are the offset and gain maps, respectively. The signal-to-noise ratio (SNR) was calculated with the following equation:

$$SNR = \frac{QE*S}{\sqrt{QE^*(S+I_b) + N_r^2}} = \frac{electrons}{\sqrt{electrons + readout\ noise^2}}, \qquad (2)$$

where S are the photons per pixel and $I_b$ is the signal in the background. A quantum efficiency (QE) of 0.72 (to calculate photons) and reading noise $N_r$ = 1e- were used. S was considered as the average of photons in the region where the fluorescent ring is located, while $I_b$ is

the average value of the pixels that belong to the background of the image. Methods used for Supplementary Fig. S31.

## Entropy-based super resolution imaging (ESI)

ESI is a FF-SRM method that calculates the entropy of a sequence of fluorescence images and generates a magnified image that contains the actual information of the fluorophores. ESI is available as a plugin for ImageJ[13]. The ESI implementation allows you to create a super resolution image with a magnification of 2× the original size of the input images. The algorithm was iterated twice to achieve a magnification of 4. The input parameters are the number of final images in the output data; the number of bins per entropy, that is, the number of bins in the intensity histogram values for the entropy; and the order of the central moment.

For the first iteration of the algorithm the sequence of 100 images was used as input data, with the parameters: 50 images in result, 2 bins for entropy and order 0. The second iteration used the 50 images resulting from the first iteration, with parameters: 25 images in result, 2 bins per entropy and order 0. The ESI plugin returns the specified number of images and the average image. The average image from the second iteration is the image that is used for subsequent analyzes. Methods used for Figs. 5 and 8, and Supplementary Figs. S29–31.

## Multiple signals classification algorithm (MUSICAL)

MUSICAL is a FF-SRM method, implemented as a plugin for ImageJ[12], which improves resolution by singular values decomposition of a set of images taken from the same scene. This decomposition results in a collection of eigen-images and their respective eigen-values, where each eigen-image characterizes a specific pattern present in the image, and the eigen-value associated with that pattern is a statistical measure of the presence of that pattern in the underlying image. The signal from fluorophores in the scene is associated with patterns whose eigenvalues are large, while noise and background are associated with patterns whose eigenvalues are small. A predefined threshold divides the eigen-image set into range space (signal) and null space (noise). The sequence of 100 images of PSFcheck was used with parameters: 610 nm as emission wavelength, 1.4 numerical aperture, 1 in the magnification of the objective (digital size of the pixel is known), and 117 nm of pixel size, and 4 subpixels per pixel. The threshold value was −0.8 and was chosen from the singular value plot calculated by the plugin. Methods used for Figs. 5, 7 and 8, and Supplementary Figs. S29–31.

## Super resolution radial fluctuations (SRRF)

SRRF is a FF-SRM that overcomes the theoretical limit of diffraction by calculating the convergence of the gradient on a magnified version of the diffraction-limited image[11]. The degree of convergence for each sub-pixel is captured on a radiality map. In this first step, each diffraction-limited image has its corresponding radiality map, while in the second step, these maps are analyzed with a temporal function that improves the final resolution. SRRF is implemented as an ImageJ plugin[98]. The parameters used for this algorithm were: 0.5 in ring radius, magnification of 4, and 6 axes in the ring. The rest of the parameters were taken as default. Methods used for Figs. 5, 7 and 8, and Supplementary Figs. S29–31.

For ESI, MUSICAL, SRRF and MSSR super-resolution reconstructions, analyses were performed only once, as these approaches are deterministic.

## Super-resolution quantitative image rating and reporting of error locations (SQUIRREL)

SQUIRREL is an algorithm, implemented as an ImageJ plugin[14], that calculates the global Error (RSE) and Pearson correlation (RSP) values at the Super Resolution reconstruction against its scaled reference limited by diffraction. The RSP and RSE values were calculated using as reference the average image of the diffraction limited images used in the super resolution analysis for each case. These global resolution indexes are a measure of how reliable the reconstruction is related to the reference image. Methods used for Fig. 8 and Supplementary Figs. S28, S30 and S31.

## Simulation of fluorescent emitters

All 1D and 2D simulated emitters used for the examples shown in Figs. 1 and 2, and in Supplementary Figs. S6–10 were generated in Matlab version 2019b. The Gaussian and Bessel distributions of emitters were generated using the Gaussian ($PSF_G$) and Bessel ($PSF_B$) PSF, respectively, following the formulas:

$$PSF_G = exp\left(-\frac{(x - x_c)^2 + (y - y_c)^2}{2\sigma^2}\right) \quad (3)$$

Where $\sigma$ is the standard deviation, $(x,y)$ are the generated coordinates and $(x_c, y_c)$ is the center of the distribution.

$$PSF_B = I_0\left(2\frac{besselj(1,\upsilon)}{\upsilon}\right)^2 \quad (4)$$

Where $I_0$ is the maximum intensity of the distribution, $\upsilon$ is dimensionless distance and $besselj(1,\upsilon)$ is the Bessel function of first kind with dimensionless parameter $\upsilon$. To achieve enough spatial detail for visualization of the Gaussian emitter distribution, $\sigma = 10$ pixels was used in a square grid of size $81 \times 81$ pixels ($x = -4\sigma{:}4\sigma$, $y = -4\sigma{:}4\sigma$), with a step size of 1 pixel.

Given that the generated Bessel PSF is undefined by zero division at the center ($x = 0, y = 0$), its value is set to maximum intensity $I_0$ at this location. The dimensionless distance $\upsilon$ was computed following $\upsilon = k\frac{NA}{n}q$, where $\lambda$ is the emission wavelength, $NA$ is the numerical aperture, $n$ is the refractive index of the medium, $q$ is the radial distance to the distribution center and $k$ is the wavenumber, given by $k = \frac{2\pi}{\lambda}$.

For 1D emitters, $q = x$ and a 1D grid ranging from $-10^{-6}{:}10^{-6}$ was used. For 2D emitters, $q = \sqrt{x^2 + y^2}$ and a 2D grid of size $x = -10^{-6}{:}10^{-6}$, $y = -10^{-6}{:}10^{-6}$ was used. Note that, in either case, a step size is of 1 pixel = 1 nm = $10^{-9}$ m was used.

## Dip computation

Since the Gaussian distribution is fitted with good accuracy to a Bessel pattern, its use is sufficient to simulate the emitters. First, two Gaussian distributions with centers positioned at different locations along $(-x_c, 0)$ and $(x_c, 0)$ were simulated (using $\sigma = 10$ pixels). The distributions were then added, and the dip was computed as the intensity value at the center of the resulting distribution. The dip values in Figs. 1c and 2b of the main document were calculated by increasing the distance between the two emitters' distribution centers from 0 to $4\sigma$. Methods used for Figs. 1c and 2b, Supplementary Figs. S13–14.

## Image decorrelation (ImDecorr)

ImDecorr is an algorithm which computes spation resolution in a single image[34]. Its principle is based on partial phase autocorrelation by applying a mask filter and calculating cross-correlation coefficients in Fourier space. Its implementations are available in Matlab and as a plugin for ImageJ. It is a fast, friendly and easy-to-use tool free of user optical parameters to analyze both real and synthetic data. In this work, the plugin version of this algorithm for Matlab version 2021a was used for Fig. 6 and Table 1.

## Single particle tracking

Single particle tracking was performed on simulated images from the Particle Tracking Challenge (http://bioimageanalysis.org/track/).

Three different levels of SNR: 2, 4, 7 and three density levels of sub-diffraction particles: low, mid, high: 100, 500, 1000 particles per imaging field were used to simulate the images. Three classes of tracking algorithms were tested in TrackMate v7.6.1[65]:

(i) LAP: the LAP framework for Brownian motion[65].
(ii) LM: a linear motion tracker based on a Kalman filter[61,63].
(iii) NN: a tracker based on Nearest neighbors[99,100].

Particles were identified using the Laplacian of Gaussian (LoG) detectors, where 2 pixels were used as diameter of particles. For SNR = 2, LoG detection results were dominated by noise, hence, the histogram of detection quality was used to select a threshold that yielded the expected particle number of the dataset. For SNR > 2, the detection quality histogram was bimodal, so the threshold was selected at the dip between distributions.

Parameters for tracking algorithms were:
LM: initial search radius = 10, search radius = 7, max frame gap = 3.
LAP: max linkage distance = 7, max gap-closing distance = 10, max frame gap = 3.
NN: max search distance 10 to the nearest neighbor.

Tracking performance was assessed with the "tracking performance evaluation tool" deployed at Icy Icy 2.4 (http://icy.bioimageanalysis.com) and reported in Supplementary Fig. S36.

### Reporting summary

Further information on research design is available in the Nature Portfolio Reporting Summary linked to this article.

## Data availability

All raw data used or generated in this study have been deposited in the Zenodo OpenAIRE database and are available under unique accession codes, located in the Methods section of this manuscript. Source data are provided with this paper. The source code used to build the plots within this paper are available from the corresponding author upon reasonable request.

## Code availability

Source code for R, python, MATLAB and FIJI/ImageJ platforms is available at https://github.com/MSSRSupport/MSSR.

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

## Acknowledgements

To the Microscopy Facility at EMBL Rome and IBYME-CONICET for providing samples and materials for experiments. To the ECE and ICIMAF institutes for theoretical and experimental input to this work. We thank the LNMA staff for providing support with microscopy imaging. The OpenSPIM project was technically executed and extended by Oliver Valdez Escalona. We thank the Analytical and Quantitative Light Microscopy course staff from the Marine Biological Laboratory for providing samples and data collection assistance. We thank Paul Hernández, Luis Mochan, Federico Lecumberry, Angelica Flores Navarrete and Yuriney Abonza Amaro for their valuable feedback during the elaboration of the manuscript and review process. The technical assistance of M. Sc. Martín Patiño Vera is acknowledged. This research was supported by Dirección General de Asuntos del Personal Académico (DGAPA) – Programa de Apoyo a Proyectos de Investigación e Innovación Tecnológica-UNAM (PAPIIT), grants IN211821 to A.G., IN211216 to C.D.W., IN204221 to J.G.D., IN210121 to A.H.; and by Mexican Consejo Nacional de Ciencia y Tecnología (CONACyT), grant A1-S-9236 to J.G.D. Microscopy equipment was provided and maintained through CONACyT grants 123007, 232708, 260541, 280487, 293624 and 294781. C.W., M.G.B., and A.G. acknowledge the Chan Zuckerberg Initiative, grants: GBI-0000000093 (to C.D.W. and A.G) 2021–240504 (to M.G.B. and A.G). M.G.B. thank to IBYME-CONICET: Grants: PICT 2017–3047 (Agencia Nacional de Promoción de la Investigación, el Desarrollo Tecnológico y la Innovación), Fundación Williams; Fundación Rene Barón. D.K. acknowledges the support of the National Science Foundation grant 2102832. A.L. acknowledges the Howard Hughes Medical Institute (HHMI Award #GT12050-2019). We also acknowledge the Programa de Becas de Posgrado of CONACYT for granting scholarships to E.T.G., R.P.C., A.L., D.M., V.A., E.B.A., C.C.C., G.V.G., D.T., H.T.M., J.L.M., H.H., and to J.P.O. Y.G. acknowledges DGAPA/UNAM for postdoctoral fellowship.

## Author contributions

Conceptualization: E.T.G., A.G. Methodology: E.T.G., A.G., R.P.C., V.A., E.B.A., H.T.M., D.T., A.H.C., R.R.M., J.G.D., R.D.A. Investigation: E.T.G., A.G., R.P.C., A.H.C., D.M., V.A., E.B.A., C.C.C., G.V.G., A.H.G., D.T., J.M.R., Y.G., M.G.B., R.R.M., M.J., H.T.M., H.O.H., J.O.O., R.D.A., A.B. Visualization: E.T.G., A.G., R.P.C., H.T.M., V.A., E.B.A., A.L., D.T. Supervision: A.G., E.T.G., A.H.C., D.K. Funding Acquisition: A.G., A.D., M.G.B., J.G.D., C.D.W. Project Administration: A.G. Drafting Main Document: E.T.G., A.G., A.L., A.H.C., C.D.W. Drafting Supplementary Material: E.T.G., A.G., A.L., R.P.C., R.R.M., J.M.R., D.K., V.A., E.B.A., H.O.H. All authors contributed to the writing and reviewing of the manuscript.

## Competing interests

The authors declare no competing interests.

## Additional information

[1]Centro de Investigación en Ciencias, Instituto de Investigación en Ciencias Básicas y Aplicadas, Universidad Autónoma del Estado de Morelos, Cuernavaca, Morelos, Mexico. [2]Laboratorio Nacional de Microscopía Avanzada, Instituto de Biotecnología, Universidad Nacional Autónoma de México, Cuernavaca, Morelos, Mexico. [3]Analytical and Quantitative Light Microscopy, Marine Biological Laboratory, Woods Hole, MA, USA. [4]Instituto de Investigaciones Biomédicas, Universidad Nacional Autónoma de México, Ciudad de México, Mexico. [5]Departamento de Química de Biomacromoléculas, Instituto de Química. Universidad Nacional Autónoma de México, Ciudad de México, Mexico. [6]Instituto de Biología y Medicina Experimental (IBYME-CONICET), Buenos Aires, Argentina. [7]Departamento de Biología Molecular de Plantas, Instituto de Biotecnología, Universidad Nacional Autónoma de México, Cuernavaca, Morelos, Mexico. [8]Departamento de Genética del Desarrollo y Fisiología Molecular, Instituto de Biotecnología, Universidad Nacional Autónoma de México, Cuernavaca, Morelos, Mexico. [9]Instituto de Investigaciones en Matemáticas Aplicadas y en Sistemas, Universidad Nacional Autónoma de México, Ciudad de México, Mexico. [10]Department of Biomedicine and Prevention, Faculty of Medicine, University of Rome Tor Vergata, Rome, Italy. [11]Crick Advanced Light Microscopy Facility, London, UK. [12]Neurobiology and Epigenetics Unit, European Molecular Biology Laboratory, Monterotondo, Rome, Italy. [13]Instituto de Cibernética, Matemática y Física, Ciudad de la Habana, Cuba. [14]Electrical and Computer Engineering and School of Biomedical Engineering, Colorado State University, Fort Collins, CO, USA. ✉e-mail: adan.guerrero@ibt.unam.mx

