## [Peer Review File · Nature Communications]

REVIEWER COMMENTS

Reviewer #1 (Remarks to the Author):

This work adapts the concept of mean shift for designing an elegant and sophisticated algorithm for processing microscopy images. The authors not only explain the development of the concept to the proposed algorithm, that they call MSSR, but also provide pertinent implementation details, open software, operation manual, and a variety of conceptual and experimental examples.

My main concern about the work is the following. The single-frame MSSR algorithm performs smoothing, sharpening of features and noise suppression simultaneously, but not super-resolution. In this sense, it is similar to radially transform used in SRRF, where super-resolution is derived from temporal analysis and radially transform mainly achieves smoothing and sharpening. It may also be considered as a sophisticated and numerically stable approach for performing deconvolution. The fact that the algorithm works by the sharpening of features and not inherently performs super-resolution is also manifested in the fact that single-frame MSSR obtains only 1.6 times the resolution enhancement. This is related to the threshold of 65% in line 227 and clearly evident at the Rayleigh resolution limit. Any further superior resolution is achieved only through temporal analysis, or by applying MSSR on images obtained by other super-resolution algorithms. If the authors disagree with me, I request the authors to show that sf-MSSR (of any order of authors' choice) can resolve two fluorophores at distance, say 0.8 times the resolution limit. Otherwise, it is important that the paper be modified to remove the super-resolution claim or to restrict the claim of super-resolution to only t-MSSR.

Beyond that there are several minor concerns, which I believe are easily addressable:

1. It will be useful to include the resolution limits for the examples of nanorods and crispr examples, which are specifically being used to show super-resolution. For example, in Figure 2, it would have been nice to see also the resolution limit in the caption in order to appreciate the order of enhancement.
2. No details of the parameters used for the other algorithms are provided. The details of simulated examples are unavailable or not straight forward to find.
3. Lines 73-74, it might be useful to clearly restrict the super-resolution algorithms on which MSSR can be applied. My impression from the theoretical development is that MSSR cannot be applied to super-resolved imaged generated by localization microscopy and may be applicable to STED only if very small scanning step size is used. It may also have limited or no applicability to confocal and Airy scan microscopes (even though they may not be considered super-resolution microscopes).
4. It will be appreciable if the authors can comment on the applicability or potential of adaption for 3D microscopy, including 3D epifluorescence stacks, light sheet microscopy generated stacks.

5. It seems like the sentence in the lines 103-105 (The MS is a vector ..) is in a direct conflict with the phrase in the lines 106-107 (thus is not necessarily ...). It will be useful to either elaborate and clear the confusion or remove the second phrase completely.

6. There are quite a few places where notations appear before they are introduced. For example, $MSSR^0$ or sf-MSSR. There are some incorrect cross-references, for example, Supplementary Fig. S19-20 in lines 257-258. In line 157 of the supplementary file, the authors use 'Sections' instead of 'Supplementary notes'.

7. The analysis discussed in lines 203-208 is not easy to follow. It will be useful to rewrite this paragraph for better readability and apprehension.

8. Gattaquant and CRISPR samples, simulation samples, other control samples: It might be useful to either include the temporal fluctuation characteristics plot or discuss the nature of fluctuations (photokinetics) of the fluorophores. Are they blinking, what are the dark periods, etc? That definitely has an implication on how many frames and what resolution to expect. These factors, including the density of fluorophores emitting per frame may have a significant consequence on, for example, Figure S21.

9. Fig. S21 and Fig. S29 need better presentation. It is difficult to say one plot from the other. In Fig. S21, if need be, authors may want to limit the number of frames to 25, but better visibility of each plot is quite important. Potentially a smaller but more important subset of SNR values may be used.

10. I suppose including one result on biological samples in the main manuscript may be useful in improving the visibility of the manuscript when published.

In all, I appreciate the work presented in this article. However, the authors need to address my major concern on super-resolution claim. While super-resolution is attractive, this work holds value even without super-resolution claim on sf-MSSR and authors have done a good job in demonstrating the value. So, I believe that this manuscript is a significant contribution even without super-resolution claim and strongly encourage the authors to be technically correct on this matter. If this concern is addressed, and possible improvements on the minor comments are incorporated, I will be happy to recommend this manuscript for publication.

Reviewer #2 (Remarks to the Author):

This work presents MSSR, a principle and an algorithm for sharpening a digital image generated from diffraction of point sources. Given that the diffraction can be modelled with a Gaussian spread (PSF_G), the pixel intensities $I(p)$ is a superimposition of spreads from emitters. The goal is to identify the positions of the emitters, or at least reduce the spread. MS stands for mean shift, or sometimes called

the "blurring process". The idea is that by blurring, one can sharpen an image. The phase "sharpening by blur" is a procedure in Photoshop, at least since 2017.

The idea is that by selecting a "spatial bandwidth", h_s , (and on top of that, a "range bandwidth", h_r), one can identify a set of intensities in the color space, compute their mean, and then perform a mean shift, or update each pixel's intensity to the mean. If the pixel is not too far from the position of the emitter, the mean shift with the same Gaussian kernel may be in the same direction of the gradient in the intensity space, and the intensity may increase. Otherwise, the mean may go to the opposite direction and the pixel may lose its intensity. It would be nice and also important to relate the transformation of the point spread function (PSF_G), essentially the sigma parameter, to the mean-shift parameter h_r (or maybe h_s) in this work. This should be easy but is not done here.

Supplementary note 3 presents the "meanshift imaging model". Each pixel is a pair [position, intensity] of (spatial, range) components. Given a pixel $[x, y]$, a neighborhood is defined in (S17), which includes pixels within h_s spatial distance and also with h_r range distance. The asterisk in B^* probably excludes the host pixel. (S17) also excludes all neighboring pixels with the same intensity, which could be an error in the formulation.

(S19) defines the kernel density estimate f^\wedge at each pixel position. It may contain a typo: By V do you mean B ?

Since the positions of pixels are the grid, the k_s part of (S18), and the g_s part of (S23) are constants and the two-component profiles $k = k_s.k_r$ and $g = k'_s.k'_r$ are unnecessary and indeed when you reach (S24), only intensity is left. q_n in (S25-27) will be the same with or without k_s or k'_s . In other words, the only component in $[x, y]$ that moves is y , and no mean shift is done on x .

Since x is not shifted, and thus all dx are zero, Supplementary note 4 and its theorem, which relies on the shift of x , becomes irrelevant.

Letter to the reviewers

We thank both reviewers for their insightful feedback and efforts which have undoubtedly improved the quality of this work. The thoughtful discussions raised by them have increased the impact of our contributions to the field of optical microscopy. We have addressed crucial aspects related to the comprehensiveness of the manuscript. We appreciate the reviewers for raising critical concerns, which have now been rigorously reasoned through and addressed in the revised versions of the main manuscript, methods section, supplementary material and user's guide for MSSR.

Much care has been taken to address the general concern of reviewer #1 about the claim of whether MSSR can be considered a super-resolution microscopy approach. This led us to revisit our own understanding about the concept of super-resolution, and the ability of MSSR to surpass the diffraction limit. The applicability of MSSR to other, previously unconsidered, microscopy setups and imaging modes was a relevant aspect also brought up by reviewer #1. We have now incorporated four new figures in the main manuscript which show, quantitatively, the spatial resolution enhancement provided by MSSR when operating in combination with confocal, Airy scan, or STED microscopy configurations.

We deeply acknowledge the pivotal suggestions provided by reviewer #2 about the very fundamental mathematical principles over which MSSR operates. The suggestion to associate the PSF to the bandwidth led us to redefine our conceptualization of the h_s parameter, which directly improved the results provided by MSSR. Moreover, the questioning about the kernels motivated us to seek more convenient alternatives for explaining the basis of our algorithm. We appreciate reviewer #2 for pointing out errors in some of the formulas shown in the supplementary material, which have now been corrected and will help the readers understand the principles of MSSR more clearly.

Reviewer comments

Reviewer #1 (Remarks to the Author):

This work adapts the concept of mean shift for designing an elegant and sophisticated algorithm for processing microscopy images. The authors not only explain the development of the concept to the proposed algorithm, that they call MSSR, but also provide pertinent implementation details, open software, operation manual, and a variety of conceptual and experimental examples.

My main concern about the work is the following. The single-frame MSSR algorithm performs smoothing, sharpening of features and noise suppression simultaneously, but not super-resolution. In this sense, it is similar to radially transform used in SRRF, where super-resolution is derived from temporal analysis and radially transform mainly achieves smoothing and sharpening. It may also be considered as a sophisticated and numerically stable approach for performing deconvolution. The fact that the algorithm works by the sharpening of features and not inherently performs super-resolution is also manifested in the fact that single-frame MSSR obtains only 1.6 times the resolution enhancement. This is related to the threshold of 65% in line 227 and clearly evident at the Rayleigh resolution limit. Any further superior resolution is achieved only through temporal analysis, or by applying MSSR on images obtained by other super-resolution algorithms. If the authors disagree with me, I request the authors to show that sf-MSSR (of any order of authors' choice) can resolve two fluorophores at distance, say 0.8 times the resolution limit. Otherwise, it is important that the paper be modified to remove the super-resolution claim or to restrict the claim of super-resolution to only t-MSSR.

Beyond that there are several minor concerns, which I believe are easily addressable:

1. It will be useful to include the resolution limits for the examples of nanorods and crisp examples, which are specifically being used to show super-resolution. For example, in Figure 2, it would have been nice to see also the resolution limit in the caption in order to appreciate the order of enhancement.
2. No details of the parameters used for the other algorithms are provided. The details of simulated examples are unavailable or not straightforward to find.
3. Lines 73-74, it might be useful to clearly restrict the super-resolution algorithms on which MSSR can be applied. My impression from the theoretical development is that MSSR cannot be applied to super-resolved imaged generated by localization microscopy and may be applicable to STED only if very small scanning step size is used. It may also have limited or no applicability to confocal and Airy scan microscopes (even though they may not be considered super-resolution microscopes).
4. It will be appreciable if the authors can comment on the applicability or potential of adaptation for 3D microscopy, including 3D epifluorescence stacks, light sheet microscopy generated stacks.
5. It seems like the sentence in the lines 103-105 (The MS is a vector ..) is in a direct conflict with the phrase in the lines 106-107 (thus is not necessarily ...). It will be useful to either elaborate and clear the confusion or remove the second phrase completely.

6. There are quite a few places where notations appear before they are introduced. For example, MSSR⁰ or sf-MSSR. There are some incorrect cross-references, for example, Supplementary Figure S19-20 in lines 257-258. In line 157 of the supplementary file, the authors use 'Sections' instead of 'Supplementary notes'.
7. The analysis discussed in lines 203-208 is not easy to follow. It will be useful to rewrite this paragraph for better readability and apprehension.
8. Gattaquant and CRISPR samples, simulation samples, other control samples: It might be useful to either include the temporal fluctuation characteristics plot or discuss the nature of fluctuations (photokinetics) of the fluorophores. Are they blinking, what are the dark periods, etc? That definitely has an implication on how many frames and what resolution to expect. These factors, including the density of fluorophores emitting per frame may have a significant consequence on, for example, Figure S21.
9. Figure S21 and Figure S29 need better presentation. It is difficult to say one plot from the other. In Figure S21, if need be, authors may want to limit the number of frames to 25, but better visibility of each plot is quite important. Potentially a smaller but more important subset of SNR values may be used.
10. I suppose including one result on biological samples in the main manuscript may be useful in improving the visibility of the manuscript when published.

In all, I appreciate the work presented in this article. However, the authors need to address my major concern on super-resolution claim. While super-resolution is attractive, this work holds value even without super-resolution claim on sf-MSSR and authors have done a good job in demonstrating the value. So, I believe that this manuscript is a significant contribution even without super-resolution claim and strongly encourage the authors to be technically correct on this matter. If this concern is addressed, and possible improvements on the minor comments are incorporated, I will be happy to recommend this manuscript for publication.

Reviewer #2 (Remarks to the Author):

This work presents MSSR, a principle and an algorithm for sharpening a digital image generated from diffraction of point sources. Given that the diffraction can be modelled with a Gaussian spread (PSF_G), the pixel intensities $I(p)$ is a superimposition of spreads from emitters. The goal is to identify the positions of the emitters, or at least reduce the spread. MS stands for mean shift, or sometimes called the "blurring process". The idea is that by blurring, one can sharpen an image. The phase "sharpening by blur" is a procedure in Photoshop, at least since 2017.

The idea is that by selecting a "spatial bandwidth", h_s , (and on top of that, a "range bandwidth", h_r), one can identify a set of intensities in the color space, compute their mean, and then perform a mean shift, or update each pixel's intensity to the mean. If the pixel is not too far from the position of the emitter, the mean shift with the same Gaussian kernel may be in the same direction of the gradient in the intensity space, and the intensity may increase. Otherwise, the mean may go to the opposite direction and the pixel may lose its intensity. It would be nice and also important to relate the transformation of the point spread function (PSF_G), essentially the sigma parameter, to the mean-shift parameter h_r (or maybe h_s) in this work. This should be easy but is not done here.

Supplementary note 3 presents the "meanshift imaging model". Each pixel is a pair [position, intensity] of (spatial, range) components. Given a pixel $[x, y]$, a neighborhood is defined in (S17), which includes pixels within h_s spatial distance and also with h_r range distance. The asterisk in B^* probably excludes the host pixel. (S17) also excludes all neighboring pixels with the same intensity, which could be an error in the formulation.

(S19) defines the kernel density estimate \hat{f} at each pixel position. It may contain a typo: By V do you mean B ?

Since the positions of pixels are the grid, the k_s part of (S18), and the g_s part of (S23) are constants and the two-component profiles $k = k_s \cdot k_r$ and $g = k_s' \cdot k_r'$ are unnecessary and indeed when you reach (S24), only intensity is left. q_n in (S25-27) will be the same with or without k_s or k_s' . In other words, the only component in $[x, y]$ that moves is y , and no mean shift is done on x .

Since x is not shifted, and thus all dx are zero, Supplementary note 4 and its theorem, which relies on the shift of x , becomes irrelevant.

Response to Reviewer #1

Remarks to the Author: This work adapts the concept of mean shift for designing an elegant and sophisticated algorithm for processing microscopy images. The authors not only explain the development of the concept to the proposed algorithm, that they call MSSR, but also provide pertinent implementation details, open software, operation manual, and a variety of conceptual and experimental examples.

Q1.1. My main concern about the work is the following. The single-frame MSSR algorithm performs smoothing, sharpening of features and noise suppression simultaneously, but not super-resolution.

A1.1. Defining the concept of resolution in optical microscopy is not a simple task. It has been the subject of scientific scrutiny for a long time. However, there is no consensus in the scientific community about which approaches are considered super-resolution (SR), as discussed in [1]. Some scientists appraise that surpassing either Abbe's or the Rayleigh limits can achieve SR [2,3]. There is a view about the problem of spatial resolution, which makes a further segmentation of the resolution concept into three main categories: extended resolution (extended-R), enhanced resolution (enhanced-R), or SR [4]. Extended-R and enhanced-R, include those optical approaches that extend or enhance optical resolution down below Abbe's or the Rayleigh limits, i.e., by means of using clever illumination or detection schemes and image processing (i.e., 4Pi, I⁵M, SIM, Airyscan), but which never surpass the diffraction barrier [5-8]. SR is accomplished only by isolating the signal of individual fluorescent emitters transiting between ON/OFF either through deterministic (STED) or stochastic processes (i.e., STORM, PALM, PAINT, FF-SRM) [9-14], or by a combination of both (i.e., RESOLFT, MINFLUX and MINSTED) [15-17]. In the former definition of optical resolution [11], sf-MSSRⁿ and t-MSSRⁿ, surpass the Rayleigh limit [18]; hence they would be considered SR approaches. However, if we segment the problem in extended-R, enhanced-R, or SR, then sf-MSSRⁿ should be considered either extended-R or enhanced-R as its procedure embraces the analysis of diffraction-limited data. In stark contrast, t-MSSRⁿ is a FF-SRM approach that overcomes the diffraction barrier by statistically analyzing fluorescence intermittency in time.

The problem of spatial resolution in optical microscopy can also be addressed from a statistical point of view. In the case of fluorescence microscopy, the process of photon emission from point sources (fluorescence emitters) can be considered as a discrete distribution of information, where the unitary element of the distribution is the photon [19]. In this scenario, spatial resolution gets reduced to finding modes of information, regardless of the shape of the distribution, hence, disconnecting the problem of optical resolution from the diffraction boundary [20]. Furthermore, by computing the local magnitude of the Mean Shift vector, MSSR creates a probability distribution of fluorescence estimates whose local magnitude peaks at the source of information. As a result of that, the spatial distribution of information becomes 'refined', i.e., for a Gaussian distribution of fluorescence, its width shrinks.

Whether the refinement of information provided in sf-MSSRⁿ should be considered SR will depend on the scientific point of view embraced to assess the problem of spatial resolution in optical microscopy. However, we appreciate the feedback provided by reviewer #1, which led us to revisit our conceptualization of the spatial resolution in optical microscopy and to develop a rationale for the meaning of deconvolution at the nanoscopic scales (see A1.1.1, A1.1.2, A1.1.3, A1.1.4, A1.1.5).

In the following lines, we provide extra comparative data against: Wiener deconvolution [21], Richardson-Lucy deconvolution [22,23] and the Radiality Maps [24].

Q1.1.1. In this sense, it is similar to radiality transform used in SRRF, where super-resolution is derived from temporal analysis and radiality transform mainly achieves smoothing and sharpening.

A1.1.1. We agree with reviewer #1 because MSSR⁰ and the Radiality Maps (RMs) are similar in the sense that both perform sharpening and smoothing. However, we disagree with reviewer #1 that super-resolution, in either MSSR or SRRF, is mainly derived from the temporal analysis. Gustafsson et al. showed that RMs provide a resolution increase down to 0.7 times the Gaussian FWHM: [] ... *“radiality distribution is capable of distinguishing two Gaussian PSFs separated by ~ 0.7 times the Gaussian FWHM (1.7 times the PSF s.d. σ) ... [],”* [25]. Moreover, the authors of SRRF indicate that when using a pixel-by-pixel projection, such as the temporal radiality maximum (TRM) or temporal radiality average (TRA), there is no further gain in spatial resolution.

In the same note (*Supplementary note 1, specifically in section “Temporal Analysis – Generating the SRRF Map”*), the authors of SRRF state that *“In datasets where peak separation is greater than 0.7 times the FWHM of the PSF, emitters can be directly resolved by the radiality maps without further enhancement brought by higher-order temporal correlations”*, hence, either TRA or TRM serve the purpose of noise and artifact suppression, but not to exceed the spatial resolution gain provided by the RMs.

Figure A1.1a shows that both MSSR⁰ and the RMs overcome the Rayleigh diffraction limit, however, the RMs produce undesired artifacts which are absent when using MSSR⁰ (Figure A1.1b). The spatial artifacts introduced by SRRF have been revisited elsewhere [26]. Nonetheless, and despite the undesired imaging artifacts introduced by computing the RMs, they extend spatial resolution by analyzing a single diffraction limited image.

MSSR⁰ reliably provides spatial resolution gains, free of image analysis artifacts, down to the Rayleigh limit, hence, allowing the study of nanoscopic regimes at the boundaries of the Sparrow limit (Figure A1.1). MSSR of higher orders (MSSRⁿ: $n > 0$) encompass further analytical operations (subtraction, complement, negative and intensity weighting) which allow the observer to distinguish between single emitters separated by ~ 0.66 times the Gaussian FWHM of the PSF (1.55 times the PSF s.d. σ). Figure A1.1 has been added to Supplementary Material as Supplementary Figure 14.

Figure A1.1. Resolution increase provided by deconvolution methods, radiality maps and sf-MSSRⁿ. **a)** All methods tested decrease the dip successfully. The horizontal line represents the Rayleigh limit in relation to the dip height (for the sum of Gaussians, 2.5σ). This value decreases the most when using radiality maps and MSSR, where the best result reaches Rayleigh limit = 1.55σ (1.61 times smaller), provided by MSSR of third order (orange line). **b)** Bidimensional representations of the reconstructed images provided by each method tested, for emitters separated at distances of 1.55σ , 1.9σ and 2σ . Simulated emitters were generated with the gaussian distribution with a sigma of 121.10 nm (~ 10 pixels) using the following parameters: Refractive index of medium = 1.33, refractive index of oil = 1.515, Numerical aperture = 1.4, wavelength = 610 nm, pixel size 11.7 nm.

Q1.1.2 It may also be considered as a sophisticated and numerically stable approach for performing deconvolution.

A1.1.2. Figure A1.1 shows that none of the tested deconvolution algorithms, except for single-frame MSSR, reliably overcomes both Sparrow's and Rayleigh's limits of optical microscopy [18,27]. Hence, this demonstrates that MSSR generates results that reach SR scales (see A1.1.1 for further discussion about the meaning of the concept of SR).

We agree with Reviewer #1 that MSSR aims to revert the effect of diffraction on optical microscopy, so it must be considered a deconvolution process. However, what makes sf-MSSRⁿ unique is that it restores spatial information below the Sparrow limit. For example, Figure A1.1 shows that Wiener deconvolution partially deconvolves the diffraction effect but without a dramatic increase in spatial resolution. Interestingly, Richardson-Lucy deconvolution provides a noticeable increase in resolution at the boundaries of the Rayleigh limit but fails to restore spatial resolution down the Sparrow limit (Figure A1.1). With such data in hand, we conclude that sf-MSSRⁿ is the first deconvolution approach of optical microscopy that restores spatial information and can achieve nanoscale information.

Q1.1.3 The fact that the algorithm works by the sharpening of features and not inherently performs super-resolution is also manifested in the fact that single-frame MSSR obtains only 1.6 times the resolution enhancement. This is related to the threshold of 65% in line 227 and clearly evident at the Rayleigh resolution limit.

A1.1.3. We acknowledge reviewer #1 for stressing this point, which has also been raised by other readers of the preprint posted at bioRxiv [28]. We wish to clarify that the 65 % is not a threshold taken over the initial Gaussian distribution, as MSSR is not performing straightforward thresholding and normalization processes over it. Instead, MSSR shrinks the Dip formed in between the distribution shaped by two neighboring emitters, by computing the negative of the MS (-MS) and introducing a negative constraint over -MS, which is followed of by normalization and multiplication process (see steps i-iv in Supplementary Figure S11a in the supplementary material of the manuscript).

Figure A1.1.3 shows the procedure of thresholding and normalizing the joint distribution shaped by two adjacent emitters located at the Rayleigh limit, note that the dip decreases from a value of 0.8 to 0.57 (Figure A1.1.3 a and b). Hence, it is tempting to conclude that there is a noticeable increase in resolution as the Rayleigh limit has been surpassed. By repeating the same procedure using a joint distribution shaped by two adjacent emitters located at the Sparrow limit there is no further gain of resolution as the dip remains constant, taking the value of 1 (Figure A1.1.3 e and f). The Sparrow limit is associated to the maximum observable spatial frequency, hence, to the diffraction barrier. Noteworthy, processing the previous joint distributions with sf-MSSR of any order leads to a decrease of the dip value (Figure A1.1.3 e and f). The latter observation becomes striking when studying the case of sf-MSSR³ processing which collapses the dip to zero for both Rayleigh and Sparrow conditions. Such rationale that illustrates how MSSR works by sharpening of features down below the diffraction barrier has been included as supplementary material (See Supplementary Note 5). Figure A1.1 has been added to Supplementary Material as Supplementary Figure 13.

Figure A1.1.3. Thresholding and normalizing processes are not enough to overcome the Sparrow limit. This figure shows the sequential decrease in dip height between two intensity Gaussian distributions from simulated point emitters (**a,e**). Thresholding and normalizing (**b,f**) provides an initial reduction in dip height, which continues to decrease as MSSR of higher order is applied (**c,d,g,h**). Upper and bottom rows depict the relative distance between the peaks of both distributions when either the Rayleigh (top) or Sparrow (bottom) criteria are met. Worth noticing is that higher-order MSSR provides a complete dip collapse to zero, separating the distributions of both emitters completely.

The sentence *“This threshold operation exerts influence on structures at σ , at about 65% of the intensity distribution of the emitters; values below this threshold will be considered as noise and set to zero value”* (lines 226-227) ..., has been substituted by the following sentence:

“The goal of this step is to remove an artifact caused by the MS calculation itself when applied to Gaussian and Bessel distributions. When calculating the complement of the resulting distribution, a valley (or depression artifact) is generated between the peak intensity and the tails, which lies in the negative values and is referred to as negative constraints. After this step, the artifact is removed”.

Q1.1.4. Any further superior resolution is achieved only through temporal analysis, or by applying MSSR on images obtained by other super-resolution algorithms.

A1.1.4. We partially agree with reviewer #1. Using the TRA or TRM pixel-wise temporal functions in combination with MSSRⁿ (t-MSSRⁿ) does not provide extra resolution enhancement. TRA provide t-MSSRⁿ reconstructions less influenced by noise (recommended in imaging scenarios of marginal signal-to-noise ratios) and TRM delivers reconstructions whose intensity distribution is less dominated by constantly emitting sources, i.e., fiducial markers. In stark contrast, higher spatial resolution regimes, ~ 0.15 times the FWHM of the PSF (PSF σ : [0.4 - 0.5]), are achieved by t-MSSRⁿ when encompassing pixel-wise temporal functions of higher-order statistics such as Variance (Var), Coefficient of Variation (CV), or SOFI (TRAC2-4).

Q1.1.5. If the authors disagree with me, I request the authors to show that sf-MSSR (of any order of authors' choice) can resolve two fluorophores at distance, say 0.8 times the resolution limit. Otherwise, it is important that the paper be modified to remove the super-resolution claim or to restrict the claim of super-resolution to only t-MSSR.

A1.1.5. We find of great value all the inquiries raised by reviewer #1, which strengthened the scientific quality of this research. We hope to convince the reviewer that the answers A1.1.1, A1.1.2, A1.1.3, A1.1.4 and this one, provide enough scientific arguments to justify the super-resolution claims about MSSR.

In A.1.1.1 we develop the idea that if we keep to the operational definition of spatial resolution provided by both the Rayleigh and Sparrow limits, then we can conclude that sf-MSSRⁿ can be considered as a SR approach. We also mention that we are aware and acknowledge that there are other definitions of resolution which consider physical aspects of the wave nature of light. Such arguments lead to the three categories to classify technologies that provide extended, enhanced, or super-resolved images.

We have changed some instances of super-resolution claims used in the main text, and replace them for the concepts of extended-R, enhanced-R and SR:

- The term extended resolution (extended-R) is used when describing the gain of spatial resolution provided by sf-MSSRⁿ.
- The term enhanced resolution (enhanced-R) is used when implementing sf-MSSRⁿ in combination with other methods which by itself provide extended or enhanced resolution, such as SIM (and its variants), and Airyscan.
- The term enhanced resolution (enhanced-R) is also used when implementing sf-MSSRⁿ in combination with other super-resolution methods (i.e., STED). The resulting image is considered as a super-resolved image (SR).
- The term SR is also used for any fluorescence fluctuation based super resolution microscopy (FF-SRM) approach, which includes t-MSSRⁿ, SOFI, SRRF, ESI and MUSICAL.

We hope that the use of segmented definitions for resolution will minimize confusion with the readers of the manuscript. Having stated that, we have included extra further experimental or analytical data which strengthen the argument that sf-MSSRⁿ restores nanoscopic information down below the Sparrow limit.

Now, we would like to tackle the following inquiry of reviewer #1: ... *“I request the authors to show that sf-MSSR (of any order of authors' choice) can resolve two fluorophores at distance, say 0.8 times the resolution limit”.*

To address the claim of resolving two fluorophores at 0.8 times the resolution limit, we will recall the theoretical case at the Rayleigh limit for a GATTA-SIM 140B nanoruler (Figure 2c of the main manuscript) with $\lambda_{em} = 525$ nm and NA = 1.4, is $d = (0.61 * 525) / 1.4 = 229$ nm (table A1.1.5). Computation of resolution on sf-MSSR⁰ using the Rayleigh criterion (at 0.81 of the Dip) gives a spatial resolution of 160 nm, which corresponds to a resolution change to 0.69 times (0.69x) the resolution limit. Using higher orders of sf-MSSRⁿ with $0 < n < 4$, gives a resolution change of 0.66 times the resolution limit (table A1.1.5).

With the data shown in table A1.1.5 we conclude that the inquiry of reviewer #1, about the resolution change provided by MSSR has been satisfied using simulated data. Table A1.1.5 also shows Resolution ($1/K_0$) computed from the maximum spatial frequency (K_0) measured on the GATTA-SIM 140B nanorulers (experimental data) through decorrelation [29] using the Image Decorrelation plug-in for Fiji/ImageJ. Note that sf-MSSRⁿ dramatically reduces the value of $1/K_0$ as a function of the n-order.

GATTA-SIM 140B	Resolution	diffraction limited	sf-MSSR ⁰	sf-MSSR ¹	sf-MSSR ²	sf-MSSR ³
	Rayleigh limit	229 nm (1x) 2.90 σ	160 nm (0.69x) 2.02 σ	152 nm (0.66x) 1.92 σ	148 nm (0.65x) 1.87 σ	146 nm (0.64x) 1.84 σ
Simulation	FWHM of PSF	192 nm 2.43 σ	114 nm 1.44 σ	84 nm 1.06 σ	56 nm 0.71 σ	38 nm 0.48 σ
	PSF sigma	79 nm 1 σ	44 nm 0.65 σ	34 nm 0.43 σ	24 nm 0.30 σ	16 nm 0.20 σ
Experiment	$1/K_0$	260 nm (1x)	58 nm (0.22x)	33 nm (0.13x)	21 nm (0.08x)	13 nm (0.05x)

Table A1.1.5 sf-MSSRⁿ extends spatial resolution on simulated and real experimental conditions. For simulated conditions the Rayleigh limit and Sparrow limit were computed by the simulation of two emitters. The values of FWHM of PSF (values in nm units) have been computed by measuring directly on the PSF distribution. The values of PSF sigma (values in nm units) have been computed by fitting a Gaussian distribution and reporting the corresponding sigma parameter. sf-MSSR parameters: AMP = 1, FWHM of PSF = 192 , order = 0 to 3. ImDecorr parameters for DL case: Rmin=0, Rmax = 0.7, Nr = 50, Ng = 10. ImDecorr parameters for sf-MSSRⁿ (n = 0-3): Rmin=0, Rmax = 0.3, Nr = 50, Ng = 10.

Q1.2 It will be useful to include the resolution limits for the examples of nanorods and crisp examples, which are specifically being used to show super-resolution. For example, in Figure 2, it would have been nice to also see the resolution limit in the caption in order to appreciate the order of enhancement.

A1.2. See A1.1.5. Table A1.1.5 and its corresponding description has been incorporated into the main manuscript. Panels b and c of Figure 2 of the main manuscript have been updated accordingly.

To further test the attainable resolution by sf-MSSRⁿ we used the ArgoLight test slide, acquiring images of the pattern formed by gradually spaced lines of fluorescent molecules (Argo-SIM, pattern E). The distance between lines increases from 0 to 390 nm with a step change of 30 nm: 0 nm, 30 nm, 60 nm, etc. Confocal Images (Zeiss LSM880) were acquired using a Plan-Apochromat 63x/1.4 Oil immersion objective, exciting the micropattern with the laser 405 nm and detecting the fluorescence in the range 420 – 480 nm. The pixel size was 44 nm. Figure A1.2 shows the application of sf-MSSRⁿ to the confocal microscopy images of the Argo-SIM micropattern. As expected, the Confocal acquisition allows to resolve parallel rows of fluorophores located at 240 nm. Remarkably, sf-MSSR⁰ processing extended spatial resolution down to 0.5 times the confocal resolution, allowing to discriminate parallel rows of fluorophores located at 120 nm. It is worth mentioning that this improvement comes at zero hardware cost, compared to other methods requiring specific optics/detectors such as the Airyscan [30] and Re-scan confocal microscopy [31]. Higher orders of sf-MSSRⁿ create a saddle point between parallel rows of emitters located in the range of 60 - 90 nm at the boundaries of the Rayleigh limit. Figure A1.2 has been added to the manuscript as Figure 3.

Figure A1.2. sf-MSSR^n extends spatial resolution in confocal microscopy. **a)** Comparison of confocal and sf-MSSR^n reconstruction ($n=0-3$) of ArgoLight test slide applied to a spaced fluorescent line pattern. Central lines are gradually being separated by 30 nm (0 nm, 30 nm, 60 nm, ... 390 nm). **b)** Results of confocal and sf-MSSR^n reconstruction ($n=0-3$) of line patterns separated at 0 nm, 30 nm, 60 nm, 120 nm, 240 nm and 390 nm. **c)** Average profiles of images obtained in (b). Images in (a) and (b) correspond to the ensemble average of 31 consecutive sections (width = 1 μm) along the ArgoSIM micropattern E.

In conclusion, it was shown that sf-MSSR^n can extend the resolution limit in both simulated and real experimental conditions down below 0.8 times the resolution limit of optical microscopy. Figure A1.1.5 and its corresponding description has been incorporated into the main manuscript as Figure 3.

Q1.3. No details of the parameters used for the other algorithms are provided.

A1.3. The description of the following methods has been added to the corresponding sections of the manuscript and summarized within the corresponding figure legends.

- In the footnote of Figures 1 and 2a-b the following text has been added "For detail see Online Methods section *Simulation of fluorescent emitters*".
- In the footnote of Figure 2c the following text has been added corresponding to the detail of parameters "sf-MSSR parameters: AMP = 10, PSF FWHM of PSF = 3.48, order = 0-3. ImDecorr parameters for DL case: Rmin=0, Rmax = 0.7, Nr = 50, Ng = 10. ImDecorr parameters for sf-MSSRⁿ ($n=0-3$): Rmin=0, Rmax = 0.3, Nr = 50, Ng = 10". This text has been added to the footnote of the corresponding figure.
- In the footnote of Figure 5 (before Figure 3) the following text has been added corresponding to the detail of parameters "MSSR parameters: AMP = 1, PSF FWHM of PSF = 2, order = 0; ESI parameters: two iterations, first iteration: image output = 250, bin = 50, order = 2, second iteration: image output = 50, bin = 5 order = 2; SRRF parameters: default parameters; MUSICAL

parameters: $\lambda_{em} = 650$ nm, NA = 1.4, pixel size = 100 nm, threshold = - 0.6, $\alpha = 4$, subpixel per pixel = 4.”

- Figure 7 (before Figure 4) panels from b to f have been updated. In panel f, the results of ESI have been removed because of insufficient iterations of this method to get an image with signal and at the same time the desired dimension (20 times the original image).
- In the footnote of Figure 7 (before Figure 4), the following text has been added corresponding to the detail of parameters. “MSSR parameters: AMP = 20, PSF FWHM of PSF for Gattapaint ATTO 488 = 2.79, Gattapaint ATTO 550 = 3.31, Gattapaint ATTO 655 = 3.74, Order = 3; ESI parameters; SRRF parameters: Default parameters; MUSICAL parameters: emission wavelength = same as ATTO emission wavelength (520, 575, 680) used for each row in e) and f), NA = 1.49, pixel size = 160 nm, threshold = selected according to singular values plot, $\alpha = 4$, subpixel per pixel = 4.”
- Figure 8 (before Figure 5). The parameters AMP = 4, PSF FWHM of PSF = 2.77, order = 1 have been added to the legend.

Q1.4 The details of simulated examples are unavailable or not straightforward to find.

A1.4. The description of the following methods has been added to the corresponding section of the manuscript as follows:

Online methods

- ArgoLight “Argo-SIM” test slide.
- STED microscopy.
- Live-cell imaging of LLC-PK1 cells microtubule dynamics.
- Simulation of fluorescent emitters.
- Dip computation.
- PSFcheck imaging.
- Entropy-based Super Resolution Imaging (ESI).
- Multiple Signals Classification Algorithm (MUSICAL).
- Super Resolution Radial Fluctuations (SRRF).
- Super-resolution Quantitative Image Rating and Reporting of Error Locations (SQUIRREL).
- Automatic Correction of sCMOS-related Noise (ACsN).
- ORCA Flash 4.0 V3 sCMOS detector characterization.
- SNR calculation for raw data.
- Image Decorrelation (ImDecorr).

Q1.5. Lines 73-74, it might be useful to clearly restrict the super-resolution algorithms on which MSSR can be applied.

A1.5. MSSR addresses the problem of spatial resolution in optical fluorescence microscopy from the statistical point of view, where the process of photon emission from punctual sources is considered as a discrete distribution of information, provided by detected photons. Enhanced or extended resolution is achieved by finding local modes of information without taking any prior about the local shape of the distribution. The latter makes the MSSR principle compatible with any imaging approach where the distribution of information is of discrete nature (i.e., photons), and emanates from discrete sources (i.e.,

fluorophores). In the discussion section of the manuscript, we discuss some extra imaging applications suitable for MSSR processing, including broader applications of optics, not restricted to optical microscopy.

Q1.6. My impression from the theoretical development is that MSSR cannot be applied to super-resolved images generated by localization microscopy

A1.6. Agree. SMLM images are built through rendering localization tables of single emitters. Normally, this processing encompasses either a smoothing or an interpolation step, where the degree of smoothing can be set up according to the precision of the localization of single emitters [32,33]. Further applying MSSR to these images would decrease the FWHM provided by the selected rendering algorithm.

We note that an interesting avenue might be using sf-MSSRⁿ prior to rendering a SMLM reconstruction. The Mean-Shift has been used for SMLM [34-36], it has been recently implemented for drift correction [37]. We have not assessed whether sf-MSSRⁿ preprocessing of DL images would improve the resolution of SMLM reconstruction. Nonetheless, we have addressed a similar problem found in the realm of Single Particle Tracking (SPT) of diffraction limited objects. Supplementary Figures S35 and S36 show that MSSR improves SPT by increasing the localization precision and decreasing the FWHM of their individual fluorescence distributions. Based on that, it may be possible to pre-process SMLM data with MSSR and then perform SMLM analysis afterwards aiming to gain further resolution through improving the capabilities of SMLM algorithms to localize individual emitters. We are exploring this further for a subsequent work.

Q1.7. MSSR may be applicable to STED only if a very small scanning step size is used.

A1.7. To examine achievable resolution by sf-MSSR in combination with Stimulated-emission depletion (STED) microscopy we set to explore the capabilities to discern posttranslational modifications of histone proteins (H3K27me3, H3K27ac) within the 2-cell stage embryo in mice. Immunofluorescence imaging was carried out using a STEDYCON mounted on an upright Zeiss microscope in confocal, or STED modes. Samples were imaged with a Zeiss 100x 1.46 NA objective, 20 nm pixel size, 20 nm pixel size, 5 us pixel dwell time, 15-line accumulations and a pinhole of 64 um. Immunofluorescence was performed as in [38]. Primary antibodies used were anti-H3K27me3 (Abcam ab6002) and anti-H3K27ac (Active Motif, 39034), both at 1:200 dilution. Secondary antibodies used were anti-mouse labeled with STAR Red and anti-rabbit labeled with STAR Orange. STED laser powers were 3% for the 640 nm and 775 nm 96.5% for the STAR red channel, whereas for the STAR Orange channel was 7.8% of the 561 nm laser and 100% for 775 nm.

Figure A1.7 a, b, show that STED, but not confocal imaging allows to discern chromosomal territories within the chromatin of 2-cell stage embryo, as observed by the lack of colocalization of acetylated (ac, i.e., 'active') or methylated (me, i.e., 'inactive') chromatin states, H3K27me3, and H3K27ac respectively. Remarkably, Figure A1.7c shows that sf-MSSR¹ processing of the confocal image allows to reach a similar experimental conclusion, that there is no colocalization between H3K27me3 and H3K27ac signals.

Figure A1.7 also shows the result of post-processing of the STED image with sf-MSSR¹ (Figure A1.7g). We note an increase in contrast and resolution from confocal to STED and then to STED + sf-MSSR¹ (compare panels e, f, and g in Figure A1.7). To assess the resolution change provided by either STED or STED + sf-MSSR¹ we used the image decorrelation approach [29], which allows to compute the maximal observable frequency in an image (K_0) as a proxy of spatial resolution (Resolution $1/ K_0$). The bottom panels of Figure A1.7 e, f, and g, show the resolution computed at a confocal image (315 – 330 nm), STED (80 – 100 nm) and STED + sf-MSSR¹ (19 – 22 nm). With these results, we conclude that sf-MSSR¹ processing of STED data provides a further increase in spatial resolution. Figure A1.7 and its corresponding description has been incorporated into the main manuscript as Figure 6.

Figure A1.7. sf-MSSR¹ enhances spatial resolution in STED microscopy. a) Confocal and b) STED micrographs of histone proteins (H3K27me3, H3K27ac) imaged in a 2-cell stage mice embryo (top-center). c) and d) show the corresponding super-resolved scenes provided by post-processing of the confocal and STED images with sf-MSSR¹, respectively. The bottom row shows the sequential increase in resolution when comparing the same sample imaged with a (e) confocal microscope, an (f) STED microscope and (g) after processing the STED image with

sf-MSSR¹. For each image, the resolution range (in nanometers) was assessed using the ImDecorr algorithm in ImageJ. Immunofluorescence Imaging was carried out using a STEDYCON mounted on an upright Zeiss microscope in confocal or STED modes. Samples were imaged with a 20 nm pixel size. Primary antibodies: anti-H3K27me3 (Abcam ab6002) and anti-H3K27ac (Active Motif, 39034). Secondary antibodies: anti-mouse labeled with STAR Red and anti-rabbit labeled with STAR Orange. The parameters used for sf-MSSR processing were AMP = 5, order = 1, FWHM of PSF = 4, interpolation = bicubic, mesh-minimization = true.

Q1.8. It may also have limited or no applicability to confocal and Airyscan microscopes (even though they may not be considered super-resolution microscopes).

A1.8. Figures A1.2 and A1.7. show that sf-MSSRⁿ extends the spatial resolution of confocal microscopy images. Figure A1.8 below shows examples of the application of MSSR to Airyscan microscopy data (same sample data as in Figure A1.2). The Argo SIM micropattern E was imaged using an Airyscan detector with the same settings for laser power, detector gain, image format, bit depth, and line averaging of Figure A1.2, using a selective optical filter BP 420-480 + LP 605, and processing the images with the Airyscan algorithm set on strength parameter 6. The pixel size was 44 nm.

Figure A1.8, shows the application of sf-MSSRⁿ to the Airyscan microscopy images of the Argo-SIM micropattern. Within the Airyscan processed image it is possible to resolve parallel rows of fluorophores located at 180 nm, but not 120 nm or less. sf-MSSR⁰ processing enhanced the spatial resolution of the same Airyscan data, allowing to discriminate parallel rows of fluorophores located at 120 nm, or less (Figure A1.8 a, b). Higher orders of sf-MSSRⁿ create a saddle point between parallel rows of emitters located in the range of 60 - 120 nm (Figure A1.8 b, c). Remarkably, compared with the confocal original data in Figure A1.2 where the last resolvable line pair is the one corresponding to a distance of 240 nm, the value of 120 nm obtained in Figure A1.8 by applying sf-MSSR⁰ to Airyscan processed data corresponds to a 2-fold improvement in resolution. The first reported applications of Airyscan technology allowed an improvement in resolution of 1.7X [30], while only following protocols claim that a 2-fold improvement might be achieved [39], compared to standard confocal detection.

We then applied sf-MSSR⁰ on a SIM image of sister meiotic chromatids of mouse chromosomes [40]. Similar to the results obtained above with Airyscan, Figure A1.8d shows that processing with SIM images with sf-MSSR⁰ enhances both the contrast and resolution of the final image. Figure A1.8. has been incorporated as Figure 4 for the main manuscript.

Figure A1.8 sf-MSSRⁿ enhances the resolution and contrast of Airyscan reconstructions. **a)** Comparison of Airyscan and sf-MSSRⁿ reconstruction ($n=0-3$) of ArgoLight test slide applied to a spaced fluorescent line pattern. Central lines are gradually being separated by 30 nm (0 nm, 30nm, 60 nm, ... 390 nm). **b)** Results of Airyscan and sf-MSSRⁿ reconstruction ($n=0-3$) of line patterns separated at 0 nm, 30 nm, 60 nm, 120 nm, 240 nm and 390 nm. **c)** Average profiles of images obtained in (b). **d)** Synapsed homologs of meiotic mouse chromosomes visualized by TIRFM (left), SIM (middle) and SIM + MSSR⁰ (right). Images in (a) a (b) correspond to the ensemble average of 31 consecutive sections (width = 1 μ m) along the ArgoSIM micropattern E. Resolution in **(d)** was measured with the image decorrelation plugin of FIJI/image [29].

Q1.9. It will be appreciable if the authors can comment on the applicability or potential of adaptation for 3D microscopy, including 3D epifluorescence stacks, light sheet microscopy generated stacks.

A1.9. Since MS theory is not limited by the number of dimensions, the extension of MSSR to three-dimensional (3D) can be achieved by adding a parameter h_z that also includes the coordinates on the z axis. However, this extension is not trivial. The diffraction in the axial plane is greater than the lateral plane, so the parameter h_z for the z-axis needs to be greater compared to the h_s for the xy axes.

Hence, the value of h_s for the z-axis will be related to the resolution limit of this axis. On the other hand, an important feature to get a reliable reconstruction when image magnification is performed, is that the number of pixels for each dimension must be increased proportionally.

Figure A1.9 shows a comparison of 3D reconstruction from a stack of epifluorescence images of bovine pulmonary artery endothelial (BPAE) cells. When a 2D image is processed by sf-MSSR¹, an improvement in spatial resolution and contrast is observed (Figure A1.8 a, b). On the contrary, when a stack is used for a 3D representation (images taken at z-planes), the resolution obtained by sf-MSSR along the z-axis is not as refined as in the xy-plane (fig. A1.9 c, d). Nonetheless, a noticeable increase in contrast is attained in the overall data set (Figure A1.9 d, see supplementary videos 11 and 12). To further extend the resolution of DL images in 3D, it is necessary to extend the current implementations of the MSSR algorithm to account 3D information for MSSR processing. Therefore, it is necessary to make a detailed study for MSSR in 3D. Figure A1.9. has been incorporated into the Supplementary Material as Supplementary Figure S41.

Figure A1.9. MSSR enhances spatial resolution in 3D fluorescence microscopy images. In this figure, a comparison between a diffraction-limited 3D stack (a,c) and the super-resolved (sf-MSSR) counterpart (b,d) is provided. BPAE cells were labeled with MitoTracker™ Red CMXRos (mitochondria, red), Alexa Fluor™ 488 - Phalloidin (actin, green) and DAPI (nuclei, blue) and imaged in a ONI Nanoimager using epifluorescence mode. 20 steps in Z were acquired to achieve a 1 microns 3D section of the sample. Bottom row provides a rotated view of the sample. sf-MSSR parameters: AMP = 10, FWHM of PSF = 2.77, order = 1. 3D reconstruction was obtained using napari viewer and adjusting the voxel size through the console commands. Scale bar = 10 μ m.

The outcome of sf-MSSR processing (2D) of 3D confocal or 3D light sheet microscopy is discussed in the Supplementary Note 11 (Supplementary Figure S40, Supplementary Videos 11 and 12). As in the epifluorescence case, the axial resolution obtained by sf-MSSR¹ is not as refined as in the lateral plane, even so, a characteristic increase in contrast is always attained in the overall data set.

The MSSR principle can be developed for explicit volumetric imaging by means of using an asymmetric kernel, which can be defined following the 3D lateral-axial aspect ratio of the PSF, but also, encompassing possible deformations of axial symmetry of the PSF (due to optical aberrations introduced by the sample or the imaging system). Handling optical aberration in 3D light sheet microscopy is an active field of research (i.e., deconvolution or adaptive optics), where we want to contribute through the further development of MSSR applications in 3D.

Q1.10. It seems like the sentence in the lines 103-105 (The MS is a vector...) is in a direct conflict with the phrase in the lines 106-107 (thus is not necessarily...). It will be useful to either elaborate and clear the confusion or remove the second phrase completely.

A1.10. We agree with reviewer #1. The text in the lines 103-105 is confusing and introduces a conflict with other sections. For this reason, these lines were rewritten. The older text was:

“The MS is a vector that always points towards the direction of the intensity gradient and its length provides a local measure of the fluorescence density and brightness [17-19]. Since the MS lies within the gradient space, its values depend on the difference between the central pixel of the neighborhood and the surrounding pixels and thus is not necessarily linked to the fluorescence intensity values of the raw image.”

The corrected text is:

“The MS is a vector that always points towards the direction of the intensity gradient and its length provides a local measure of the fluorescence density and brightness [17-19]; its magnitude depends on the value difference between the central pixel of the neighborhood and the surrounding pixels.”

Q1.11. There are quite a few places where notations appear before they are introduced. For example, MSSR⁰ or sf-MSSR. There are some incorrect cross-references, for example, Supplementary Figure S19-20 in lines 257-258. In line 157 of the supplementary file, the authors use 'Sections' instead of 'Supplementary notes'.

A1.11. We thank the reviewer for calling our attention to check for such kinds of errors. All notation and reference-related errors have been revisited and corrected. Some of them are:

- (a) MSSR of zero order term is now properly introduced and correctly denoted as **MSSR⁰**
- (b) Single-frame MSSR analysis of a given order n term is now properly introduced and correctly denoted as **sf-MSSRⁿ**
- (c) Temporal MSSR analysis of a given order n term is now properly introduced and correctly denoted as **t-MSSRⁿ**

- (d) Otherwise, using **MSSR** without prefixes or superscripts makes reference to the super resolution algorithm.
- (e) References to figures in the supplementary material are now correctly written in the main manuscript or supplementary material. Every supplementary “Figure X” has been rewritten as “Supplementary Figure X”.
- (f) In the Supplementary Material, every “Section X” has been rewritten as “Supplementary Note X”.

Q1.12. The analysis discussed in lines 203-208 is not easy to follow. It will be useful to rewrite this paragraph for better readability and apprehension.

A1.12. The paragraph has been rewritten to provide a clearer explanation of the results shown in Figure 6 (previously Figure 4). On the other hand, because the algorithm parameters changed, the distance values among nanorulers have also been updated. The older text was:

“To determine the distance between two dCas12a sites along the DNA chain we obtained the distribution of distances between dCas12a binding sites taking in consideration their unidimensional association to a semi-flexible polymer such as the DNA [28]. Estimated distances after t-MSSR³-Var in the CRISPR/dCas12a nanoruler are 85 ± 14 nm, 152 ± 21 nm, 232 ± 37 nm (Figure 4d). These results confirm that t-MSSR³ can successfully resolve nanoscopic distances.”

The corrected text is:

“Once the CRISPR/dCas12a nanorulers were super-resolved by t-MSSR³-Var (Figure 7c), we scanned all the individual emitters in close proximity to determine the average distances between different dCas12a associated to the dsDNA binding sites (theoretically interspaced by 100 nm, Supplementary Figure S41). Since DNA in solution is a semi-flexible polymer [4], the measured distances resulted in three different distributions: 91 ± 31 nm, 220 ± 52 nm, 323 ± 19 nm (Figure 7d), which accounts for dCas12a separated by one, two and three binding sites along the dsDNA, respectively. These results confirm that t-MSSR³ can successfully resolve nanoscopic distances.”

Q1.13. Gattaquant and CRISPR samples, simulation samples, other control samples: It might be useful to either include the temporal fluctuation characteristics plot or discuss the nature of fluctuations (photokinetics) of the fluorophores. Are they blinking, what are the dark periods, etc? That definitely has an implication on how many frames and what resolution to expect. These factors, including the density of fluorophores emitting per frame may have a significant consequence on, for example, Figure S21.

A1.13. We acknowledge reviewer #1 for raising this comment. Analyzing the temporal dynamics of fluorescence is central to achieving the highest attainable gain of resolution by fluorescence fluctuation based super resolution microscopy (FF-SRM) approaches (reviewed in [37]). FF-SRM methods analyze higher order temporal statistics looking to discern correlated temporal information (due to fluorescent fluctuations) from uncorrelated noise (i.e., noise detector). t-MSSRⁿ encompasses a pixel-wise temporal analysis which delivers further resolution gain, hence, t-MSSRⁿ can be considered an FF-SRM approach.

Figure A1.13. Temporal fluctuation characteristics of two GattaPaint Nanorulers. Overlay of DL image and SR reconstruction by t-MSSR Var (middle). Two GattaPaint ATTO 550 nanorulers have been highlighted in white color containing three binding sites, each nanoruler is distributed in two pixels corresponding to the equivalent image DL. Upper strokes show the temporal fluctuation of pixels that match with the nanorulers, lower strokes show the temporal fluctuation of each binding site for the two nanorulers. The comparison of the dark times can be seen (unbound DNA-fluorescent strand), when the diffraction-limited data is analyzed against the data resolved by sf-MSSR³.

Figure A1.13 shows the pixel-wise temporal dynamics of two GATTA paint nanorulers imaged for 15 seconds at a 50 ms interval. In the provided examples (nanorulers 1 and 2), the temporal dynamics of a nanoruler containing three binding sites can be studied by analyzing the signal fluctuations at two pixels (px1 and px2). Note that, by analyzing either px1 or px2, it is difficult to distinguish the temporal dynamics of individual PAINT binding sites. Notably, sf-MSSR³ processing allows to untangle, in space and in time, the fluorescence dynamics at individual PAINT binding sites, where the fluorescence signal peaks due to the transient binding of a fluorescent labeled DNA strand to its corresponding binding site within the nanoruler. Figure A1.13 has been added to Supplementary Material as Supplementary Figure 44.

Q1.14. Figures S21 and S29 need better presentation. It is difficult to tell one plot from the other. In Fig. S21, if need be, authors may want to limit the number of frames to 25, but better visibility of each plot is quite important. Potentially a smaller but more important subset of SNR values may be used.

A1.14. For Figure S21: To increase its clarity we have restricted the number of frames to 25 (see Figure A1.14.1). In addition, a smaller subset of SNR is now being considered, this allows the reader to compare the effect of SNR in RSP and RSE more straight forward. Figure A1.14.1 has been added to the supplementary material as Supplementary Figure 27.

Figure A1.14.1 Effect of SNR and number of frames (temporal analysis) over RSP and RSE indexes. RSP (left) and RSE (right) stabilize at about 25 and 5 frames for low and high SNR, respectively.

We changed the presentation of Supplementary Figure 29 (now Supplementary Figure S36). Note that, contrary to the theory that has been established in Supplementary Note 5.1.1, the FWHM of PSF to process these images is unusually high. This might be due to the size of the objects being larger than the PSF of the microscope itself (pixel-wise). Similarly, for the case of the comet-like structures analyzed in Supplementary Figure S37 (known to be ~ 25 nm in diameter [43]), given that the pixel size of those images is very small and the information associated with the diffraction of the objects is spread across multiple pixels, a large FWHM of PSF must be used (see Figure A1.15.2). This phenomenon is subject to further examination in the future.

Figure A1.14.2 Pre-processing with MSSR improves tracking of single particles. Tracking performance study using a data set of diffusing vesicles from the “Single Particle Tracking Challenge” generated at three different SNR: {2, 4, 7} and in three density levels of sub-diffraction particles: {low, mid, high} corresponding to {100, 500, 1000} particles per imaging field. The proficiency of three trackers {LAP framework, linear motion tracker, nearest neighbor} were tested over raw or pre-processed data with sf-MSSR (AMP = 1, FWHM of PSF = 11.4, order = 1). Tracking performance was assessed with the “tracking performance evaluation tool” deployed at Icy Icy 2.1.3.0 (<http://icy.bioimageanalysis.com>).

Q1.15. I suppose including one result on biological samples in the main manuscript may be useful in improving the visibility of the manuscript when published.

A1.15. Thank you for the suggestions. We have prepared three new figures (A1.2, A1.7, A1.8), which have been included into the main body of the manuscript as Figures 3, 6, and 4, respectively. In addition, a new figure which presents the main applications of MSSR has been added to the end of the manuscript as Figure 9 (see Figure A1.16). This summarizing figure provides the readers with a friendly and intuitive, yet informative graphical representation of what they can do with MSSR. We present examples of fluorescence images of different organisms acquired with various microscopy setups, which prove the range of possibilities that users can achieve with MSSR. One core property of MSSR is its compatibility with most commercially available microscopy instrumentation.

Figure A1.15.1 MSSR applications in fluorescence microscopy. MSSR operates over images acquired with most fluorescence microscopy modalities available (e.g., widefield, confocal, light-sheet, etc.), denoted by the text in green. It can be applied to achieve enhanced image resolution, SNR and contrast in live-cell imaging, single-particle tracking, deep tissue and volumetric imaging, among others (denoted by the text in pale blue). Some examples of these applications are detailed in the main text and Supplementary Note 10. Abbreviations: FF-SRM, Fluorescence Fluctuation-based Super-Resolution Microscopy; SMLM, Single-Molecule Localization Microscopy; STED, Stimulated Emission Depletion microscopy; SIM, Structured Illumination Microscopy; SPIM, Selective Plane Illumination Microscopy; EPI, Epifluorescence; HiLo, highly inclined and Laminated optical microscopy; TIRFM, Total Internal Reflection Fluorescence Microscopy; CLSM, Confocal Laser-Scanning Microscopy. Scale bars: **a)** 500 nm; **b)** 2 μ m; **c)** no scale provided; **d)** 2 μ m; **e)** 20 μ m, insets = 5 μ m.

Additionally, a new subsection (10.4) has been added to the Supplementary Material in order to support our discussion over MSSR providing image quality enhancement for SPT applications. Moreover, this new supplementary note serves as an additional example of MSSR performance over a diverse range of biological samples. For this, live porcine kidney (LLC-PK1) cells stably expressing plus-end-tracking protein EB3 fused to mEmerald were imaged to investigate the growth dynamics of microtubules at the tip. Figure A1.15.2 below (also Supplementary Figure S37) shows a comparison between a diffraction-limited scene of live LLC-PK1 cells displaying microtubule dynamics and its sf-MSSR super-resolved counterpart.

Figure A1.15.2 Nanoscopic, single-frame, live-cell imaging of microtubule dynamics in LLC-PK1 cells. LLC-PK1 cells stably expressing mEmerald-EB3 were cultured and imaged using a sCMOS sensor (Hamamatsu, C15440-20UP) and a 100x/1.47 NA oil-immersion objective in a Zeiss CellDiscoverer 7 microscope with CO₂ and temperature control set at 5% and 37°C, respectively. Fluorescence excitation was provided by a 488 nm laser at 1% laser power and emission light ($\lambda_{em} = 510$ nm) was collected using a FITC filter. The comet-like structures depict the activity of plus-end-binding protein 3 (EB3) as it coordinates microtubule growth dynamics. Right side contains two 2x enlargements of the areas depicted by the white squares. **a)** A microtubule organizing center (MTOC), from which microtubule growth emerges. **b)** Microtubule dynamics at the edge of a cell with a left-to-right growth sense (see Supplementary Movie S10). Stable cell lines were generated and provided by Michael W. Davidson. sf-MSSR parameters: AMP = 3, FWHM of PSF = 30, order = 0.

Q1.16. In all, I appreciate the work presented in this article. However, the authors need to address my major concern on super-resolution claims. While super-resolution is attractive, this work holds value even without super-resolution claim on sf-MSSR and authors have done a good job in demonstrating the value. So, I believe that this manuscript is a significant contribution even without super-resolution claim and strongly encourage the authors to be technically correct on this matter. If this concern is addressed, and possible improvements on the minor comments are incorporated, I will be happy to recommend this manuscript for publication.

A1.16. We thank reviewer #1 for his/her insightful comments and valuable suggestions. As part of this review process, we have provided new evidence that, both qualitatively and quantitatively, proves the capability of MSSR of pushing the resolution limit even further down to ~ 120 nm from the processing of a single fluorescence image (demonstrated with both synthetic and experimental data). Even though we

understand the concerns of reviewer #1, regarding whether super-resolution is in fact achieved through the processing of one image or not. The main message we want to provide to the reader is that MSSR (whether sf- or t-, of any order) extends/enhances the resolution attained with any tested application in optical fluorescence microscopy. We believe that the latter claim will have deep impact in the optical microscopy field, because it expands the global access to fluorescence nanoscopy as it can be potentially applied to any application of fluorescence microscopy, regardless of its experimental or instrumental complexity. We are excited to see further applications of MSSR processing in imaging context not addressed in this manuscript.

Response to Reviewer #2

Remarks to the Author: This work presents MSSR, a principle and an algorithm for sharpening a digital image generated from diffraction of point sources. Given that the diffraction can be modeled with a Gaussian spread (PSF_G), the pixel intensities $I(p)$ is a superimposition of spreads from emitters. The goal is to identify the positions of the emitters, or at least reduce the spread. MS stands for mean shift, or sometimes called the "blurring process". The idea is that by blurring, one can sharpen an image. The phrase "sharpening by blur" is a procedure in Photoshop, at least since 2017.

Q2.1. The idea is that by selecting a "spatial bandwidth", h_s , (and on top of that, a "range bandwidth", h_r), one can identify a set of intensities in the color space, compute their mean, and then perform a mean shift, or update each pixel's intensity to the mean. If the pixel is not too far from the position of the emitter, the mean shift with the same Gaussian kernel may be in the same direction of the gradient in the intensity space, and the intensity may increase. Otherwise, the mean may go in the opposite direction and the pixel may lose its intensity. It would be nice and also important to relate the transformation of the point spread function (PSF_G), essentially the sigma parameter, to the mean-shift parameter h_r (or maybe h_s) in this work. This should be easy but is not done here.

A2.1. We appreciate this suggestion, it leads us to rethink the seminal conceptualization of theory which supports the MSSR principle and algorithms. We redefine the definition of h_s . Before, its value was set as the distance to the first minimum on Figure S6-d in the supplementary document (we removed this panel on this figure). Now h_s is defined as the half of the FWHM of the PSF distribution (Figure A2.1).

In Fourier space, the objective lens of a microscope act as a finite aperture, the the optical transfer function of a traditional microscope can conveniently be represented by a circle, the diameter of which can be linked directly to the maximum observable spatial frequency K_0 , where $K_0 \approx 1/PSF_{FWHM}$, reviewed in [42]. The PSF can be simulated as an Airy pattern according to the features of the optical system: wavelength, numerical aperture, and the refractive index of the medium. It can also be measured experimentally from imaging diffraction limited fiducial markers. The following figure shows the relationship between the h_s and FWHM in a Bessel distribution. Figure A2.1 has been added to the supplementary material as Supplementary Figure S8.

Figure A2.1. Selection of spatial parameter h_s related to FWHM. Considering a PSF given by an Airy pattern, the h_s is chosen as the half of the FWHM.

We deeply thank reviewer #2 for this suggestion. The change in the definition h_s , had a profound impact on the overall resolution enhancement provided by MSSR. Hence, any MSSR reconstruction provided in Figures 1-9 of the main manuscript have been recomputed using the modified h_s domain. The same rationale was applied to figures in both the *Supplementary Material* and *User Manual*. Note that, in all cases there is a noticeable increase in the resolution (or contrast) attained. The overall conclusions of the main manuscript remained without change.

Q2.2 Supplementary Note 3 presents the "meanshift imaging model". Each pixel is a pair [position, intensity] of (spatial, range) components. Given a pixel $[x, y]$, a neighborhood is defined in (S17), which includes pixels within h_s spatial distance and also with h_r range distance. The asterisk in B^* probably excludes the host pixel. (S17) also excludes all neighboring pixels with the same intensity, which could be an error in the formulation.

A2.2. True. We have now changed the formula S17 from

$$B^*(p_0) = \{p = [x, y]: 0 < \|x_0 - x\| < h_s, 0 < \|y_0 - y\| < h_r\}$$

to

$$B^*(p_0) = \{p = [x, y]: 0 < \|x_0 - x\| \leq h_s, 0 \leq \|y_0 - y\| \leq h_r\}.$$

With the "greater or equal to" symbol, related to the expression $0 < \|x_0 - x\| \leq h_s$, we include all members of the spatial neighborhood, except to the host pixel.

With the "two greater or equal to symbols", related to the expression $0 \leq \|y_0 - y\| \leq h_r$, we include all intensities of the range of the neighborhood.

Q2.3 (S19) defines the kernel density estimate \hat{f} at each pixel position. It may contain a typo: By V do you mean B?

A2.3. Yes, a mistake was made here. We have now changed the formula S19 from

$$\hat{f}(x_0) = \frac{1}{Nh_s^d h_r^m} \sum_{p_n \in V_h^*(x_0)} k_h(p_0, p_n)$$

to

$$\hat{f}(x_0) = \frac{1}{Nh_s^d h_r^m} \sum_{p_n \in B_h^*(x_0)} k_h(p_0, p_n)$$

where the expression V_h^* was substituted for B_h^* , which is related to the neighborhood in the summation subscript.

Q2.4 Since the positions of pixels are the grid, the k_s part of (S18), and the g_s part of (S23) are constants and the two-component profiles $k = k_s \cdot k_r$ and $g = k'_s \cdot k'_r$ are unnecessary and indeed when you reach (S24), only intensity is left. g_n in (S25-27) will be the same with or without k_s or k'_s . In other words, the only component in $[x, y]$ that moves is y , and no mean shift is done on x .

A2.4 It is true that the pixels are positioned in a grid, but the values of k_s and g_s are not constant in formulas (S18) and (S23), respectively. This is because k_s depends on the normalized difference related to h_s between the central pixel p_0 and the member p_n of the neighborhood $B^*(p_0)$.

Figure A2.4a shows the normalized difference related to h_s in a neighborhood of radius equal to 5 pixels. Note that as you move away from the central pixel, the relative distance increases until you reach the maximum value equal to 1. If a Gaussian kernel is used, its values decrease as you move away from the center of the neighborhood, this fact is shown in Figure A2.4b. Figure A2.4 has been added to Supplementary Material as Supplementary Figure S1.

We wish to clarify that the uniform kernel is the only one that provides no variation in its values, which are always constant or equal to 1. Any other type of kernel, such as Gaussian or Epanechnikov, presents value variations along the neighborhood.

Figure A2.4. Values taken by k_s according to the location within a neighborhood. (a) Normalized distance of pixels in a neighborhood $B_h(p_0)$ related to the central pixel. The neighborhood has 5 pixels of radius. As the position moves away from the central pixel, the distance increases. (b) k_s calculated from (a). As the position moves away from the central pixel, values for k_s the distance decreases. Pixels beyond the neighborhood are set to zero value.

Q2.5. Since x is not shifted, and thus all dx are zero, Supplementary Note 4 and its theorem, which relies on the shift of x , becomes irrelevant.

A2.5. Answer A2.4 shows that there is a change in the values of k_s along the neighborhood. Therefore, it makes sense to shift the values of x along the neighborhood, which this leads to Δx being greater than zero. For this reason, we believe that Supplementary Note 4 and its theorem, which relies on Δx , is well justified.

New changes

Main manuscript

- New figures:
 - Figure 3: sf-MSSRⁿ extends spatial resolution in confocal microscopy,
 - Figure 4: sf-MSSRⁿ enhances the resolution and contrast of Airy scan and SIM reconstructions.
 - Figure 6: sf-MSSRⁿ enhances spatial resolution in STED microscopy.
 - Figure 9: Showcase of MSSR wide-range fluorescence microscopy applications.
- Other figures have been updated.
- Some texts have been rewritten in response to suggestions made by the reviewers.
- Changes in the manuscript have been highlighted in red.

Supplementary material

- A description of Fourier interpolation and its comparison with bicubic interpolation has been added to the Supplementary Material (see section 6.2).
- Potential adaptation of MSSR to 3D images has been added to Supplementary (see section 11).
- New supplementary figures:
 - Supplementary Figure S1: Behavior of k_s according to the distance in a neighborhood.
 - Supplementary Figure S8: Selection of spatial parameter h_s related to FWHM.
 - Supplementary Figure S13: Thresholding and normalizing processes are not enough to overcome the Sparrow limit.
 - Supplementary Figure S14: Resolution increase provided by deconvolution methods, radially maps and sf-MSSRⁿ.
 - Supplementary Figure S23: Fourier interpolation in 2D.
 - Supplementary Figure S24: sf-MSSR processing times for bicubic and Fourier interpolations.
 - Supplementary Figure S25: Comparison between Fourier and bicubic interpolation for temporal analysis on experimental dataset.
 - Supplementary Figure S37: Nanoscopic, single-frame, live-cell imaging of microtubule dynamics in LLC-PK1 cells.
 - Supplementary Figure S41: Comparison DL images of sf-MSSR reconstruction of epifluorescence BPAE cells for 2D and 3D images.
 - Supplementary Figure S44: Temporal fluctuation characteristics of two GattaPaint Nanorulers.

Note that when adding figures, the actual number of these has been shifted compared to the previous version.

- Most of the figures have been updated with the mpl-infierno LUT using FIJI/ImageJ.
- Some texts have been rewritten in response to the reviewers and improve the overall quality of the article.
- Changes in the document have been highlighted in red.

Manual

- Figures 3, 5, 6, 8 and 9 have been updated.
- Some texts have been rewritten in response to suggestions made by the reviewers.

Movies

- New movies:
 - Supplementary Movie 10: sf-MSSR video of live LLC-PK1 cells expressing mEmerald-EB3.
 - Supplementary Movie 11: sf-MSSR video of an apoptotic LLC-PK1 cell expressing mEmerald-EB3.
 - Supplementary Movie 13: DL reconstruction of epifluorescence BPAE cells for 2D and 3D images.
 - Supplementary Movie 14: sf-MSSR¹ reconstruction of epifluorescence BPAE cells for 2D and 3D images.
- Other movies have been updated.

MSSR Plugin Fiji/ImageJ and other codes

- With the new changes MSSR Plugin has been updated to version 2.0.0.
- “PSF_p” parameter has been redefined as “FWHM of PSF”.
- An option to select the type of interpolation has been added.
- An option related to perform intensity normalization on sequence of images has been added.
- The selection of batch analysis can now process several images in a given folder.
- Codes for Matlab, R, and Python have been updated according to the new changes on the MSSR theory.

References

1. Prakash, K. et al. Super-resolution structured illumination microscopy: past, present and future. *Phil. Trans. R. Soc. A* **379**: 202001432020014, (2021).
2. Gao, Y., Zu, C., Xie, X. & Yu, X. Information-theoretical resolution limit of a far-field subwavelength diffraction system. *Phys. Rev.* **A103**, 033519 (2021).
3. Siegel N. & Brooker, G. Single shot holographic super-resolution microscopy. *Opt. Express* **29**, 15953–15968 (2021).
4. Manton, J. “Answering some questions about structured illumination microscopy,” *Philos. Transactions Royal Soc. A: Math. Phys. Eng. Sci.* **380**, 20210109 (2022).
5. Tychinsky V. & Odintsov A. New concept of optical super resolution, in *Laser Dimensional Metrology: Recent Advances for Industrial Application*, M. J. Downs, International Society for Optics and Photonics, **2088**, 206 – 210. (1993).
6. Hell, S. W. & Wichmann, J. Breaking the diffraction resolution limit by stimulated emission: stimulated-emission-depletion fluorescence microscopy. *Opt. Lett.* **19**, 780–782, (1994).
7. Guerra, J. Super-resolution through illumination by diffraction-born evanescent waves,” *Appl. Phys. Lett.* **66**, 3555–3557 (1995).
8. Gustafsson, M., Agard, D., & Sedat, J. Sevenfold improvement of axial resolution in 3D wide-field microscopy using two objective-lenses, in *Three-Dimensional Microscopy: Image Acquisition and Processing II*, T. Wilson and C. J. Cogswell. International Society for Optics and Photonics, **2412**, 147 – 156 (1995).
9. Betzig, E. et al. Imaging intracellular fluorescent proteins at nanometer resolution. *Science* **313**, 1642–1645, (2006).
10. Hess, S., Girirajan, T., & Mason, M. Ultra-high resolution imaging by fluorescence photoactivation localization microscopy. *Biophys. J.* **91**, 4258–4272 (2006).
11. Rust, M., Bates, M. & Zhuang, X. Sub-diffraction-limit imaging by stochastic optical reconstruction microscopy (STORM). *Nat. Methods* **3**, 793–796 (2006).
12. Nieves, D., Gaus, K., & Baker, M. Dna-based super-resolution microscopy: DNA-PAINT. *Genes* **9** (2018).
13. Opstad, I. et al. Fluorescence fluctuation-based super-resolution microscopy using multimodal waveguided illumination. *Opt. Express* **29**, 23368–23380 (2021).
14. Bastian, C., Sampieri, A., Benavides, M., Guerrero, A & L. Vaca. Super-resolution microscopy for the study of store-operated calcium entry. *Cell Calcium* **104**, 102595 (2022).
15. Balzarotti, F. et al. Nanometer resolution imaging and tracking of fluorescent molecules with minimal photon fluxes. *Science* **355**, 606–612 (2017).
16. Sharma, R., Singh, M., Sharma, R. Recent advances in STED and RESOLFT super-resolution imaging techniques. *Spectrochimica Acta Part A: Mol. Biomol. Spectrosc.* **231**, 117715 (2020).
17. Weber, M. et al. MINSTED fluorescence localization and nanoscopy. *Nat. photonics* **15**, 361–366 (2021).
18. Rayleigh, L. On the theory of optical images, with special reference to the microscope,” *J. Royal Microsc. Soc.* **23**, 474–482 (1903).
19. Kulaitis, G. Munk, A. and Werner, F. What is resolution? A statistical minimax testing perspective on superresolution microscopy. *The Annals Stat.* **49**, 2292 – 2312 (2021).
20. Carrington, W, Lynch, R., Moore, E., Isenberg, G. Fogarty, K., & Fay, F. Superresolution three-dimensional images of fluorescence in cells with minimal light exposure. *Science* **268**, 1483–1487 (1995).
21. Gonzalez, R. & Woods, R. Digital image processing. (Addison-Wesley, 1992).

22. Richardson, W. Bayesian-based iterative method of image restoration. *J. Opt. Soc. Am.* **62**, 55–59 (1972).
23. Lucy, L. An iterative technique for the rectification of observed distributions. *J. Astron.* **79**, 745–754 (1974).
24. Parthasarathy, R. Rapid, accurate particle tracking by calculation of radial symmetry centers. *Nat Methods* **9**, 724–726 (2012).
25. Gustafsson, N., Culley, S., Ashdown, G., Owen, D., Pereira, P. & Henriques R. Fast live-cell conventional fluorophore nanoscopy with ImageJ through super-resolution radial fluctuations. *Nat. Commun.* **7**, 12471 (2016).
26. Marsh, R. et al. Artifact-free high-density localization microscopy analysis. *Nat. Methods* **15**, 689–692 (2018).
27. Sparrow, C. M. On Spectroscopic Resolving Power. *Astrophys. J.* **44**, 76, (1916).
28. Torres-García, E, et al. Nanoscopic resolution within a single imaging frame, bioRxiv (2021).
29. Descloux, A., Großmayer, K. & Radenovic, A. Parameter-free image resolution estimation based on decorrelation analysis. *Nat. methods* **16**, 918–924 (2019).
30. Huff, J. The Airyscan detector from ZEISS: confocal imaging with improved signal-to-noise ratio and super-resolution. *Nat Methods* **12**, i–ii (2015).
31. De Luca, G. et al. Re-scan confocal microscopy: scanning twice for better resolution. *Biomed. Opt. Express* **4**, 2644–2656 (2013).
32. Thompson, R., Larson, D., and Webb, W. Precise nanometer localization analysis for individual fluorescent probes. *Biophys. J.* **82**, 2775–2783 (2002).
33. Mortensen, K., Churchman, L., Spudich, J., & Flyvbjerg, H. Optimized localization analysis for single-molecule tracking and super-resolution microscopy. *Nat. Methods* **7**, 377–381 (2010).
34. Boulanger, J., Kervrann, C., Bouthemy, P., Elbau, P., Sibarita, J.-B., & Salamero, J. Patch-based nonlocal functional for denoising fluorescence microscopy image sequences. *IEEE Transactions on Medical Imaging* **29**, 442–453 (2010).
35. Khater, IMeng, F. Wong, T., Nabi, I., & Hamarneh, G. Super resolution network analysis defines the molecular architecture of caveolae and caveolin-1 scaffolds,” *Sci. Reports* **8**, 9009 (2018).
36. Delcanale, P., Miret-Ontiveros, B., Arista-Romero, M., Pujals, S., and Albertazzi, L. Nanoscale mapping functional sites on nanoparticles by points accumulation for imaging in nanoscale topography (PAINT). *ACS Nano* **12**, 7629–7637 (2018).
37. Fazekas, F., Shaw, T., Kim, S., Bogucki, R., and Veatch, S. A mean shift algorithm for drift correction in localization microscopy. *Biophys. Reports* **1** (2021). Publisher: Elsevier.
38. Bošković A., Bender, A., Gall, L., Ziegler-Birling, C., Beaujean, N., & Torres-Padilla, M.-E. Analysis of active chromatin modifications in early mammalian embryos reveals uncoupling of h2a.z acetylation and h3k36 trimethylation from embryonic genome activation. *Epigenetics* **7**, 747–757 (2012). PMID: 22647320.
39. Wu X., Hammer J.A. ZEISS Airyscan: Optimizing Usage for Fast, Gentle, Super-Resolution Imaging. In: Brzostowski J., Sohn H. (eds) *Confocal Microscopy. Methods in Molecular Biology* **2304**, (2021).
40. S. Yoon, E.-H. Choi, J.-W. Kim, & K. P. Kim. Structured illumination microscopy imaging reveals localization of replication protein A between chromosome lateral elements during mammalian meiosis. *Exp. & Mol. Medicine* **50**, 1–12 (2018).
41. Dertinger, T., Colyer, R., Iyer, G., Weiss, S. & Enderlein, J. Fast, background-free, 3d super-resolution optical fluctuation imaging (sofi). *Proc. Natl. Acad. Sci.* **106**, 22287–22292, (2009).
42. Pawlowska M. *et al.* Embracing the uncertainty: the evolution of SOFI into a diverse family of fluctuation-based super-resolution microscopy methods. *J. Phys. Photonics* **4** 012002 (2022).

43. Cooper, G. M., & Hausman, R. O. B. E. R. T. E.. A molecular approach. The Cell. 2nd ed. Sunderland, MA: Sinauer Associates (2000).

REVIEWERS' COMMENTS

Reviewer #1 (Remarks to the Author):

I congratulate the authors for a thorough revision and addressing each comment and concern in sufficient detail. One may argue over some technical aspect of one or two arguments. However, concerning the article itself, I am satisfied with the revised version of the manuscript except for one major point, that is the title of the manuscript after revision.

I believe that the current title is a misrepresentation of the work especially after the authors have revised the manuscript. The term 'nanoscopic' in the title generally implies super-resolution, which is a misnomer when used with 'single frame' in the title because the authors in the revised manuscript explain that they achieve only extended or enhanced resolution using a single image.

I request that the authors change the title of the manuscript so that it is not prone to misinterpretation. Once this is taken care of, I will be happy to recommend the article for publication.

Reviewer #2 (Remarks to the Author):

I think my concerns in the original review as Review #2 have been addressed by the revision.

The phrase "sharpening by blur" borrowed from Photoshop in my original review as a description of the method used in this work gets used in line 165, with a very fine definition in the following sentence "In MSSR, the computation of the MS is the blurring process used to sharpen the image". This may give the phrase a new and correct definition, which may be different from that of Photoshop, and may be worth of further scrutiny.

However, references [30] and [31] do not make precedents of the current work. First, [31] clearly is about "sharpening of blur", which is not "by blur". Secondly, these works may utilize temporal sets or series of images for super resolution, which may be additional available information that this work does not need. The paragraph will be fine without these references. Of course, we do not cite Photoshop because the meaning is different there.

Response to Reviewer #1

Remarks to the Author: I congratulate the authors for a thorough revision and addressing each comment and concern in sufficient detail. One may argue over some technical aspect of one or two arguments. However, concerning the article itself, I am satisfied with the revised version of the manuscript except for one major point, that is the title of the manuscript after revision.

Q1.1. I believe that the current title is a misrepresentation of the work especially after the authors have revised the manuscript. The term 'nanoscopic' in the title generally implies super-resolution, which is a misnomer when used with 'single frame' in the title because the authors in the revised manuscript explain that they achieve only extended or enhanced resolution using a single image. I request that the authors change the title of the manuscript so that it is not prone to misinterpretation. Once this is taken care of, I will be happy to recommend the article for publication.

A1. The title of the manuscript has been changed to "Extending spatial resolution within a single imaging frame".

Response to Reviewer #2

Remarks to the Author: I think my concerns in the original review as Review #2 have been addressed by the revision. The phrase "sharpening by blur" borrowed from Photoshop in my original review as a description of the method used in this work gets used in line 165, with a very fine definition in the following sentence "In MSSR, the computation of the MS is the blurring process used to sharpen the image". This may give the phrase a new and correct definition, which may be different from that of Photoshop, and may be worthy of further scrutiny.

Q2.1. However, references [30] and [31] do not make precedents of the current work. First, [31] clearly is about "sharpening of blur", which is not "by blur". Secondly, these works may utilize temporal sets or series of images for super-resolution, which may be additional available information that this work does not need. The paragraph will be fine without these references. Of course, we do not cite Photoshop because the meaning is different there.

A2.1. References [30] and [31] have been removed.